# Beam Enumeration: Probabilistic Explainability For Sample Efficient Self-conditioned Molecular Design

**Jeff Guo & Philippe Schwaller**
Laboratory of Artificial Chemical Intelligence (LIAC), Institut des Sciences et Ingénierie Chimiques
National Centre of Competence in Research (NCCR) Catalysis
École Polytechnique Fédérale de Lausanne (EPFL)
Lausanne, Switzerland
`{jeff.guo,philippe.schwaller}@epfl.ch`

## Abstract

Generative molecular design has moved from proof-of-concept to real-world applicability, as marked by the surge in very recent papers reporting experimental validation. Key challenges in explainability and sample efficiency present opportunities to enhance generative design to directly optimize expensive high-fidelity oracles and provide actionable insights to domain experts. Here, we propose Beam Enumeration to exhaustively enumerate the most probable sub-sequences from language-based molecular generative models and show that molecular substructures can be extracted. When coupled with reinforcement learning, extracted substructures become meaningful, providing a source of explainability and improving sample efficiency through self-conditioned generation. Beam Enumeration is generally applicable to any language-based molecular generative model and notably further improves the performance of the recently reported Augmented Memory algorithm, which achieved the new state-of-the-art on the Practical Molecular Optimization benchmark for sample efficiency. The combined algorithm generates more high reward molecules and faster, given a fixed oracle budget. Beam Enumeration shows that improvements to explainability and sample efficiency for molecular design can be made synergistic. The code is available at `https://github.com/schwallergroup/augmented_memory`.

## 1 Introduction

Molecular discovery requires identifying candidate molecules possessing desired properties amidst an enormous chemical space (Sanchez-Lengeling & Aspuru-Guzik (2018). Generative molecular design has become a popular paradigm in drug discovery, offering the potential to navigate chemical space more efficiently with promise for accelerated discovery. Very recently, efforts have come to fruition and a large number of works have reported experimental validation of generated inhibitors, notably for both distribution learning (Merk et al. (2018); Moret et al. (2021); Grisoni et al. (2021); Yu et al. (2021); Eguida et al. (2022); Li et al. (2022); Tan et al. (2021); Jang et al. (2022); Chen et al. (2022); Hua et al. (2022); Song et al. (2023); Moret et al. (2023); Ballarotto et al. (2023) and goal-directed generation (Korshunova et al. (2022); Yoshimori et al. (2021); Zhavoronkov et al. (2019); Ren et al. (2023); Li et al. (2023); Salas-Estrada et al. (2023) approaches. Perhaps now more than ever, existing challenges in explainability and sample efficiency offer an avenue to propel generative molecular design towards outcomes that are not yet possible. Specifically, if one can elucidate *why* certain substructures or molecules satisfy a target objective, the model's *knowledge* can be made actionable, for example, in an interplay with domain experts. Moreover, sample efficiency concerns with how many experiments, i.e., oracle calls, are required for a model to optimize the target objective. This is a pressing problem as the most informative high-fidelity oracles are computationally expensive, e.g., molecular dynamics (MD) for binding energy prediction (Wang et al. (2015); Moore et al. (2023). If a generative model can *directly* optimize these expensive oracles, the capabilities of generative design can be vastly advanced.

In this work, we propose Beam Enumeration to exhaustively enumerate the most probable token sub-sequences in language-based molecular generative models and show that valid molecular sub-structures can be extracted from these partial trajectories. We demonstrate that the extracted sub-structures are informative when coupled with reinforcement learning (RL) and show that this information can be made *actionable* to self-condition the model's generation by only evaluating sampled molecules containing these substructures with the oracle. The results show significantly enhanced sample efficiency with an expected small trade-off in diversity. Beam Enumeration jointly addresses explainability and sample efficiency. Our contribution is as follows:

1. We propose Beam Enumeration as a task-agnostic method to exhaustively enumerate sub-sequences and show that molecular substructures can be extracted.

2. We demonstrate that during the course of RL, extracted substructures are on track to yield high rewards and can be used for self-conditioned molecular generation. We extract structural insights from these substructures, thereby providing a source of explainability.

3. We perform exhaustive hyperparameter investigations (2,224 experiments and 144 with molecular docking) and provide insights on the predictable behavior of Beam Enumeration and recommend default hyperparameters for out-of-the-box applications.

4. We introduce a new metric: Oracle Burden, which measures how many oracle calls are required to generate N *unique* molecules above a reward threshold as one is often interested in identifying a small set of *excellent* candidate molecules amongst many *good* ones.

5. We combine Beam Enumeration with the recently reported Augmented Memory (Guo & Schwaller (2023) optimization algorithm and show that the sample efficiency becomes sufficient (up to a 29-fold increase) to find high reward molecules satisfying a docking objective with only 2,000 oracle calls in three drug discovery case studies.

## 2 RELATED WORK

**Sample Efficiency in Molecular Design.** Tailored molecular generation is vital for practical applications as every use case requires optimizing for a bespoke property profile. Over the past several years, so-called goal-directed generation has been achieved using a variety of architectures, including Simplified molecular-input line-entry system (SMILES) (Weininger (1988)-based recurrent neural networks (RNNs) (Olivecrona et al. (2017); Popova et al. (2018); Segler et al. (2018); Goel et al. (2021), generative adversarial networks (GANs) (Goodfellow et al. (2014); Sanchez-Lengeling et al. (2017); Guimaraes et al. (2018), variational autoencoders (VAEs) (Kingma & Welling (2022); Gómez-Bombarelli et al. (2018); Zhavoronkov et al. (2019), graph-based models (You et al. (2019); Jin et al. (2020); Mercado et al. (2021c); Atance et al. (2022), GFlowNets (Bengio et al. (2021a), and genetic algorithms (Fu et al. (2022a). However, while all methods can be successful in optimizing for various properties, the oracle budget, i.e., how many oracle calls (computational calculations) were required to do so, is rarely reported. To address this, Gao et al. (Gao et al. (2022) proposed the Practical Molecular Optimization (PMO) benchmark, which assesses 25 models across 23 tasks and enforces a budget of 10,000 oracle calls. Recently, Guo et al. proposed Augmented Memory (Guo & Schwaller (2023), which uses a language-based molecular generative model and achieves the new state-of-the-art on the PMO benchmark. In this work, Beam Enumeration is proposed as an addition to language-based molecular generative models, and we show that coupling it with Augmented Memory drastically improves the sample efficiency.

**Explainability for Molecules.** Explainable AI (XAI) (Gohel et al. (2021) to interpret and explain model predictions is a vital component for decision-making. Existing methods include Gradient-weighted Class Activation Mapping (Grad-CAM) (Selvaraju et al. (2017), which uses gradient-based heat maps for convolutional layers and Local Interpretable Model-agnostic Explanations (LIME) (Ribeiro et al. (2016), which uses a locally interpretable model. Other methods include permutation importance (Altmann et al. (2010), which measures the performance change when shuffling feature values, and SHAP values (Shapley (1953), which measure the average contribution of each feature to the model's prediction. For molecules, the Molecular Model Agnostic Counterfactual Explanations (MMACE) (Wellawatte et al. (2022) method was proposed to search for the most similar counterfactual (model predicts the opposite label) molecule. Recently, the pBRICS (Vangala et al. (2023) algorithm was proposed to decompose a molecule into functional groups and then applying

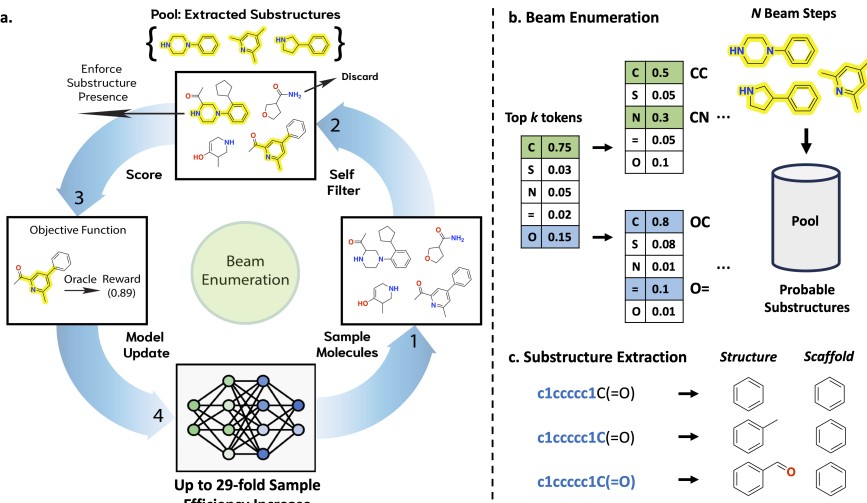

Figure 1: Beam Enumeration overview. **a.** The proposed method proceeds via 4 steps: **1.** generate batch of molecules. **2.** filter molecules based on pool to enforce substructure presence, discarding the rest. **3.** compute reward **4.** update the model. After updating the model, if the reward has improved for consecutive epochs, execute Beam Enumeration. **b.** Beam Enumeration sequentially enumerates the top $k$ tokens by probability for $N$ beam steps, resulting in an exhaustive set of token sub-sequences. **c.** All valid substructures (either by the *Structure* or *Scaffold* criterion) are extracted from the sub-sequences. The most frequent substructures are used for self-conditioned generation.

Grad-CAM to explain matched molecular pairs (MMPs), i.e., pairs of molecules differing by a single chemical group. In the context of generative models, previous works have explicitly addressed explainability (Guo et al. (2022); Fu et al. (2022b) and jointly with sample efficiency (Fu et al. (2022b). In this work, we aim to make explainability actionable *during* a generative design experiment.

To achieve this, we introduce *Beam Enumeration*, which extracts molecular substructures directly from the model's token sampling probabilities and derives explainability from a generative probabilistic perspective that is modulated by reward feedback. Our approach is based on the fact that during a successful optimization trajectory, it must become increasingly likely to generate desirable molecules. It is thus reasonable to hypothesize that the most probable substructures are *on track* to receiving high reward. We verify this statement in the Results section and show that Beam Enumeration can also jointly address explainability and sample efficiency.

## 3 PROPOSED METHOD: BEAM ENUMERATION

In this section, each component of Beam Enumeration (Fig. 1) is described: the base molecular generative model, the Beam Enumeration algorithm, and how Beam Enumeration harnesses the model's built-in explainability which can be used to improve sample efficiency through self-conditioned generation (further details on Beam Enumeration are presented in Appendix A).

**Autoregressive Language-based Molecular Generative Model.** The starting point of Beam Enumeration is any autoregressive language-based molecular generative model. The specific model used in this work is Augmented Memory (Guo & Schwaller (2023) which recently achieved the new state-of-the-art performance on the PMO (Gao et al. (2022) benchmark for sample efficiency, outperforming modern graph neural network-based approaches (Fu et al. (2020); Xie et al. (2021) and GFlowNets (Bengio et al. (2021b). Augmented Memory builds on REINVENT (Olivecrona et al. (2017); Blaschke et al. (2020) which is a SMILES-based (Weininger (1988) RNN using long-short-term memory (LSTM) cells (Hochreiter & Schmidhuber (1997). The optimization process is cast as an on-policy RL problem. We define the state space, $S_t$, as all intermediate token sequences and the action space, $A_t(s_t)$, as the token sampling probabilities (conditioned on a given state). $A_t(s_t)$ is given by the policy, $\pi_\theta$, which is parameterized by the RNN. The objective is to iteratively update the policy such that token sampling, $A_t(s_t)$, yields trajectories (SMILES) with increasing reward.

Formally, sampling a SMILES, $x$, is given by the product of conditional state probabilities (Equation 1), and the token sampling is Markovian:

$$P(x) = \prod_{t=1}^{T} P(s_t \mid s_{t-1}, s_{t-2}, \ldots, s_1) \tag{1}$$

The Augmented Likelihood is defined, (Equation 2) where the Prior is the pre-trained model and $S$ is the objective function which yields a reward given a SMILES, $x$.

$$\log \pi_{\theta_{\text{Augmented}}} = \log \pi_{\theta_{\text{Prior}}} + \sigma S(x) \tag{2}$$

The policy is directly optimized by minimizing the squared difference between the Augmented Likelihood and the Agent Likelihood given a sampled batch, $B$, of SMILES constructed following the actions, $a \in A^*$ (Equation 3):

$$L(\theta) = \frac{1}{|B|} \left[ \sum_{a \in A^*} (\log \pi_{\theta_{\text{Augmented}}} - \log \pi_{\theta_{\text{Agent}}}) \right]^2 \tag{3}$$

Minimizing $L(\theta)$ is equivalent to maximizing the expected reward as shown previously (Guo & Schwaller (2023); Fialková et al. (2022).

**Beam Enumeration.** Beam Enumeration is proposed based on two facts: firstly, on a successful optimization trajectory, the model's weights must change such that it becomes increasingly likely to generate high reward molecules. Secondly, generation involves sampling from conditional probability distributions. It is therefore reasonable to assume that the highest probability trajectories are more likely to yield high reward. Correspondingly, Beam Enumeration (Fig. 1) enumerates the top $k$ tokens (by probability) sequentially for $N$ beam steps (as it is infeasible to sample the full trajectories due to combinatorial explosion), resulting in an exhaustive set of token *sub-sequences*. We show that meaningful molecular substructures can be extracted from these sub-sequences, which we harness and demonstrate how it can be made *actionable*. We note the closest work to ours is the application of Beam Search (Graves (2012); Boulanger-Lewandowski et al. (2013) for molecular design (Moret et al. (2021) to find the highest probability trajectories. Our work differs as the objective is not to find a small set of the most probable sequences. Rather, we *exhaustively* enumerate the highest probability *sub-sequences* to extract molecular substructures for self-conditioned generation. We further detail the differences to Beam Search in Appendix G.

**Probabilistic Explainability.** Here, we describe how probabilistic explainability can be extracted from the exhaustive set of token sub-sequences. Usually, token sequences are only translated into SMILES once the sequence is complete, i.e., the "end" token has been sampled. We hypothesized that molecular substructures can be extracted from a given sub-sequence by iteratively considering every (sub)-sub-sequence (Fig. 1). For example, given the sub-sequence "ABCD", the set of (sub)-sub-sequences are: "A", "AB", "ABC", and "ABCD". In practice, we only consider (sub)-sub-sequences with at least three characters ("ABC" and "ABCD") since each character loosely maps to one atom and three is approximately the minimum for meaningful functional groups, e.g., "C=O", a carbonyl. It is expected that not every sub-sequence possesses (sub)-sub-sequences mapping to valid molecular substructures. Still, we show that a sufficient signal can be extracted (Appendix C). Finally, we implement two types of substructures: *Scaffold*, which extracts the Bemis-Murcko (Bemis & Murcko (1996) scaffold and *Structure*, which extracts *any* valid substructure. In the Results section, we discuss the predictable difference in behavior.

**Self-conditioned Generation.** The sub-sequences were enumerated by taking the most probable $k$ tokens, and the model's weights should be updated such that high reward molecules are increasingly likely to be generated. Correspondingly, it is reasonable to posit that the most frequent molecular substructures are on track to becoming high reward full molecules and that the substructures themselves possess properties aligned with the target objective. Beam Enumeration saves a *Pool* of these substructures and filters future sampled batches of molecules to contain them, discarding those that do not. Effectively, the generative process is self-conditioned as the model will only be updated by generated molecules containing the extracted substructures (Fig. 1).

## 4 METRICS

**Sample Efficiency Metrics.** We define two metrics to assess sample efficiency: Generative Yield (referred to as Yield from now on) and Oracle Burden. Yield (Equation 4) is defined as the number of *unique* generated molecules above a reward threshold, where $g \in G$ are the molecules in the generated set, $\mathbb{I}$ is the indicator function which returns 1 if the reward, $R(g)$, is above a threshold, $T$. Yield is a useful metric for drug discovery as it is ubiquitous to triage the generated set (thus, a higher Yield is desirable) and prioritize molecules, e.g., based on synthetic feasibility, for experimental validation or more expensive computational oracles.

$$Generative\ Yield = \sum_{g=1}^{G} \mathbb{I}[R(g) > T] \tag{4}$$

Oracle Burden (Equation 5) is defined as the number of oracle calls ($c$) required to generate $N$ *unique* molecules above a reward threshold. This is a direct measure of sample efficiency as high reward molecules satisfy the target objective, and the metric becomes increasingly important with expensive high-fidelity oracles. In this work, all Oracle Burden metrics are computed by not allowing more than 10 molecules to possess the same Bemis-Murcko (Bemis & Murcko (1996) scaffold, thus also explicitly considering diversity in the generated set.

$$Oracle\ Burden = c \mid \sum_{g=1}^{G} \mathbb{I}[R(g) > T] = N \tag{5}$$

## 5 RESULTS AND DISCUSSION

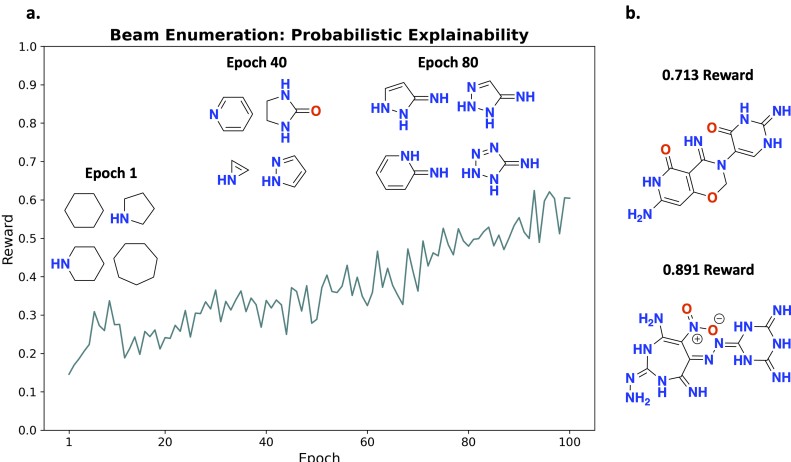

Figure 2: Illustrative experiment with the following multi-parameter optimization objective: maximize tPSA, molecular weight <350 Da, number of rings $\geq$ 2. **a.** Augmented Memory (Guo & Schwaller (2023) reward trajectory with annotated top-4 (excluding benzene) most frequent molecular substructure scaffolds at varying epochs using Beam Enumeration. **b.** Examples of molecules with high reward.

We first design an illustrative experiment to demonstrate the feasibility of Beam Enumeration to extract meaningful substructures and, in turn, enable self-conditioned generation. Next, three drug discovery case studies to design inhibitors against dopamine type 2 receptor (DRD2) (Wang et al. (2018), MK2 kinase (Argiriadi et al. (2010), and acetylcholinesterase (AChE) (Kryger et al. (1999) were performed to demonstrate real-world application. The key result we convey is that Beam Enumeration can be added directly to existing algorithms, and it both provides structural insights into why certain molecules receive high rewards and improves sample efficiency to not only generate more high reward molecules, but also faster, given a fixed oracle budget.

Table 1: Illustrative experiment: Beam Enumeration improves the sample efficiency of Augmented Memory. All experiments were run for 100 replicates with an oracle budget of 5,000 calls, and reported values are the mean and standard deviation. *Scaffold* and *Structure* indicate the type of substructure, and the number after is the *Structure Minimum Size*. Parentheses after Oracle Burden denote the cut-off number of molecules. Parentheses after values represent the number of unsuccessful replicates (for achieving the metric).

| Metric | Augmented Memory | | | | |
|---|---|---|---|---|---|
| | **Beam Scaffold 15** | **Beam Structure 15** | **Beam Scaffold** | **Beam Structure** | **Baseline** |
| Generative Yield$_{>0.7}$ ($\uparrow$) | **1757 $\pm$ 305** | 1669 $\pm$ 389 | 1117 $\pm$ 278 | 864 $\pm$ 202 | 496 $\pm$ 108 |
| Generative Yield$_{>0.8}$ ($\uparrow$) | **819 $\pm$ 291** | 700 $\pm$ 389 | 425 $\pm$ 256 | 199 $\pm$ 122 | 85 $\pm$ 56 |
| Oracle Burden$_{>0.7}$ (1) ($\downarrow$) | **577 $\pm$ 310** | 616 $\pm$ 230 | 1037 $\pm$ 414 | 897 $\pm$ 347 | 1085 $\pm$ 483 |
| Oracle Burden$_{>0.7}$ (10) ($\downarrow$) | 947 $\pm$ 350 | **926 $\pm$ 332** | 1881 $\pm$ 259 | 1745 $\pm$ 292 | 2392 $\pm$ 216 |
| Oracle Burden$_{>0.7}$ (100) ($\downarrow$) | **1530 $\pm$ 468** | 1547 $\pm$ 513 | 2736 $\pm$ 335 | 2713 $\pm$ 402 | 3672 $\pm$ 197 |
| Oracle Burden$_{>0.8}$ (1) ($\downarrow$) | **1311 $\pm$ 628** | 1401 $\pm$ 695 | 2423 $\pm$ 487 | 2295 $\pm$ 482 | 3164 $\pm$ 492 |
| Oracle Burden$_{>0.8}$ (10) ($\downarrow$) | **1794 $\pm$ 617 (1)** | 2009 $\pm$ 804 (1) | 3124 $\pm$ 497 | 3241 $\pm$ 492 | 4146 $\pm$ 326 |
| Oracle Burden$_{>0.8}$ (100) ($\downarrow$) | **2704 $\pm$ 689 (1)** | 2943 $\pm$ 811 (6) | 3973 $\pm$ 592 (6) | 4415 $\pm$ 437 (20) | 4827 $\pm$ 170 (69) |

## 5.1 ILLUSTRATIVE EXPERIMENT

**Extracted Substructures are Meaningful.** The illustrative experiment aims to optimize the following multi-parameter optimization (MPO) objective: maximize topological polar surface area (tPSA), molecular weight (MW) <350 Da, and number of rings $\geq$ 2. This specific MPO was chosen because it is plausible to predict what structural features would be *necessary* to optimize the objective: rings saturated with heteroatoms. Due to constraining the molecular weight, the model cannot just learn to generate large molecules that would, on average, possess a higher tPSA. Augmented Memory (Guo & Schwaller (2023) was used to optimize the MPO objective. The reward trajectory tends towards 1, indicating the model gradually learns to satisfy the target objective, as desired (Fig. 2) Next, we investigate the top $k$ and $N$ beam steps parameters for Beam Enumeration and show that while the majority of sub-sequences do not possess valid substructures, a meaningful signal can still be extracted (Appendix C). We hypothesize that the optimal parameters are using a low top $k$ as we are interested in the most probable sub-sequences and large $N$ beam steps, which would enable extracting larger (and potentially more meaningful) substructures. Fig. 2 shows the top four substructures from Beam Enumeration at varying epochs. As hypothesized, the substructures are informative when considering the MPO objective: the most frequent substructures gradually become rings saturated with heteroatoms, which possess a high tPSA.

**Self-conditioned Generation Improves Sample Efficiency.** Thus far, the results only show that Beam Enumeration can extract meaningful molecular substructures. To enable self-conditioned generation, a criterion is required to decide *when* to execute Beam Enumeration. We consider *when* extracted substructures would be meaningful and propose to execute Beam Enumeration when the reward improves for *Patience* number of successive epochs (to mitigate sampling stochasticity). We combine Beam Enumeration with Augmented Memory (Guo & Schwaller (2023) and perform an exhaustive hyperparameter grid search (with replicates) using Yield and Oracle Burden as the performance metrics (Appendix A). The results elucidate the behavior of Beam Enumeration with three key observations: firstly, *Structure* extraction is much more permissive compared to *Scaffold* and often leads to small functional groups, e.g., carbonyl, being the most frequent substructures which diminish the sample efficiency benefits (Appendix C). Secondly, enforcing larger substructures to be extracted (*Structure Minimum Size*) improves performance across all hyperparameter combinations. This reinforces that extracted substructures are meaningful as larger substructures heavily bias molecular generation during self-conditioning. If they were not meaningful, sample efficiency would not improve (and would likely be detrimental). Thirdly, *Structure* extraction while enforcing a higher *Structure Minimum Size* prevents small functional group extraction which significantly enhances performance.

We perform five experiments (N=100 replicates each) based on the optimal hyperparameters identified from the grid search: Augmented Memory (Guo & Schwaller (2023) (baseline) and Augmented Memory with Beam Enumeration (*Scaffold* and *Structure* with and without *Structure Minimum Size* = 15). Table 1 shows that Beam Enumeration drastically improves the Yield and Oracle Burden compared to the baseline at both the >0.7 and >0.8 reward thresholds, particularly when *Structure*

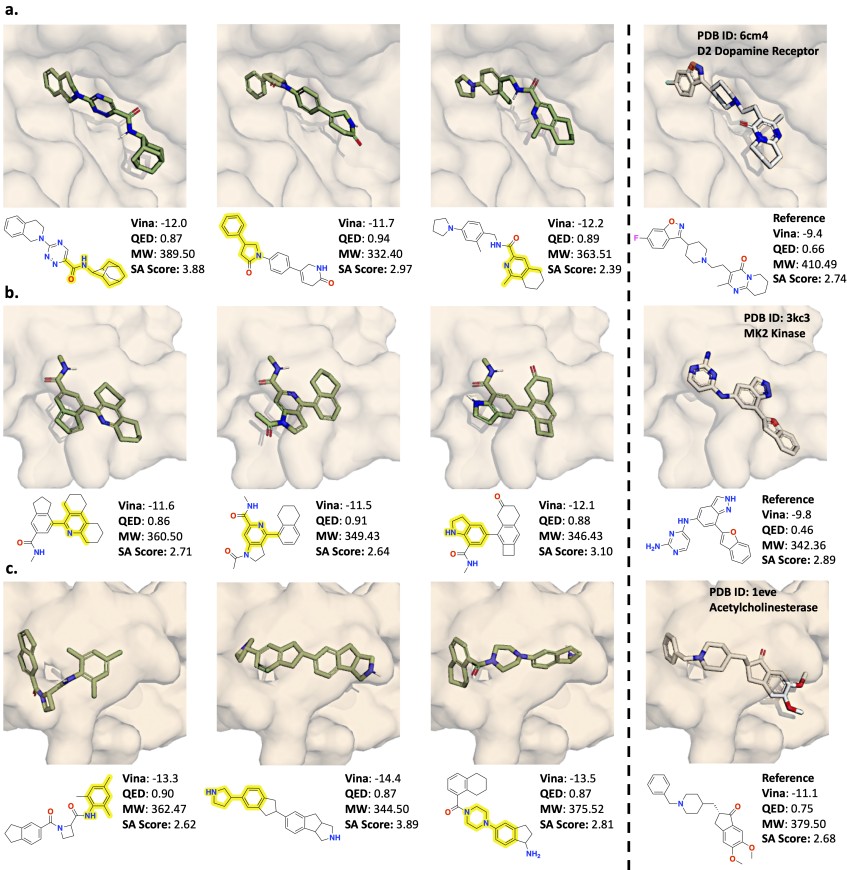

Figure 3: Three drug discovery case studies showing the top generated molecule (triplicate experiments) using Augmented Memory (Guo & Schwaller (2023)) with Beam Enumeration *Structure* Minimum Structure Size = 15 and the reference ligand. Extracted substructures from Beam Enumeration are highlighted. The multi-parameter optimization objective is: Minimize Vina score, maximize QED, and molecular weight <500 Da. The values, with the Synthetic Accessibility (SA) score (Ertl & Schuffenhauer (2009)) are annotated. **a.** Dopamine type 2 receptor (Wang et al. (2018). **b.** MK2 kinase (Argiriadi et al. (2010). **c.** Acetylcholinesterase (Kryger et al. (1999).

*Minimum Size* = 15 is enforced. We highlight that the improved sample efficiency is especially significant as baseline Augmented Memory could not find 100 molecules >0.8 reward in 69/100 replicates. The results are further compared using Welch's t-test, and all p-values are significant at the 95% confidence level (Appendix C).

## 5.2 Drug Discovery Case Studies

The positive results from the illustrative experiment suggest that Beam Enumeration can be applied to real-world drug discovery case studies to design inhibitors against DRD2 which is implicated in neurodegenerative diseases (Wang et al. (2018), MK2 kinase which is involved in pro-inflammatory responses (Argiriadi et al. (2010), and AChE which is a target of interest against Alzheimer's disease (Kryger et al. (1999). Following Guo et al. (Guo et al. (2021b); Guo & Schwaller (2023), we formulate the following MPO objective: minimize the AutoDock Vina (Trott & Olson (2010) docking score as a proxy for binding affinity, maximize the Quantitative Estimate of Druglikeness (QED) (Bickerton et al. (2012) score, and constrain MW <500 Da. The QED and MW objectives prevent the generative model from exploiting the weaknesses of docking algorithms to give inflated docking scores to large, lipophilic molecules, which can be promiscuous binders (Arnott & Planey (2012). Moreover, an oracle budget of 5,000 Vina calls was enforced which is almost half the budget of the original Augmented Memory (Guo & Schwaller (2023) work (9,600).

Since the observations made from the illustrative experiment hyperparameter grid search may not be generalizable to docking tasks, we perform an additional hyperparameter grid search (with replicates). The results (Appendix D) show that the optimal hyperparameters from the illustrative experiment are also the optimal hyperparameters across all three drug discovery case studies. We designate these the default hyperparameters and demonstrate the applicability of Beam Enumeration to both Augmented Memory (Guo & Schwaller (2023)) and REINVENT (Olivecrona et al. (2017); Blaschke et al. (2020) which is the second most (behind Augmented Memory) sample efficient model in the PMO (Gao et al. (2022) benchmark.

**Qualitative Inspection: Why Were These Molecules Generated?** We first show that Augmented Memory with Beam Enumeration generates molecules that satisfy the MPO objective (Fig. 3). We emphasize that results were not cherry-picked and the three generated examples shown are the top 1 (by reward) across triplicate experiments. All molecules possess better Vina scores and higher QED than the reference molecules, as desired. Fig. 3 shows the highlighted substructures extracted using *Structure* extraction with *Structure Minimum Size* = 15 with three key observations: firstly, "uncommon" molecular substructures may be extracted such as the bridged cycle against DRD2. The exact substructure extracted was an amide bond with a long carbon chain which implicitly enforces the bridged cycle, and the Vina pose shows that it fits in the binding cavity with no clashes, despite being a bulky group. Secondly, bicyclic or double-ring systems are often extracted (for all case studies), forming central scaffolds of the full molecule. Thirdly, scaffolds with branch points, i.e., a central ring with single carbon bond extensions, are often extracted (for all case studies). These substructures are particularly interesting as they heavily bias what can be generated in the remaining portion of the full molecule. An exemplary example of this is in the first generated molecule against MK2, where the branch points are effectively a part of two other ring systems (Fig. 3). Beam Enumeration can provide insights into the tolerability and suitability of certain substructures in the context of the full molecules (see Appendix D for more examples of extracted substructures). Finally, we posit that the extreme bias of *Structure* extraction is the reason why it can be more performant than *Scaffold*. Overall, the extracted substructures are meaningful and act both as a source of generative explainability and can self-direct the generative model into specific regions of chemical space with high reward.

**Quantitative Analysis: Sample Efficiency.** Next, we reinforce results from previous work showing that Augmented Memory (Guo & Schwaller (2023) is significantly more sample efficient than REINVENT (Olivecrona et al. (2017); Blaschke et al. (2020) (Table 2). Notably, the Yield of Augmented Memory is much greater than REINVENT at both the >0.7 and >0.8 reward thresholds, indicating that more high reward molecules are generated. Moreover, Augmented Memory has a lower Oracle Burden than REINVENT in all cases, except for Oracle Burden$_{>0.8}$ (1) for DRD2 and AChE where there is essentially no difference. The reason for this is because molecules with >0.8 reward were already generated at epoch 1, indicating the pre-trained model (trained on ChEMBL (Gaulton et al. (2012a)) is a good Prior for these case studies. By contrast, the MK2 case study is considerably more challenging as extremely few >0.8 reward molecules are generated under a 5,000 oracle calls budget. Augmented Memory significantly outperforms REINVENT as the latter could not find 10 molecules with reward >0.8 (Table 2).

Subsequently, we demonstrate that Beam Enumeration can be applied out-of-the-box on top of Augmented Memory and REINVENT. Firstly, the addition of Beam Enumeration improves the sample efficiency of both base algorithms, as evidenced by the Yield and Oracle Burden metrics in Table 2 with a small trade-off in diversity (Appendix D). However, the benefits are more pronounced in Augmented Memory as observed by the Yield$_{>0.8}$ improving by >4x in all cases (MK2 improves by 29x) and the Oracle Burden $_{>0.8}$ (10 and 100) over halved in most cases. Notably, for MK2 Oracle Burden $_{>0.8}$ (100), baseline Augmented Memory could not accomplish the task while Beam Enumeration is successful in almost under 2,000 oracle calls (Table 2). These findings are in agreement with the original Augmented Memory (Guo & Schwaller (2023) work in that the algorithm is much more data efficient and capitalizes on learning from high reward molecules via experience replay (Lin (1992). Beam Enumeration decreases the diversity of the generated set (as measured by Int-Div1Polykovskiy et al. (2020b), but finds considerably more *unique* scaffolds above the 0.8 reward threshold (up to 19x) (Appendix D). With many unique scaffolds built around (often) central substructures, the generated set could conceivably provide insights into structure-activity relationships. These results demonstrate that the combined algorithm achieves both exploration and exploitation. Overall, the results show that Beam Enumeration is task-agnostic and can be applied on top of ex-

isting algorithms to improve their sample efficiency. The combined algorithms generate more high reward molecules and faster, even in challenging (MK2) scenarios under a limited oracle budget. Furthermore, in reference to all the Oracle Burden metrics (Table 2), Augmented Memory with Beam Enumeration can identify a small set of *excellent* (high reward) candidate molecules in under 2,000 oracle calls and in some cases, even under 1,000 oracle calls.

Table 2: Drug discovery case studies: effect of Beam Enumeration on sample efficiency. All experiments were run in triplicate with an oracle budget of 5,000 calls and reported values are the mean and standard deviation. *Scaffold* and *Structure* indicate the type of substructure (*Structure Minimum Size* = 15) extracted. The Generative Yield and Oracle Burden are reported at varying reward thresholds. Parentheses after Oracle Burden denote the cut-off number of molecules. Best performance is bolded with the exception of Oracle Burden (1) (DRD2/AChE) which have essentially identical performance due to the pre-trained model. * and ** denote one and two replicates were unsuccessful, respectively.

| Metric | Target | Augmented Memory | | | REINVENT | | |
|---|---|---|---|---|---|---|---|
| | | Beam Structure 15 | Beam Scaffold 15 | Baseline | Beam Structure 15 | Beam Scaffold 15 | Baseline |
| Generative Yield$_{>0.7}$ ($\uparrow$) | DRD2 | **3474 ± 158** | 3412 ± 95 | 2513 ± 442 | 2392 ± 699 | 2686 ± 235 | 1879 ± 16 |
| | MK2 | **3127 ± 138** | 2584 ± 443 | 1446 ± 173 | 1822 ± 444 | 1553 ± 391 | 879 ± 10 |
| | AChE | 3824 ± 162 | **3902 ± 189** | 3288 ± 85 | 2511 ± 369 | 2684 ± 242 | 2437 ± 53 |
| Generative Yield$_{>0.8}$ ($\uparrow$) | DRD2 | **1780 ± 439** | 1607 ± 379 | 363 ± 195 | 417 ± 275 | 687 ± 366 | 102 ± 6 |
| | MK2 | **987 ± 211** | 523 ± 438 | 34 ± 13 | 179 ± 241 | 19 ± 7 | 2 ± 0 |
| | AChE | 2059 ± 327 | **2124 ± 326** | 556 ± 47 | 323 ± 58 | 310 ± 207 | 147 ± 11 |
| Oracle Burden$_{>0.8}$ (1) ($\downarrow$) | DRD2 | 126 ± 90 | 83 ± 29 | 187 ± 51 | 63 ± 0 | 127 ± 52 | 168 ± 149 |
| | MK2 | **736 ± 166** | 1221 ± 564 | 1360 ± 543 | 1110 ± 268 | 808 ± 524 | 1724 ± 802 |
| | AChE | 105 ± 29 | 63 ± 0 | 62 ± 0 | 62 ± 0 | 84 ± 29 | 83 ± 29 |
| Oracle Burden$_{>0.8}$ (10) ($\downarrow$) | DRD2 | 582 ± 83 | **571 ± 104** | 711 ± 120 | 1099 ± 930 | 604 ± 71 | 883 ± 105 |
| | MK2 | **1122 ± 154** | 2426 ± 1525 | 3833 ± 394 | 1778 ± 0** | 3891 ± 631 | Failed |
| | AChE | 462 ± 25 | 418 ± 27 | **380 ± 0** | 441 ± 132 | 421 ± 120 | 481 ± 108 |
| Oracle Burden$_{>0.8}$ (100) ($\downarrow$) | DRD2 | **1120 ± 194** | 1056 ± 146 | 2558 ± 30* | 1928 ± 117* | 2109 ± 1090 | 4595 ± 0** |
| | MK2 | **2189 ± 181** | 2676 ± 403* | Failed | 3208 ± 0** | Failed | Failed |
| | AChE | 1110 ± 265 | **884 ± 162** | 2021 ± 89 | 3073 ± 427 | 3596 ± 678 | 3931 ± 286 |

## 6 CONCLUSION

In this work, we propose Beam Enumeration to exhaustively enumerate sub-sequences from language-based molecular generative models and show that substructures can be extracted, providing a source of generative explainability. Next, we show that the extracted molecular substructures can be used to self-condition the generative model to only perform oracle evaluation for molecules possessing these substructures (discarding the rest). We show that Beam Enumeration can be coupled with existing RL-based algorithms including Augmented Memory (Guo & Schwaller (2023) and REINVENT (Olivecrona et al. (2017); Blaschke et al. (2020) to improve their sample efficiency. Moreover, enforcing the extraction of larger substructures improves performance across all hyperparameter combinations. We believe this is a particularly interesting observation as it demonstrates the model's remarkable robustness and tolerability to extreme bias. Subsequently, in three drug discovery case studies to design molecules that dock well, the addition of Beam Enumeration to Augmented Memory and REINVENT substantially improves sample efficiency as assessed by the Yield (number of unique molecules generated above a reward threshold) and Oracle Burden (number of oracle calls required for the model to generate $N$ unique molecules above a reward threshold) with a small trade-off in diversity (which is expected). The extracted substructures themselves provide valuable structural insights, often enforcing the generation of specific cyclic systems and scaffolds with branch points which impose an overall molecular geometry that complements the protein binding cavity. Beam Enumeration shows that improvements to explainability and sample efficiency for molecular design can be made synergistic. The improvements in the latter will enable more expensive high-fidelity oracles to be explicitly optimized. We note, however, that sparse reward environments (Korshunova et al. (2022) remain a difficult optimization task. Finally, Beam Enumeration is a task-agnostic method and can be combined with recent work integrating active learning with molecular generation to further improve sample efficiency (Dodds et al. (2024); Kyro et al. (2023). If the benefits can be synergistic, we may approach sufficient sample efficiency to directly optimize expensive state-of-the-art (in predictive accuracy) physics-based oracles such as MD simulations (Wang et al. (2015); Moore et al. (2023). Excitingly, this would in turn enhance explainability as high-fidelity oracles are inherently more informative.

## 7 REPRODUCIBILITY STATEMENT

The code is provided in the GitHub link in the Abstract and also provided here: `https://github.com/schwallergroup/augmented_memory`. In the repository, there are prepared configuration files that can be directly run to reproduce all experiments in this work.

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

# A APPENDIX

The Appendix contains further experiments, ablation studies, experiment hyperparameters, and algorithmic details.

# A BEAM ENUMERATION

This section contains full details on Beam Enumeration including hyperparameters, design decisions, and pseudo-code.

## A.1 ALGORITHM OVERVIEW

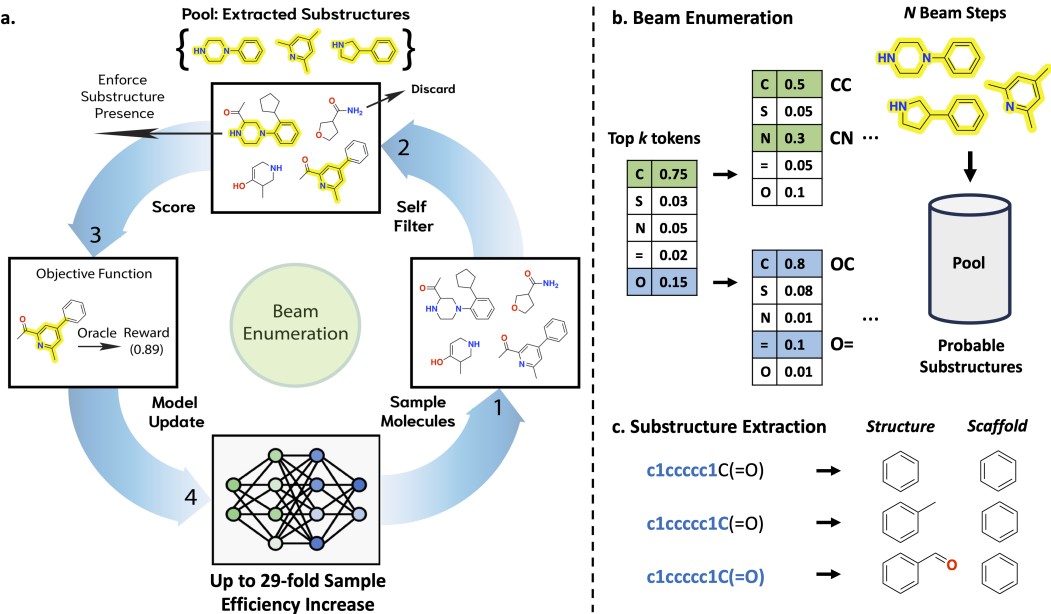

Figure A4: Beam Enumeration overview. **a.** The proposed method proceeds via 4 steps: **1.** generate batch of molecules. **2.** filter molecules based on pool to enforce substructure presence, discarding the rest. **3.** compute reward **4.** update the model. After updating the model, if the reward has improved for consecutive epochs, execute Beam Enumeration. **b.** Beam Enumeration sequentially enumerates the top $k$ tokens by probability for $N$ beam steps, resulting in an exhaustive set of token sub-sequences. **c.** All valid substructures (either by the *Structure* or *Scaffold* criterion) are extracted from the sub-sequences. The most frequent substructures are used for self-conditioned generation. This overview figure is the same as in the main text.

Beam Enumeration (Fig. A4) is an algorithm that extracts molecular substructures from a generative model's weights for self-conditioned generation. The problem set-up is any molecular design task to optimize for a target property profile, e.g., high predicted solubility and binding affinity. When molecular generative models are coupled with an optimization algorithm, it should be increasingly likely to generate desirable molecules, i.e., molecules that possess the target property profile.

Beam Enumeration is proposed based on two facts:

1. On a successful optimization trajectory, the model's weights must change such that desirable molecules are more likely to be generated, on average.
2. The act of generating molecules in an autoregressive manner involves sequentially sampling from conditional probability distributions.

In this work, Beam Enumeration is applied to a language-based autoregressive generative model operating on the simplified molecular-input line-entry system (SMILES) (Weininger (1988) rep-

resentation. The optimization algorithm is Augmented Memory (Guo & Schwaller (2023) which builds on REINVENT (Olivecrona et al. (2017); Blaschke et al. (2020) and casts the optimization process as an on-policy reinforcement learning (RL) problem. Following RL terminology, sampling from the generative model involves sampling *trajectories*, which in this case, are SMILES, and the desirability of the corresponding molecule is given by the *reward*.

The underlying hypothesis of Beam Enumeration is that during the RL optimization process, *partial trajectories* provide a source of signal that can be exploited. Usually, full trajectories are sampled which map to a complete SMILES sequence that can be translated to a molecule. Our assumption is that partial trajectories (partial SMILES sequence) can be mapped to molecular substructures (a part of the full molecule). This statement is not guaranteed as SMILES and molecules are discrete and small perturbations often leads to invalid SMILES. We prove this assumption in Section C by showing that although the vast number of partial trajectories do not map to valid SMILES, the raw number is sufficient to extract a meaningful signal. Correspondingly, Beam Enumeration leverages partial trajectories on the assumption that molecular substructures are *on track* to becoming full molecules that *would* receive high reward.

## A.2 ENUMERATING PARTIAL TRAJECTORIES

In order to extract molecular substructures, a set of partial trajectories must be sampled from the generative model. Recalling the fact that on a successful optimization trajectory, it must become increasingly likely to generate desirable molecules, partial trajectories are sampled by enumerating the top $k$ tokens, based on the conditional probability. Therefore, the process of enumerating partial trajectories involves sequentially extending each token sequence by their next top $k$ probable tokens, resulting in the total number of partial trajectories as $2^N$ where $N$ is the number of beam steps, i.e., how many tokens in the partial trajectory. We note that taking the top $k$ most probable tokens does not guarantee that the partial trajectories are indeed the most probable, as paying a probability penalty early can lead to higher probabilities later. However, our assumption is that on average, this leads to a set of partial trajectories that are at the very least, amongst the most probable. Moreover, there is a practical limit to how many partial trajectories are sampled due to exponential growth which makes scaling quickly computationally prohibitive. In the later section, we discuss this thoroughly. Finally, from here, partial trajectories will be referred to as token sub-sequences.

## A.3 EXTRACTING MOLECULAR SUBSTRUCTURES

Given a set of token sub-sequences, the goal is to extract out the most frequent molecular substructures. This is done by taking each sequence, considering every (sub)-sub-sequence, and counting the number of valid substructures (Fig. A). For example, given the sub-sequence "ABCD", the set of (sub)-sub-sequences are: "A", "AB", "ABC", and "ABCD". In practice, we only consider (sub)-sub-sequences with at least three characters ("ABC" and "ABCD") since each character loosely maps to one atom and three is approximately the minimum for meaningful functional groups, e.g., "C=O", a carbonyl. The set of most frequent substructures is assumed to be on track to receive a high reward.

## A.4 DEFINING MOLECULAR SUBSTRUCTURES: *Scaffold* VS. *Structure*

As shown in Fig. A, molecular substructures can be defined on the *Scaffold* or *Structure* level. The former extracts the Bemis-Murcko (Bemis & Murcko (1996) scaffold while the latter extracts *any* valid structure. The *any* valid structure is an important distinction as our experiments find that extracting by *Structure* leads to the most frequent molecular substructures being small functional groups that do not have corresponding scaffolds. By contrast, extracting the scaffold always leads to ring structures. Moreover, extracting specifically the Bemis-Murcko scaffold is important as heavy atoms, e.g., nitrogen, are important for biological activity. Consequently, extracted substructures are also enforced to contain at least one heavy atom as we find that benzene, perhaps unsurprisingly, is commonly the most frequent substructure. See Section B for more details on the differing behavior of 'Scaffold' vs. 'Structure'.

### A.5 SELF-CONDITIONED GENERATION

Self-conditioned generation is achieved by filtering sampled batches of molecules from the generative model to only keep the ones that possess at least one of the most frequent substructures. The effect is that the generative process is self-biased to focus on a narrower chemical space which we show can drastically improve sample efficiency at the expense of some diversity, which is acceptable when expensive high-fidelity oracles are used: we want to identify a small set of *excellent* candidate molecules under minimal oracle calls.

### A.6 PROBABILISTIC EXPLAINABILITY

The set of most frequent molecular substructures should be meaningful as otherwise, the model's weights would not have been updated such that these substructures have become increasingly likely to be generated. We verify this statement in the illustrative experiment in the main text and in Section C. In the drug discovery case studies (Appendix D), the extracted substructures are more subtle in why they satisfy the target objective but certainly must possess meaning, however subtle, as otherwise, they would not receive a high reward. In the main text, we show that extracted substructures form core scaffolds and structural motifs in the generated molecules that complement the protein binding cavity. Finally, we emphasize that the *correctness* and *usefulness* of this explainability deeply depends on the oracle(s) being optimized for. The extracted substructures do not explain why the generated molecules satisfy the target objective. Rather, they explain why the generated molecules satisfy the oracle. The assumption in a generative design task is that optimizing the oracle is a good proxy for the target objective, e.g., generating molecules that dock well increases the likelihood of the molecules being true binders. This observation directly provides additional commentary on why sample efficiency is so important: the ability to directly optimize expensive high-fidelity oracles would inherently enhance the *correctness* of the extracted substructures.

## B BEAM ENUMERATION: FINDINGS FROM HYPERPARAMETER SCREENING

In this section, we introduce all seven hyperparameters of Beam Enumeration and then present results on an exhaustive hyperparameter search which elucidates the behavior and interactions of all the hyperparameters. In the end, we present our analyses and provide hyperparameter recommendations for Beam Enumeration which can serve as default values to promote out-of-the-box application.

### B.1 BEAM ENUMERATION HYPERPARAMETERS

**Beam k.** This hyperparameter denotes how many tokens to enumerate at each step. Given that our hypothesis is that the most probable sub-sequences yield meaningful substructures, we fix Beam $k$ to 2. A larger value would also decrease the number of Beam Steps possible as the total number of sub-sequences is $k^N$ and the exponential growth quickly leads to computational infeasibility.

**Beam Steps N.** This hyperparameter denotes how many token enumeration steps to execute and is the final token length of the enumerate sub-sequences. This parameter leads to exponential growth in the number of sequences which can quickly become computationally prohibitive. An important implication of this hyperparameter is that larger Beam Steps means that larger substructures *can* be extracted. In our experiments, we find that enforcing size in the extracted substructures can drastically improve sample efficiency with decreased diversity as the trade-off. We thoroughly discuss this in a later sub-section. Finally, in our experiments, the upper-limit investigated is 18 Beam Steps.

**Substructure Type.** This hyperparameter has two possible values: *Scaffold* or *Structure*. *Scaffold* extracts Bemis-Murcko (Bemis & Murcko (1996) scaffolds while *Structure* extracts *any* valid substructure.

**Structure Structure Minimum Size.** This hyperparameter enforces the partial SMILES to contain at least a certain number of characters. In effect, this enforces extracted molecular substructures to be larger than a Structure Minimum Size. From the illustrative experiment in the main text and Section C, *Structure* extraction often leads to small functional groups being the most frequent in the sub-sequences. By enforcing a minimum structure size, *Structure* extraction leads to partial

structures which may carry more meaning. We find that this hyperparameter greatly impacts sample efficiency and we present all our findings in a later sub-section.

**Pool Size.** This hyperparameter controls how many molecular substructures to keep track of. These *pooled* substructures are what is used to perform self-conditioning. The hypothesis is that the most frequent ones carry the most meaning and thus, a very large pool size may not be desired.

**Patience.** This hyperparameter controls how many successive reward improvements are required before Beam Enumeration executes and molecular substructures are extracted. Recalling the first fact in which Beam Enumeration was proposed on: On a successful optimization trajectory, the model's weights must change such that desirable molecules are more likely to be generated, on average. Patience is effectively an answer to "when would extracted substructures be meaningful?" Too low a patience and stochasticity can lead to negative effects while too high a patience diminishes the benefits of Beam Enumeration on sample efficiency.

**Token Sampling Method.** This hyperparameter has two possible values: "topk" or "sample" and denotes *how* tokens sub-sequences are enumerated. "topk" takes the top k most probable tokens at each Beam Step while "sample" samples from the distribution just like during batch generation. Our results show interesting observations surrounding this hyperparameter as "sample" can work just as well and *sometimes* even better than taking the "topk". These results were unexpected as the underlying hypothesis is that the most probable sub-sequences lead to the most useful substructures being extracted. However, our findings are not in contradiction as sampling the conditional probability distributions would still lead to sampling the top k tokens, on average. Moreover, after extensive experiments, we find that "sample" leads to more variance in performance across replicates which is in agreement with the assumption that sampling the distributions can lead to more improbable structures. We thoroughly discuss our findings in a later sub-section where we provide hyperparameter recommendations and analyses to the effects of tuning each hyperparameter.

### B.2 HYPERPARAMETERS: GRID SEARCH

We performed two exhaustive hyperparameter grid searches on the illustrative experiment which has the following multi-parameter optimization (MPO) objective: maximize topological polar surface area (tPSA), molecular weight <350 Da, number of rings $\geq 2$ with an oracle budget of 5,000. The first grid search investigated the following hyperparameter combinations:

- Beam K = 2
- Beam Steps = [15, 16, 17, 18]
- Substructure Type = [*Scaffold*, *Structure*]
- Pool Size = [3, 4, 5]
- Patience = [3, 4, 5]
- Token Sampling Method = ['topk', 'sample']

**All hyperparameter combinations (144) were tried and run for 10 replicates each for statistical reproducibility, total of 1,440 experiments.** Next, an additional grid search was performed with the following hyperparameter combinations:

- Beam K = 2
- Beam Steps = [17, 18]
- Substructure Type = [*Scaffold*, *Structure*]
- Structure Structure Minimum Size = [10, 15]
- Pool Size = [4, 5]
- Patience = [4, 5]
- Token Sampling Method = ['topk', 'sample']

We take the general trends from the first grid search and narrow down the most optimal hyperparameters to further investigate Substructure Type and structure Structure Minimum Size. **As from**

**before, all hyperparameter combinations (64) were tried and run for 10 replicates each for statistical reproducibility, total of 640 experiments.**

The following heatmaps performance by the Generative Yield and Oracle Burden (10) metrics at the >0.8 reward threshold and under a 5,000 oracle budget. The Generative Yield measures how many unique molecules above 0.8 reward were generated. The Oracle Burden (10) measures how few oracle calls were required to generate 10 molecules above 0.8 reward.

### B.3    ANALYSIS OF GRID SEARCH RESULTS

In this section, we summarize our analysis on the grid search experiments. Unless stated, each bullet point means the observation was observed for both Generative Yield and Oracle Burden (10). For example the point: *Scaffold* >*Structure* means *Scaffold* is generally more performant than *Structure* across all hyperparameters on both the Generative Yield and Oracle Burden (10).

- For *Scaffold*, higher Pool, higher Patience, and higher Beam Steps improves performance
- For *Structure*, lower pool and lower patience improves performance
- *Scaffold* >*Structure*
- *Scaffold* and *Structure* become more performant with increasing Structure Minimum Size
- *Scaffold* and *Structure* with Structure Minimum Size: "sample" sampling *can* be better than "topk" sampling but with more variance

Based on the above analysis, we propose the optimal hyperparameters for the illustrative experiment as:

- *Scaffold*
- "topk" sampling ("sample" sampling can be more performant but exhibits higher variance)
- Patience = 5
- Pool Size = 4
- Beam Steps = 18

Finally, we provide more commentary on interesting observations from the grid search results. *Structure* without Structure Minimum Size enforcing often leads to small functional groups being the most frequent molecular substructures extracted with Beam Enumeration. Enforcing Structure Minimum Size puts it almost on par with *Scaffold*, suggesting (perhaps not surprisingly) that larger substructures can carry more meaningful information. Moreover, when using "sample" sampling, the generative model undergoes more "filter rounds". Specifically, at each epoch, the sampled batch is filtered to contain the extracted substructures. When using "sample" sampling, the model is more prone to some epochs containing no molecules with the substructures. In practice, this is inconsequential as sampling is computationally inexpensive and a next batch of molecules can easily be sampled. However, specifically in the *Structure* with "sample" sampling and Structure Minimum Size = 15 experiment, "filter round" can be quite extensive, taking up to 100,000 epochs (maximum observed) for an oracle budget of 5,000 (adding about an hour to the wall time which is minor when the oracle is expensive). This means that many epochs contained molecules without the extracted substructures. There are two observations here: firstly, "sample" sampling can lead to more improbable substructures which are hence less likely to be sampled and secondly, *Structure* with Structure Minimum Size enforcement leads to extreme biasing (which improves sample efficiency). We believe the remarkable tolerability of the generative model sampling to such bias is an interesting observation. By contrast, *Scaffold* with Structure Minimum Size enforcement is not as prone to "filter rounds" because *Scaffold* "truncates" the substructure to its central shape (scaffold). For example, toluene (benzene with a methyl group) has a Bemis-Murcko (Bemis & Murcko (1996) scaffold of just benzene. The consequence is that *Structure* leads to more extreme biasing (it is more likely for a molecule to contain benzene than specifically toluene) which is in agreement with the general observation that the diversity of the generated set decreases when using *Structure*. Overall, both *Scaffold* and *Structure* with Structure Minimum Size enforcing exhibits the best performance and "sample" sampling *can* be more performant than "topk" sampling but exhibits notably higher variance.

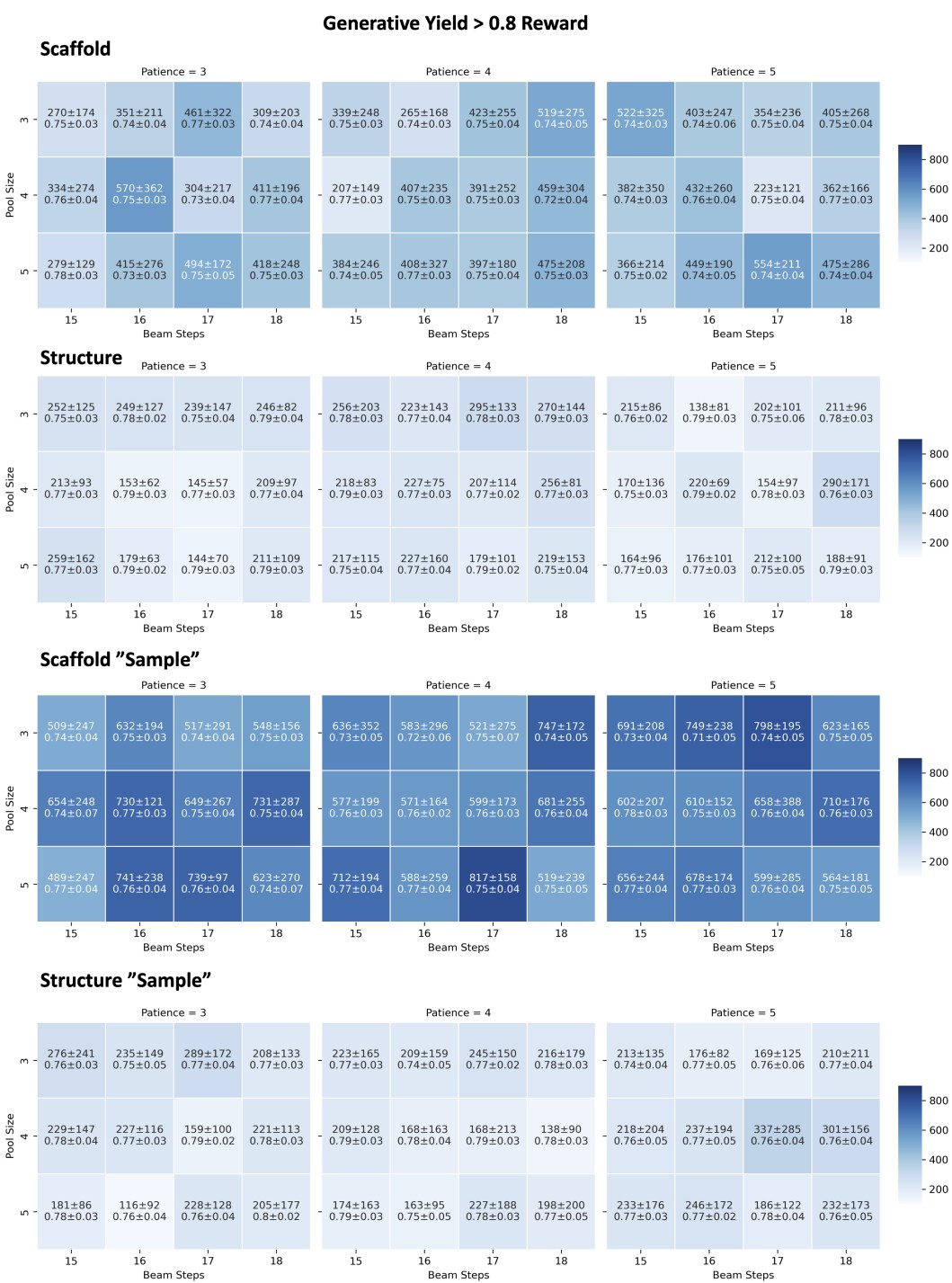

Figure B5: illustrative experiment Generative Yield >0.8. The IntDiv1 (Polykovskiy et al. (2020b) is annotated.

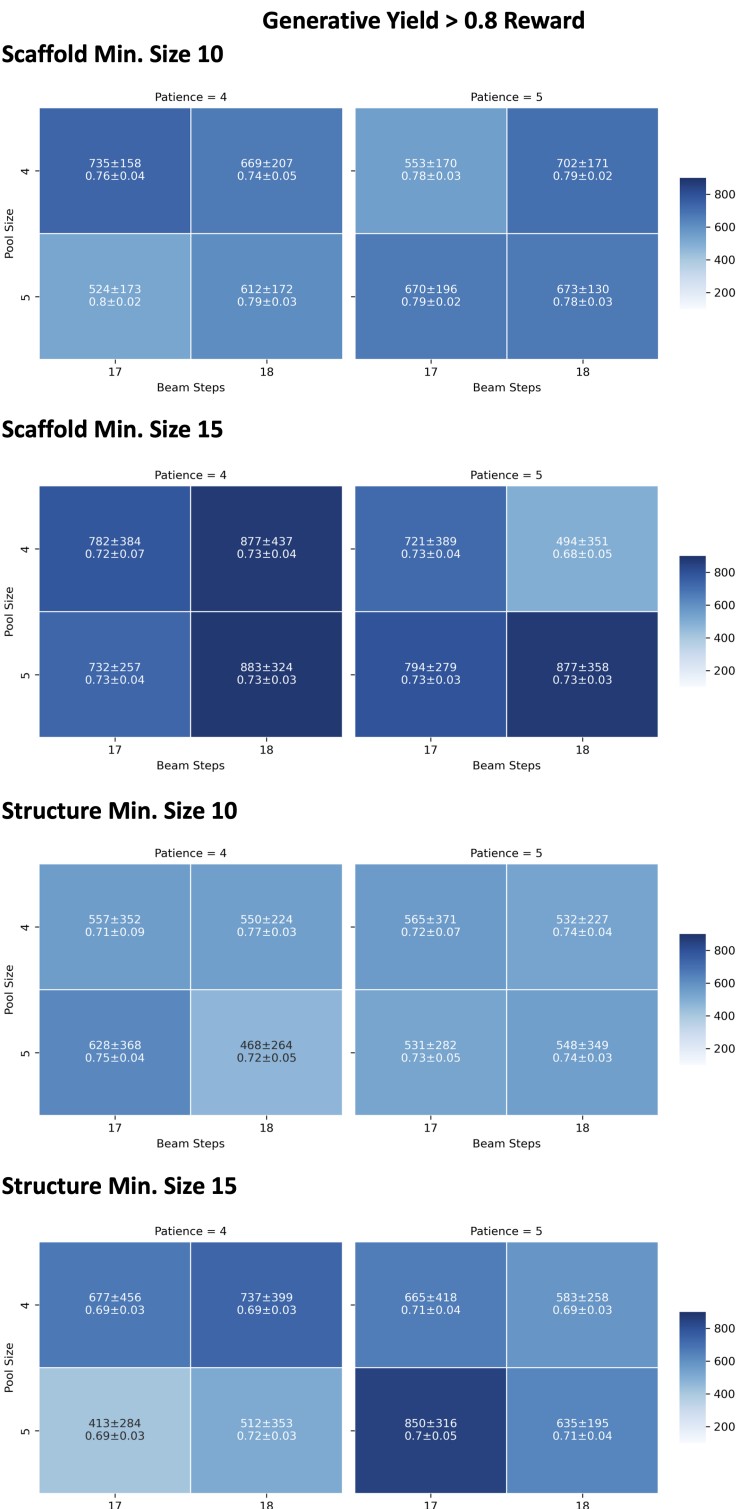

Figure B6: illustrative experiment Generative Yield >0.8 with Structure Minimum Size. The Int-Div1 (Polykovskiy et al. (2020b) is annotated.

**Generative Yield > 0.8 Reward**

**Scaffold "Sample" Min. Size 10**

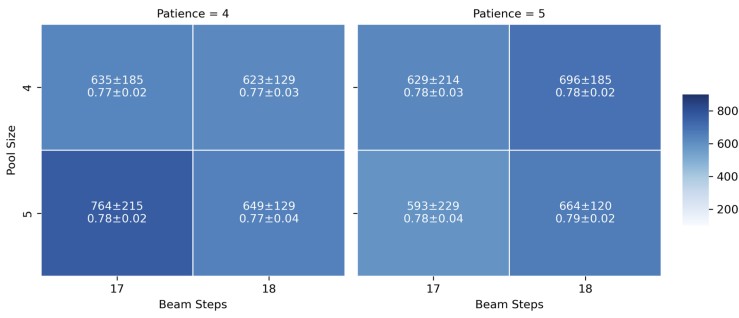

**Scaffold "Sample" Min. Size 15**

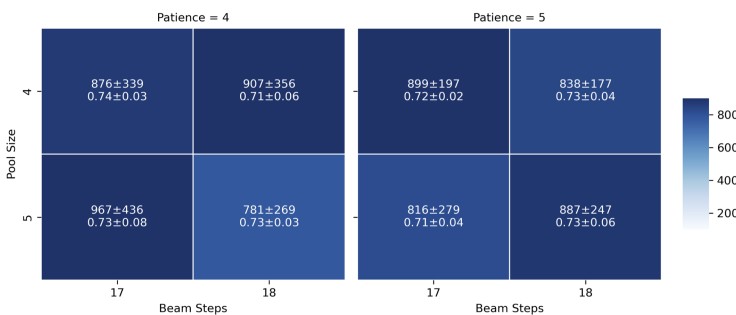

**Structure "Sample" Min. Size 10**

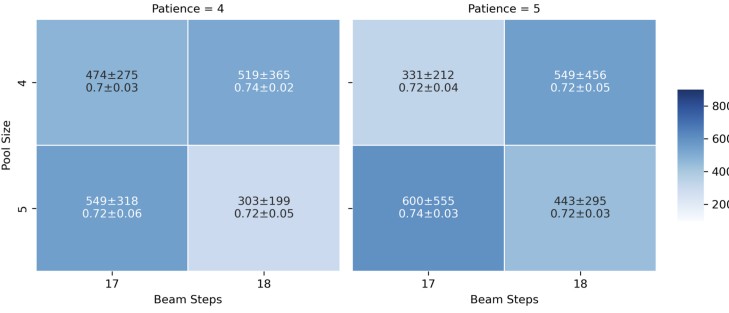

**Structure "Sample" Min. Size 15**

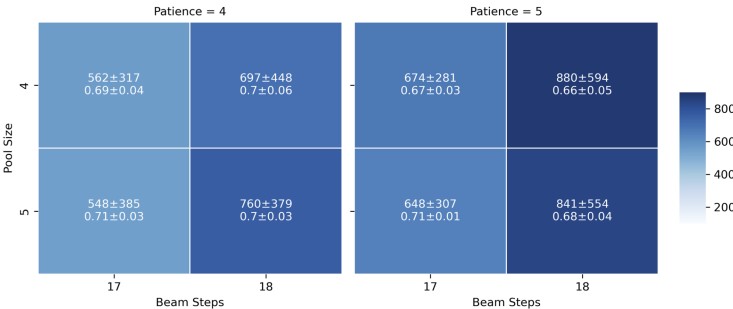

Figure B7: illustrative experiment Generative Yield >0.8 with Structure Minimum Size and "Sample" token sampling. The IntDiv1 (Polykovskiy et al. (2020b) is annotated.

**Oracle Burden (10) > 0.8 Reward**

## Scaffold

### Patience = 3

| Pool Size \ Beam Steps | 15 | 16 | 17 | 18 |
|---|---|---|---|---|
| 3 | 3200±548 | 3274±466 | 3162±617 | 3375±491 |
| 4 | 3265±462 | 2978±505 | 3361±351 | 3075±351 |
| 5 | 3374±508 | 3135±554 | 2938±330 | 3134±441 |

### Patience = 4

| Pool Size \ Beam Steps | 15 | 16 | 17 | 18 |
|---|---|---|---|---|
| 3 | 3378±500 | 3424±492 | 3040±412 | 2946±604 |
| 4 | 3502±538 | 3179±392 | 3249±549 | 3082±737 |
| 5 | 3089±523 | 3139±675 | 3051±369 | 2908±393 |

### Patience = 5

| Pool Size \ Beam Steps | 15 | 16 | 17 | 18 |
|---|---|---|---|---|
| 3 | 2984±567 | 3018±454 | 3176±447 | 3147±298 |
| 4 | 3227±583 | 3131±412 | 3569±315 | 3171±350 |
| 5 | 3230±521 | 3134±308 | 2859±303 | 3116±614 |

## Structure

### Patience = 3

| Pool Size \ Beam Steps | 15 | 16 | 17 | 18 |
|---|---|---|---|---|
| 3 | 3072±535 | 2807±577 | 3034±605 | 2880±477 |
| 4 | 3024±562 | 3238±519 | 3498±574 | 3114±470 |
| 5 | 2978±381 | 3134±429 | 3389±496 | 3102±410 |

### Patience = 4

| Pool Size \ Beam Steps | 15 | 16 | 17 | 18 |
|---|---|---|---|---|
| 3 | 3040±612 | 3103±494 | 2851±648 | 2910±449 |
| 4 | 3075±512 | 2969±368 | 3091±521 | 2868±395 |
| 5 | 3129±449 | 3014±579 | 3225±564 | 3118±514 |

### Patience = 5

| Pool Size \ Beam Steps | 15 | 16 | 17 | 18 |
|---|---|---|---|---|
| 3 | 3112±289 | 3518±544 | 3123±458 | 2931±293 |
| 4 | 3411±684 | 2917±524 | 3517±618 | 3057±610 |
| 5 | 3467±569 | 3285±549 | 3075±510 | 3292±540 |

## Scaffold "Sample"

### Patience = 3

| Pool Size \ Beam Steps | 15 | 16 | 17 | 18 |
|---|---|---|---|---|
| 3 | 2555±597 | 2504±345 | 2878±800 | 2351±355 |
| 4 | 2491±482 | 2224±331 | 2395±473 | 2409±453 |
| 5 | 2548±670 | 2147±366 | 2428±185 | 2516±537 |

### Patience = 4

| Pool Size \ Beam Steps | 15 | 16 | 17 | 18 |
|---|---|---|---|---|
| 3 | 2626±714 | 2817±716 | 2559±586 | 2254±414 |
| 4 | 2743±448 | 2626±364 | 2449±564 | 2480±484 |
| 5 | 2250±305 | 2556±434 | 2316±422 | 2684±688 |

### Patience = 5

| Pool Size \ Beam Steps | 15 | 16 | 17 | 18 |
|---|---|---|---|---|
| 3 | 2297±427 | 2143±403 | 2122±378 | 2323±298 |
| 4 | 2462±425 | 2475±435 | 2388±574 | 2532±390 |
| 5 | 2413±494 | 2233±329 | 2499±453 | 2419±561 |

## Structure "Sample"

### Patience = 3

| Pool Size \ Beam Steps | 15 | 16 | 17 | 18 |
|---|---|---|---|---|
| 3 | 3136±697 | 3450±476 | 3347±472 | 3427±510 |
| 4 | 3331±559 | 3375±538 | 3627±527 | 3298±389 |
| 5 | 3418±397 | 3874±452 | 3444±568 | 3574±578 |

### Patience = 4

| Pool Size \ Beam Steps | 15 | 16 | 17 | 18 |
|---|---|---|---|---|
| 3 | 3297±622 | 3356±854 | 3401±404 | 3448±503 |
| 4 | 3553±440 | 3439±512 | 3745±682 | 3730±504 |
| 5 | 3541±502 | 3459±414 | 3624±610 | 3513±600 |

### Patience = 5

| Pool Size \ Beam Steps | 15 | 16 | 17 | 18 |
|---|---|---|---|---|
| 3 | 3370±586 | 3491±504 | 3740±418 | 3567±647 |
| 4 | 3451±469 | 3419±732 | 3172±544 | 3249±495 |
| 5 | 3469±441 | 3486±480 | 3465±576 | 3528±572 |

Figure B8: illustrative experiment Oracle Burden (10) >0.8

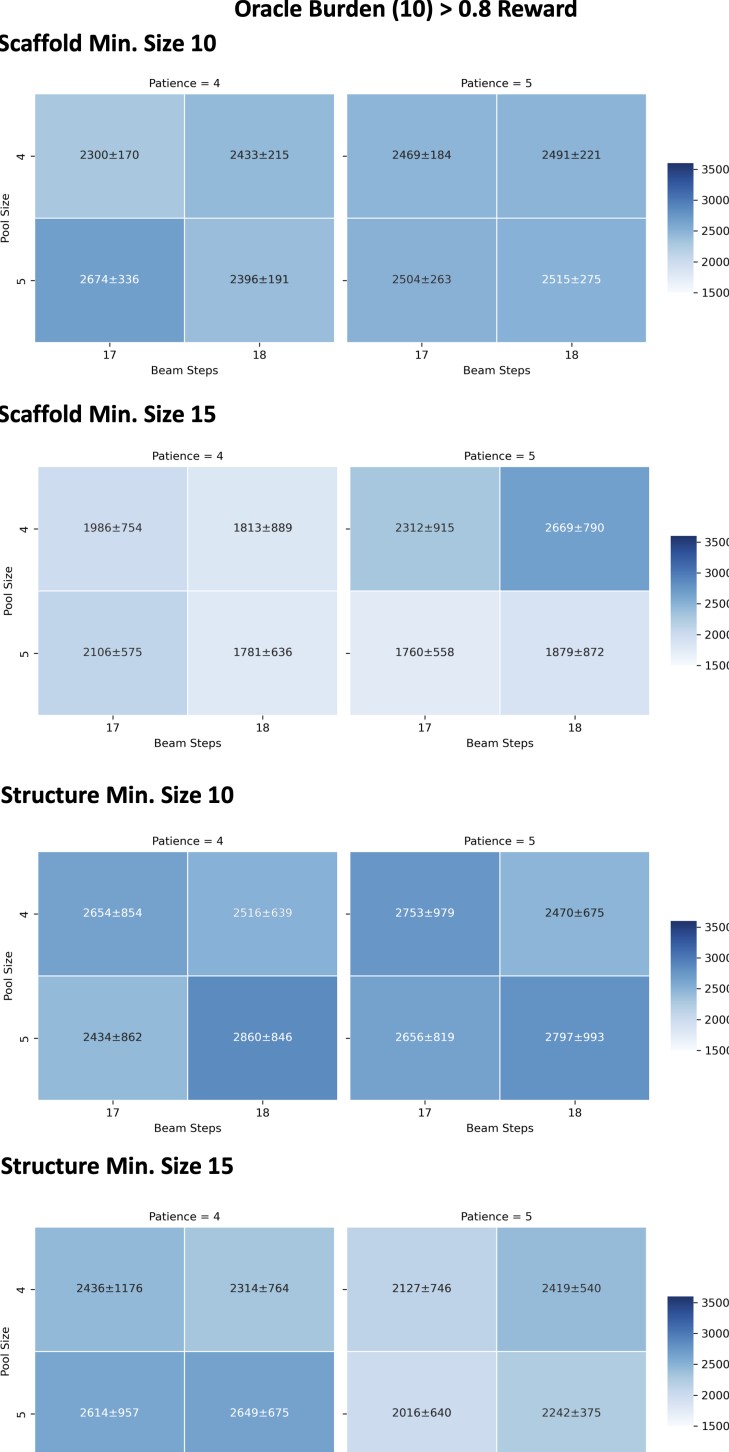

Figure B9: illustrative experiment Oracle Burden (10) >0.8 with Structure Minimum Size

**Oracle Burden (10) > 0.8 Reward**

**Scaffold "Sample" Min. Size 10**

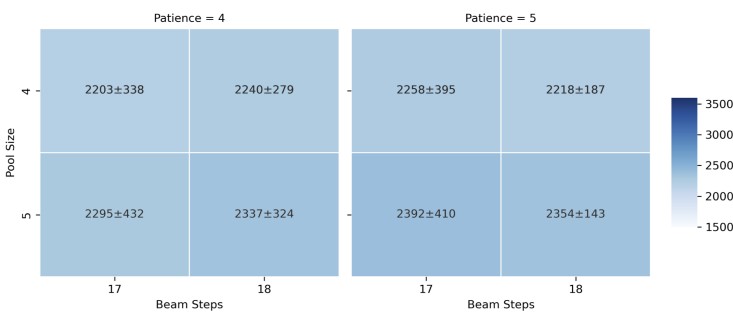

**Scaffold "Sample" Min. Size 15**

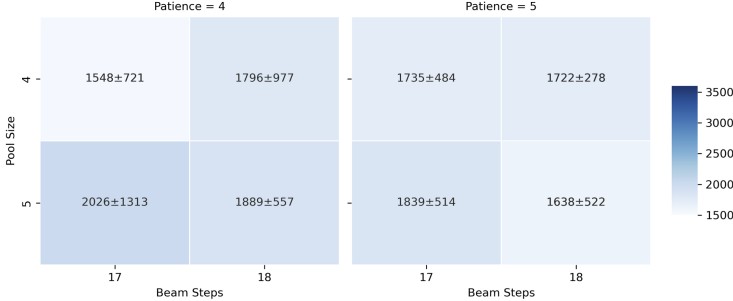

**Structure "Sample" Min. Size 10**

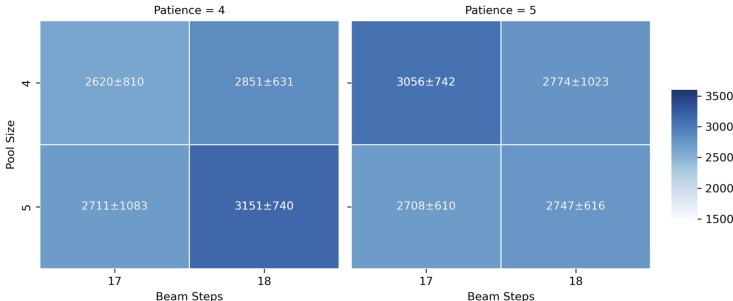

**Structure "Sample" Min. Size 15**

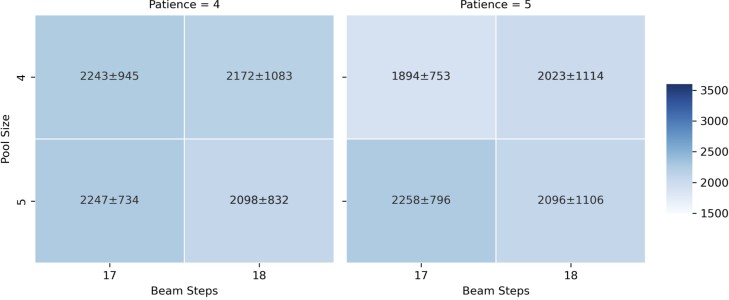

Figure B10: illustrative experiment Oracle Burden (10) >0.8 with Structure Minimum Size and "Sample" token sampling

The set of optimal hyperparameters found here were used in drug discovery case studies. In order to be rigorous with our investigation, we only fix the following hyperparameters:

- Patience = 5 (lower variance)
- Pool Size = 4 (lower variance, higher Yield, lower Oracle Burden)
- Beam Steps = 18 (lower variance, higher Yield, lower Oracle Burden)

with these hyperparameters, we do a small grid search (on the drug discovery case studies) by changing the Structure Type, Token Sampling Method, and Structure Minimum Size hyperparameters as the optimal hyperparameters in the illustrative experiment are not necessarily the optimal ones in the drug discovery experiments. **The purpose of this is not to necessarily report the best performance on the drug discovery case studies but to gain insights into the optimal general parameters such that Beam Enumeration can be used out-of-the-box. In real-world expensive oracle settings, tuning hyperparameters is infeasible.**

All results from the drug discovery case studies are shown in Section D.

### B.4 BEAM ENUMERATION: RECOMMENDED DEFAULT HYPERPARAMETERS

Taking into consideration all grid search experiments for the illustrative experiment and Drug Discovery case studies, the following optimal hyperparameters are recommended: Patience = 5, Pool Size = 4, Beam Steps = 18, *Structure*, Structure Minimum Size = 15, "topk" sampling.

Notable differences between the final recommended hyperparameters compared to those found from the illustrative experiment is that *Structure* and "topk" sampling are more performant than *Scaffold* and "sample" sampling. In the illustrative experiment, "sample" sampling was sometimes more performant than "topk" sampling. We rationalize these observations as follows: in MPO objectives that include physics-based oracles, structure specificity becomes increasingly important, e.g., specific chemical motifs dock well because they form interactions with the protein. Therefore, "topk" sampling is more robust as there is less variance in the extract substructures compared to "sample" sampling. We empirically observe the increased variance when using "sample" sampling measured by the standard deviation between replicate experiments (Appendix D). In the illustrative experiment where the oracle was more permissive, i.e., *any* rings saturated with heteroatoms would satisfy the MPO objective, small deviations in the extracted structure do not have as prominent an effect as physics-based oracles which require specificity. Another observation is that *Structure* sampling often extracts scaffolds with "branch points" which enforces extreme bias that can lead to more focused chemical space exploration. We discuss this in detail in Section D and believe the insights are generally interesting in the context of molecular optimization landscape.

**Finally, we end this section by stating that we cannot try every single hyperparameter combination and the recommended values are from our grid search results which we make an effort to be robust, given that we perform 10 replicates of each experiment. We find that the optimal hyperparameters in the drug discovery case studies are generally the same as in the illustrative experiment.**

## B.5 PSEUDO-CODE

This section contains the pseudo-code for Beam Enumeration and Augmented Memory with Beam Enumeration to show the integration.

### B.5.1 BEAM ENUMERATION

This subsection contains the Beam Enumeration pseudo-code. The $\oplus$ operator denotes every element on the left is being extended by every element on the right.

---

**Algorithm 1:** Beam Enumeration

---

**Input:** Generative Agent $\pi_{\theta_{\text{Agent}}}$, Top $k$, $N$ Beam Steps
**Output:** Enumerated Token Sub-sequences $S$
**Initialization:**
Hidden State = None;
Sub-sequences = [Top $k$ ¡START¿ Tokens];
Input Vector = top $k$ number of start tokens;
**for** $i = 1$ **to** $N$ **do**
    Logits, New Hidden State $\leftarrow \pi_{\theta_{\text{Agent}}}$(Input Vector, Hidden State);
    Tokens$_K \leftarrow$ top $k$ tokens from Softmax(Logits);
    **if** $i = 1$ **then**
        Sub-sequences $\leftarrow$ Tokens$_K$;
        Input Vector $\leftarrow$ Tokens$_K$;
        Hidden State = New Hidden State;
    **else**
        Create empty list $temp$;
        **for** *each seq in Sub-sequences* **do**
            $seq \leftarrow seq \oplus$ Tokens$_K$;
            Append $seq$ to $temp$;
        Sub-sequences $\leftarrow temp$;
        Clear $temp$;
        Input Vector $\leftarrow$ Flatten Tokens$_K$;
        Hidden State $\leftarrow$ (New Hidden State$[i]$.repeat_interleave(top $k$, dim $= 1$))$_{i=0,1}$;
**return** *Sub-sequences*

---

### B.5.2 BEAM ENUMERATION WITH AUGMENTED MEMORY

This subsection contains the Augmented Memory pseudo-code (from Guo et al. (Guo & Schwaller (2023)) with Beam Enumeration integrated. All Beam Enumeration operations are bolded.

---

**Algorithm 2:** Beam Enumeration Integrated with Augmented Memory

---

**Input:** Prior $\pi_{\text{Prior}}$, Epochs $N$, Augmentation Rounds $A$, Scoring Function $S$, Sigma $\sigma$, Replay Buffer Size $K$, Patience $P$

**Output:** Fine-tuned Agent Policy $\pi_{\theta_{\text{Agent}}}$, Generated Molecules $G$

**Initialization:**

Generative Agent $\pi_{\theta_{\text{Agent}}} = \pi_{\text{Prior}}$;

Diversity Filter $DF$;

Replay Buffer $B = \{\}$;

Substructure Pool $Pool = \{\}$;

**for** $i \leftarrow 1$ **to** $N$ **do**

    Sample batch of SMILES $X = \{x_1, \ldots, x_b\}$ with $x_i \sim \pi_{\theta_{\text{Agent}}}$;

    **if** *Pool is not empty* **then**

        ⎿ **Filter sampled batch of SMILES to contain pooled substructures** $X_{filtered}$

    ;

    Compute reward using the scoring function $S(X_{filtered})$;

    Modify reward based on the diversity filter $DF(S(X_{filtered}))$;

    Update replay buffer $B_i = TopK(X_{filtered_i} \cup B_{i-1})$;

    **if** *reward has improved for $P$ successive epochs* **then**

        ⎿ **Execute Beam Enumeration to update the** $Pool$

    ;

    (Optionally) purge replay buffer;

    Compute Augmented Likelihood $\log \pi_{\theta_{\text{Augmented}}} = \log \pi_{\text{Prior}}(X_{filtered}) + \sigma S(X_{filtered})$;

    Compute loss $J(\theta) = (\log \pi_{\text{Augmented}} - \log \pi_{\theta_{\text{Agent}}}(X_{filtered}))^2$;

    Update the Agent's policy $\pi_{\theta_{\text{Agent}}}$;

    **for** $j \leftarrow 1$ **to** $A$ **do**

        Augment filtered SMILES $X_{filtered_{\text{Augmented}}}$;

        Compute Augmented Likelihood of augmented filtered SMILES (reward is unchanged) $\log \pi_{\text{Augmented}} = \log \pi_{\text{Prior}}(X_{filtered_{\text{Augmented}}}) + \sigma S(X_{filtered})$;

        Compute loss $J(\theta)_{\text{Augmented}} = (\log \pi_{\text{Augmented}} - \log \pi_{\theta_{\text{Agent}}}(X_{filtered_{\text{Augmented}}}))^2$;

        Augment entire replay buffer $B_{\text{Augmented}}$;

        Compute Augmented Likelihood on the augmented buffer (reward is the buffer stored rewards) $\log \pi_{\text{Buffer Augmented}} = \log \pi_{\text{Prior}}(B_{\text{Augmented}}) + \sigma S(B)$;

        Compute augmented buffer loss $J(\theta)_{\text{Buffer Augmented}} = (\log \pi_{\text{Buffer Augmented}} - \log \pi_{\theta_{\text{Agent}}}(B_{\text{Augmented}}))^2$;

        Concatenate the augmented sampled SMILES loss and the augmented buffer loss $J(\theta)_{\text{Augmented Memory}} = J(\theta)_{\text{Augmented}} + J(\theta)_{\text{Buffer Augmented}}$;

        Update the Agent's policy $\pi_{\theta_{\text{Agent}}}$;

---

# C ILLUSTRATIVE EXPERIMENT

This section contains additional results from initial investigations into the feasibility of Beam Enumeration. The illustrative experiment was performed with the following multi-parameter optimization (MPO) objective: maximize topological polar surface area (tPSA), molecular weight (MW) $<350$ Da, number of rings $\geq 2$.

## C.1 SUBSTRUCTURE EXTRACTION

The first experiments investigated whether a sufficient substructures signal could be extracted from enumerated sub-sequences. The two parameters of Beam Enumeration (without self-conditioning) are top $k$ denoting the top $k$ number of highest probability tokens to enumerate and $N$ number of beam steps denoting how many steps to perform token expansion for (which is also the length of the final sub-sequence). Our hypothesis is that a lower top $k$ is desirable as we are interested in the most probably substructures. Thus, the initial experiments were a grid-search with a top $k$ of 2 and $N$ beam steps of [15, 16, 17, 18]. The illustrative experiment was run for 100 epochs (6,400 oracle calls which is different from the 5,000 used in the main text experiments as this set of results is only to demonstrate that meaningful substructures can be extracted) and Beam Enumeration was applied at epochs 1, 20, 40, 60, 80, and 100.

Table 3: Feasibility of Beam Enumeration to extract valid substructures. Top-$k = 2$.

| $N$ Beam Steps | Epoch 1 | Epoch 20 | Epoch 40 | Epoch 60 | Epoch 80 | Epoch 100 |
|---|---|---|---|---|---|---|
| 15 | 2294/32768 (7.00%) | 3123/32768 (9.53%) | 5843/32768 (17.83%) | 5538/32768 (16.90%) | 5674/32768 (17.32%) | 8004/32768 (24.43%) |
| 16 | 4789/65536 (7.31%) | 5890/65536 (8.99%) | 5771/65536 (8.81%) | 11159/65536 (17.03%) | 7657/65536 (11.68%) | 9771/65536 (14.91%) |
| 17 | 9998/131072 (7.63%) | 15266/131072 (11.65%) | 26163/131072 (19.96%) | 24352/131072 (18.58%) | 21442/131072 (16.36%) | 31160/131072 (23.77%) |
| 18 | 20747/262144 (7.91%) | 33969/262144 (12.96%) | 72126/262144 (27.51%) | 48417/262144 (18.47%) | 45349/262144 (17.30%) | 46994/262144 (17.93%) |

Table 3 shows the absolute counts and percentage of sub-sequences containing valid substructures. While the percentage may appear low, we note the absolute counts is more than enough to extract some notion of most probable substructures. We use $N$ beam steps of 18 for all experiments as we hypothesize that larger substructures can carry more information. The reason the max beam steps investigated was 18 is because of the memory overhead required for sequence expansion.

## C.2 EXTRACTED SUBSTRUCTURES

To illustrate the capability of Beam Enumeration to extract meaningful substructures, Fig. C11 shows the top 5 most probable substructures at epochs 1, 20, 40, 60, 80, and 100 based on *Structure* (extract any valid structure) and *Scaffold* (extract valid Bemis-Murcko (Bemis & Murcko (1996) scaffold) using a top $k$ of 2 and 18 beam steps. We make two crucial observations here. Firstly, *Structure* often extracts small functional groups which makes the self-conditioned filtering much more permissive as it is more likely for a molecule to possess a specific functional group than a specific scaffold. Secondly, benzene appears often and perhaps unsurprisingly as it is ubiquitous in nature. Based on these observations, we design Beam Enumeration to only extract substructures containing at least one heteroatom on the assumption that heteroatoms are much more informative in forming polar interactions in drug molecules, e.g., a hydrogen-bond cannot form from benzene. Finally, the general observation is that the most probable substructures gradually contain more heteroatoms, as desired.

## C.3 SUPPLEMENTARY MAIN TEXT RESULTS

In this section, we present the same table as the main text illustrative experiment. The only difference is that the IntDiv1 (Polykovskiy et al. (2020a) is also annotated in the table here to show that the sample efficiency improvements of Beam Enumeration come only at a small trade-off in diversity (Table 4). In agreement with our observations in the hyperparameters grid search (Appendix

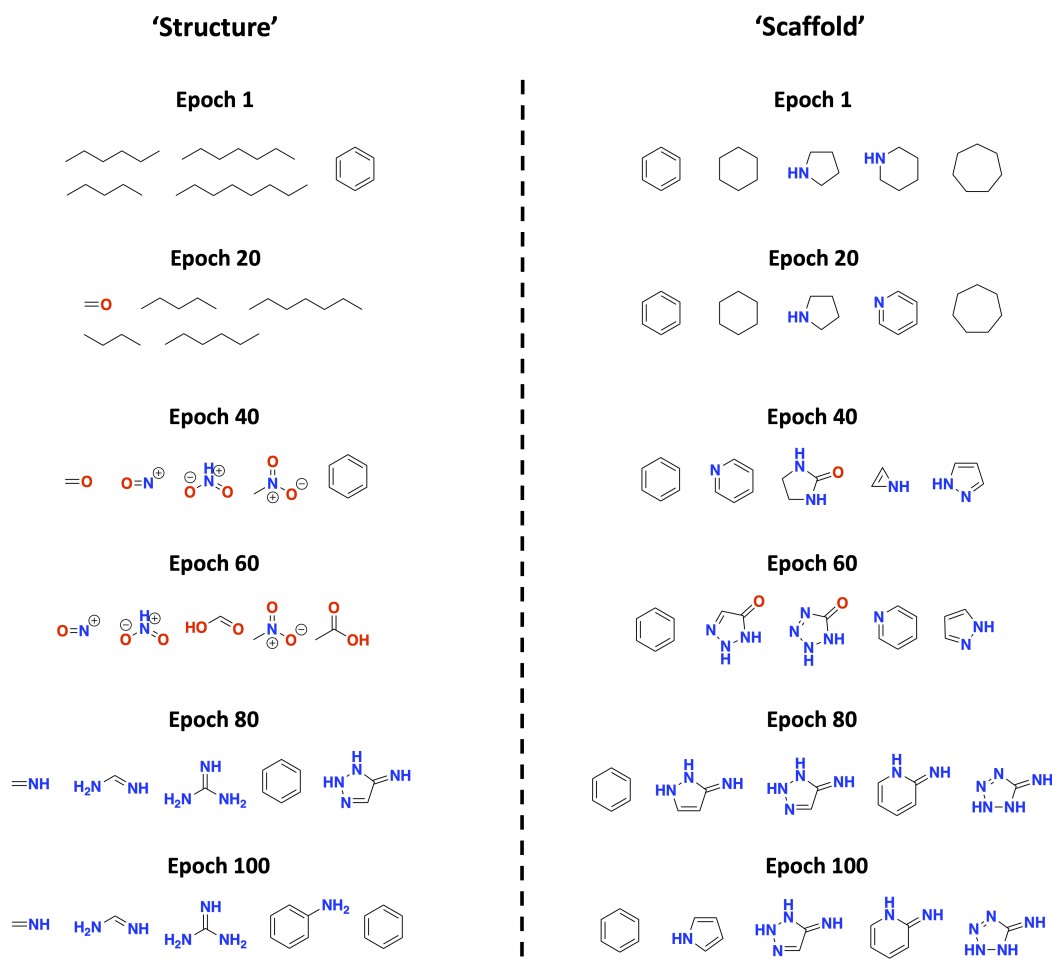

Figure C11: Substructures extracted in the illustrative example at varying epochs based on *Structure* and *Scaffold*.

A), *Structure* extraction with 'Structure Minimum Size' enforcement leads to highly specific substructures which decrease diversity relative to *Scaffold* extraction but with potential gains in sample efficiency as evidenced in the drug discovery case studies (Appendix D). We further perform statistical testing using Welch's t-test to compare all metrics for *Scaffold* with 'Structure Minimum Size' = 15 and Baseline Augmented Memory (Guo & Schwaller (2023). For the experiments that had unsuccessful replicates, we use the total number of successful experiments, e.g., Oracle Burden$_{>10}$ (100), the Baseline was unsuccessful in 69/100 replicates so a 31 sample size was used. Overall, all p-values are significant at the 95% confidence level.

Table 4: Illustrative experiment: Beam Enumeration improves the sample efficiency of Augmented Memory. All experiments were run for 100 replicates with an oracle budget of 5,000 calls and reported values are the mean and standard deviation. *Scaffold* and *Structure* indicate the type of substructure and the number after is the 'Structure Minimum Size'. Parentheses after Oracle Burden denote the cut-off number of molecules. Parentheses after values represent the number of unsuccessful replicates (for achieving the metric). The IntDiv1 (Polykovskiy et al. (2020b) is annotated under each Generative Yield. Welch's t-test is used to compare the difference between *Scaffold* with 'Structure Minimum Size' = 15 and Baseline Augmented Memory (Guo & Schwaller (2023). All p-values are significant.

| Metric | Augmented Memory | | | | | Welch's t-test (95%) |
|---|---|---|---|---|---|---|
| | Beam Scaffold 15 | Beam Structure 15 | Beam Scaffold | Beam Structure | Baseline | p-value (N=100) |
| Generative Yield$_{>0.7}$ ($\uparrow$) | **1757 $\pm$ 305** | 1669 $\pm$ 389 | 1117 $\pm$ 278 | 864 $\pm$ 202 | 496 $\pm$ 108 | $2.60 \times 10^{-75}$ |
| - Diversity | 0.77 $\pm$ 0.03 | 0.73 $\pm$ 0.04 | 0.79 $\pm$ 0.03 | 0.83 $\pm$ 0.03 | 0.85 $\pm$ 0.02 | |
| Generative Yield$_{>0.8}$ ($\uparrow$) | **819 $\pm$ 291** | 700 $\pm$ 389 | 425 $\pm$ 256 | 199 $\pm$ 122 | 85 $\pm$ 56 | $3.70 \times 10^{-48}$ |
| - Diversity | 0.73 $\pm$ 0.04 | 0.69 $\pm$ 0.05 | 0.75 $\pm$ 0.04 | 0.77 $\pm$ 0.04 | 0.78 $\pm$ 0.03 | |
| Oracle Burden$_{>0.7}$ (1) ($\downarrow$) | **577 $\pm$ 310** | 616 $\pm$ 230 | 1037 $\pm$ 414 | 897 $\pm$ 347 | 1085 $\pm$ 483 | $3.06 \times 10^{-19}$ |
| Oracle Burden$_{>0.7}$ (10) ($\downarrow$) | 947 $\pm$ 350 | **926 $\pm$ 332** | 1881 $\pm$ 259 | 1745 $\pm$ 292 | 2392 $\pm$ 216 | $4.99 \times 10^{-87}$ |
| Oracle Burden$_{>0.7}$ (100) ($\downarrow$) | **1530 $\pm$ 468** | 1547 $\pm$ 513 | 2736 $\pm$ 335 | 2713 $\pm$ 402 | 3672 $\pm$ 197 | $2.34 \times 10^{-86}$ |
| Oracle Burden$_{>0.8}$ (1) ($\downarrow$) | **1311 $\pm$ 628** | 1401 $\pm$ 695 | 2423 $\pm$ 487 | 2295 $\pm$ 482 | 3164 $\pm$ 492 | $6.07 \times 10^{-65}$ |
| Oracle Burden$_{>0.8}$ (10) ($\downarrow$) | **1794 $\pm$ 617 (1)** | 2009 $\pm$ 804 (1) | 3124 $\pm$ 497 | 3241 $\pm$ 492 | 4146 $\pm$ 326 | $6.48 \times 10^{-79}$ |
| Oracle Burden$_{>0.8}$ (100) ($\downarrow$) | **2704 $\pm$ 689 (1)** | 2943 $\pm$ 811 (6) | 3973 $\pm$ 592 (6) | 4415 $\pm$ 437 (20) | 4827 $\pm$ 170 (69) | $6.17 \times 10^{-21}$ |

## C.4 BEAM ENUMERATION WORKS IN EXPLOITATION SCENARIOS

Table 5: Beam Enumeration works in exploitation scenarios. all experiments were run for 100 replicates with an oracle budget of 5,000 calls and reported values are the mean and standard deviation. Parentheses after Oracle Burden denote the cut-off number of molecules. The IntDiv1 (Polykovskiy et al. (2020b) is annotated under each Generative Yield. Welch's t-test is used to compare the difference between *Scaffold* with 'Structure Minimum Size' = 15 and Baseline Augmented Memory (Guo & Schwaller (2023). All p-values are significant.

| Metric | Augmented Memory | | Welch's t-test (95%) |
|---|---|---|---|
| | Beam Scaffold 15 | Baseline | p-value (N=100) |
| Generative Yield$_{>0.7}$ ($\uparrow$) | **1325 $\pm$ 468** | 496 $\pm$ 108 | $1.54 \times 10^{-29}$ |
| - Diversity | 0.76 $\pm$ 0.04 | 0.85 $\pm$ 0.02 | |
| Generative Yield$_{>0.8}$ ($\uparrow$) | **601 $\pm$ 298** | 85 $\pm$ 56 | $1.35 \times 10^{-28}$ |
| - Diversity | 0.70 $\pm$ 0.09 | 0.78 $\pm$ 0.03 | |
| Oracle Burden$_{>0.7}$ (1) ($\downarrow$) | **626 $\pm$ 260** | 1085 $\pm$ 483 | $4.52 \times 10^{-15}$ |
| Oracle Burden$_{>0.7}$ (10) ($\downarrow$) | **997 $\pm$ 326** | 2392 $\pm$ 216 | $2.26 \times 10^{-80}$ |
| Oracle Burden$_{>0.7}$ (100) ($\downarrow$) | **1487 $\pm$ 352** | 3672 $\pm$ 197 | $4.01 \times 10^{-100}$ |
| Oracle Burden$_{>0.8}$ (1) ($\downarrow$) | **1415 $\pm$ 645** | 3164 $\pm$ 492 | $2.21 \times 10^{-53}$ |
| Oracle Burden$_{>0.8}$ (10) ($\downarrow$) | **1794 $\pm$ 553 (2)** | 4146 $\pm$ 326 | $1.14 \times 10^{-76}$ |
| Oracle Burden$_{>0.8}$ (100) ($\downarrow$) | **2490 $\pm$ 576 (2)** | 4827 $\pm$ 170 (69) | $1.68 \times 10^{-25}$ |

In the main text illustrative experiment, Augmented Memory (Guo & Schwaller (2023) was used with Selective Memory Purge activated which is the mechanism to promote chemical space exploration, as described in the original work. For completeness, we show that Beam Enumeration also works in pure exploitation scenarios where the goal is only to generate high reward molecules even if the same molecule is repeatedly sampled (Table 5). We perform statistical testing using Welch's t-test to compare all metrics for *Scaffold* with 'Structure Minimum Size' = 15 and Baseline Augmented Memory (Guo & Schwaller (2023). For the experiments that had unsuccessful replicates,

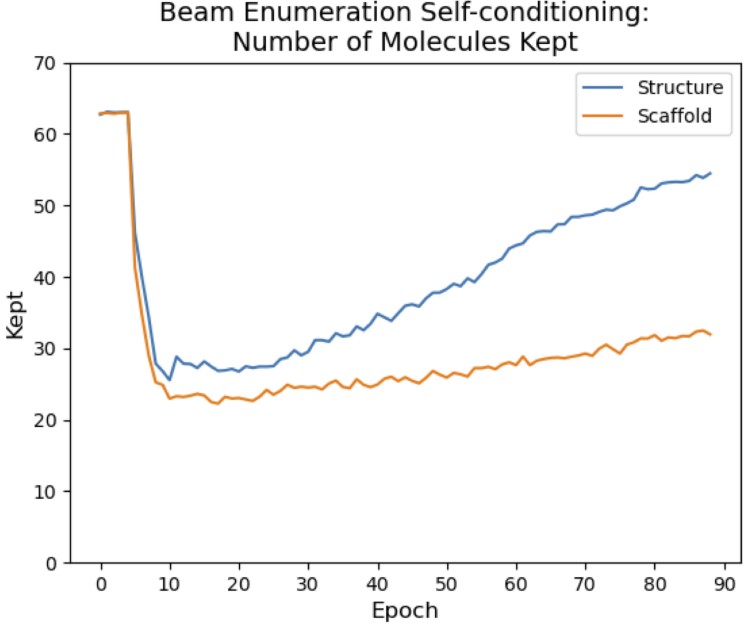

Figure C12: Behaviour of Beam Enumeration using *Structure* and *Scaffold* on self-conditioning.

we use the total number of successful experiments, e.g., Oracle Burden$_{>10}$ (100), the Baseline was unsuccessful in 69/100 replicates so a 31 sample size was used. Overall, all p-values are significant at the 95% confidence level.

### C.5 SELF-CONDITIONED FILTERING: *Structure* VS *Scaffold*

There is a clear discrepancy in the substructures extracted by *Structure* and *Scaffold*. In particular, *Structure* substructures contain small functional groups which is much more permissive when used as a filter criterion compared to full scaffolds. Therefore, one would expect that many molecules in the sampled batches would be kept when using *Structure* Beam Enumeration. We plot the average number of molecules kept out of 64 (batch size) across the generative run when using Beam Enumeration. Note that the experiments ran for variable epochs due to the stochasticity of Beam Enumeration self-filtering. The number of epochs shown in C12 is the minimum number of epochs out of 100 replicates. Therefore, the average values shown are averaged over 100 replicates. It is evident that *Structure* is more lenient as many generated molecules make it through the filter compared to *Scaffold* which maintains a relatively strict filter. One interesting observation is that self-conditioning does not lead to obvious mode collapse. Self-conditioning is inherently biased and one would be concerned that the model gets stuck at generating the same molecules repeatedly. The fact that self-conditioning with *Scaffold* continues to filter throughout the entire generative run shows that the model is continually moving to new chemical space, supporting findings from the original Augmented Memory (Guo & Schwaller (2023) work that Selective Memory Purge (built-in diversity mechanism) is capable of preventing mode collapse.

## D    DRUG DISCOVERY CASE STUDIES

This section contains information on the Autodock Vina (Trott & Olson (2010) docking protocol from receptor grid preparation to docking execution. The Beam Enumeration hyperparameters grid search results are presented for all three drug discovery case studies followed by analysis. Examples of extracted substructures are also shown and commentary provided to their significance and explainability. Finally, the wall times of all experiments are presented.

## D.1 AUTODOCK VINA RECEPTOR PREPARATION AND DOCKING

All docking grids were prepared using DockStream (Guo et al. (2021a) which uses PDBFixer (Eastman et al. (2017) to refine receptor structures. The search box for all grids was 15Å x 15Å x 15Å. Docking was also performed through DockStream and followed a two step process: conformer generation using the RDKit Universal Force Field (UFF) (Rappé et al. (1992) with the maximum convergence set to 600 iterations and then Vina docking was parallelized over 36 CPU cores (Intel(R) Xeon(R) Platinum 8360Y processors).

**DRD2 - Dopamine Type 2 Receptor.** The PDB ID is 6CM4 (Wang et al. (2018) and the docking grid was centered at (x, y, z) = (9.93, 5.85, -9.58).

**MK2 - MK2 Kinase.** The PDB ID is 3KC3 (Argiriadi et al. (2010) and one monomer was extracted. The docking grid for the extracted monomer was centered at (x, y, z) = (-61.62, 30.31, -21.9).

**AChE - Acetylcholinesterase.** The PDB ID is 1EVE (Kryger et al. (1999) and the docking grid was centered at (x, y, z) = (2.78, 64.38, 67.97).

## D.2 BEAM ENUMERATION HYPERPARAMETERS GRID SEARCH RESULTS

We performed an additional hyperparameter grid search on all three drug discovey case studies based on the insights drawn from the illustrative experiment grid search results. We fix the following hyperparameters:

- Beam K = 2
- Beam Steps = 18
- Pool Size = 4
- Patience = 5

and vary the following:

- Optimization Algorithm = [Augmented Memory (Guo & Schwaller (2023), REINVENT (Olivecrona et al. (2017); Blaschke et al. (2020)]
- Substructure Type = [*Scaffold*, *Structure*]
- Structure Minimum Size = [10, 15]
- Token Sampling Method = ["topk", "sample"]

**All hyperparameter combinations (8) were tried and run for 3 replicates each for statistical reproducibility, total of 144 experiments.** There are two main results we want to convey: firstly, the optimal hyperparameters are the same for all three drug discovery case studies and only the Substructure Type differs between the optimal hyperparameters here and the illustrative experiment. Secondly, Beam Enumeration is a task-agnostic general method that can be applied to existing algorithms including Augmented Memory (Guo & Schwaller (2023) and REINVENT (Olivecrona et al. (2017); Blaschke et al. (2020). At the end of this section, we present these hyperparameters and designate these the default values. All grid search results are now presented in following tables:

Based on the results from the hyperparameters grid search in the drug discovery case studies, we make two key observations: firstly, *Structure* extraction with 'Structure Minimum Size' = 15 is now the most performant, on average (for both Augmented Memory (Guo & Schwaller (2023) and REINVENT (Olivecrona et al. (2017); Blaschke et al. (2020)). This is in contrast to *Scaffold* extraction in the illustrative experiment which we rationalize through the permissive nature of the experiment compared to the docking experiments which require structure specificity. Previously, small deviations in the substructures may not have a significant impact on the reward. In physics-based oracles such as Vina (Trott & Olson (2010) docking used here, small substructure differences can have an enormous impact on the outcome since the pose requires specific complementary to the protein binding site. The second observation we make which is in agreement with the illustrative experiment is that "sample" token sampling has more variance and does not perform better than "topk". The rationale is the same in that docking requires specificity and lower probability substructures exhibit more variable performance. Based on all the observations from the illustrative experiment and the drug discovery case studies, we designate the following default hyperparameter values:

Table 6: DRD2 (Wang et al. (2018) case study hyperparameters grid search results for Augmented Memory (Guo & Schwaller (2023). All experiments were run in triplicate and the reported values are the mean and standard deviation. "Sample" denotes "sample" token sampling. All metrics are for the reward threshold >0.8. The IntDiv1 (Polykovskiy et al. (2020b) is annotated under Generative Yield. * and ** denote one and two replicates were unsuccessful, respectively.

| Experiment Augmented Memory DRD2 | Generative Yield | Unique Scaffolds | Oracle Burden (1) | Oracle Burden (10) | Oracle Burden (100) |
|---|---|---|---|---|---|
| Baseline | $363 \pm 195$ | $322 \pm 166$ | $187 \pm 51$ | $711 \pm 120$ | $2558 \pm 30^*$ |
| - Diversity | $0.802 \pm 0.019$ | | | | |
| Scaffold | $957 \pm 75$ | $749 \pm 62$ | $82 \pm 29$ | $668 \pm 25$ | $1818 \pm 107$ |
| - Diversity | $0.765 \pm 0.006$ | | | | |
| Scaffold Size 15 | $1607 \pm 379$ | $1023 \pm 351$ | $83 \pm 29$ | $571 \pm 104$ | $1056 \pm 146$ |
| - Diversity | $0.724 \pm 0.027$ | | | | |
| Scaffold Sample | $948 \pm 123$ | $776 \pm 128$ | $126 \pm 89$ | $505 \pm 17$ | $1746 \pm 20$ |
| - Diversity | $0.734 \pm 0.018$ | | | | |
| Scaffold Sample Size 15 | $1552 \pm 106$ | $1274 \pm 154$ | $84 \pm 29$ | $598 \pm 110$ | $1511 \pm 416$ |
| - Diversity | $0.660 \pm 0.041$ | | | | |
| Structure | $887 \pm 112$ | $711 \pm 133$ | $63 \pm 0$ | $595 \pm 63$ | $1862 \pm 154$ |
| - Diversity | $0.764 \pm 0.008$ | | | | |
| Structure Size 15 | $1780 \pm 439$ | $1323 \pm 368$ | $126 \pm 90$ | $582 \pm 83$ | $1120 \pm 194$ |
| - Diversity | $0.699 \pm 0.020$ | | | | |
| Structure Sample | $912 \pm 86$ | $757 \pm 30$ | $63 \pm 0$ | $583 \pm 37$ | $2132 \pm 148$ |
| - Diversity | $0.767 \pm 0.015$ | | | | |
| Structure Sample Size 15 | $1752 \pm 105$ | $1352 \pm 180$ | $188 \pm 103$ | $776 \pm 129$ | $1289 \pm 193$ |
| - Diversity | $0.641 \pm 0.059$ | | | | |

Table 7: DRD2 (Wang et al. (2018) case study hyperparameters grid search results for REINVENT (Olivecrona et al. (2017); Blaschke et al. (2020). All experiments were run in triplicate and the reported values are the mean and standard deviation. "Sample" denotes "sample" token sampling. The IntDiv1 (Polykovskiy et al. (2020b) is annotated under Generative Yield. All metrics are for the reward threshold >0.8. * and ** denote one and two replicates were unsuccessful, respectively.

| Experiment REINVENT DRD2 | Generative Yield | Unique Scaffolds | Oracle Burden (1) | Oracle Burden (10) | Oracle Burden (100) |
|---|---|---|---|---|---|
| Baseline | $102 \pm 6$ | $101 \pm 6$ | $168 \pm 149$ | $883 \pm 105$ | $4595 \pm 0^{**}$ |
| - Diversity | $0.833 \pm 0.001$ | | | | |
| Scaffold | $190 \pm 32$ | $184 \pm 32$ | $63 \pm 1$ | $836 \pm 178$ | $3516 \pm 575$ |
| - Diversity | $0.814 \pm 0.007$ | | | | |
| Scaffold Size 15 | $687 \pm 366$ | $377 \pm 204$ | $127 \pm 52$ | $604 \pm 71$ | $2109 \pm 1090$ |
| - Diversity | $0.730 \pm 0.013$ | | | | |
| Scaffold Sample | $176 \pm 86$ | $149 \pm 49$ | $105 \pm 59$ | $720 \pm 121$ | $3875 \pm 883$ |
| - Diversity | $0.801 \pm 0.030$ | | | | |
| Scaffold Sample Size 15 | $363 \pm 249$ | $225 \pm 144$ | $84 \pm 30$ | $754 \pm 183$ | $3170 \pm 1188$ |
| - Diversity | $0.704 \pm 0.044$ | | | | |
| Structure | $184 \pm 14$ | $183 \pm 14$ | $104 \pm 31$ | $897 \pm 100$ | $3426 \pm 282$ |
| - Diversity | $0.817 \pm 0.006$ | | | | |
| Structure Size 15 | $417 \pm 275$ | $290 \pm 178$ | $63 \pm 0$ | $1099 \pm 930$ | $1928 \pm 117^*$ |
| - Diversity | $0.730 \pm 0.014$ | | | | |
| Structure Sample | $169 \pm 24$ | $167 \pm 24$ | $126 \pm 52$ | $711 \pm 179$ | $3568 \pm 440$ |
| - Diversity | $0.826 \pm 0.003$ | | | | |
| Structure Sample Size 15 | $261 \pm 225$ | $182 \pm 132$ | $209 \pm 128$ | $840 \pm 107$ | $3690 \pm 1266^*$ |
| - Diversity | $0.734 \pm 0.057$ | | | | |

- Beam K = 2

- Beam Steps = 18

- Pool Size = 4

- Patience = 5

- Substructure Type = *Structure*

- Structure Minimum Size = 15

- Token Sampling Method = "topk"

Table 8: MK2 (Argiriadi et al. (2010) case study hyperparameters grid search results for Augmented Memory (Guo & Schwaller (2023). All experiments were run in triplicate and the reported values are the mean and standard deviation. "Sample" denotes "sample" token sampling. All metrics are for the reward threshold >0.8. The IntDiv1 (Polykovskiy et al. (2020b) is annotated under Generative Yield. * and ** denote one and two replicates were unsuccessful, respectively.

| Experiment Augmented Memory MK2 | Generative Yield | Unique Scaffolds | Oracle Burden (1) | Oracle Burden (10) | Oracle Burden (100) |
|---|---|---|---|---|---|
| Baseline - Diversity | $34 \pm 13$ $0.794 \pm 0.008$ | $32 \pm 12$ | $1360 \pm 543$ | $3833 \pm 394$ | Failed |
| Scaffold - Diversity | $179 \pm 63$ $0.743 \pm 0.038$ | $131 \pm 16$ | $1163 \pm 457$ | $2550 \pm 148$ | $4421 \pm 344$ |
| Scaffold Size 15 - Diversity | $523 \pm 438$ $0.676 \pm 0.016$ | $330 \pm 269$ | $1221 \pm 564$ | $2426 \pm 1525$ | $2676 \pm 403^*$ |
| Scaffold Sample - Diversity | $106 \pm 71$ $0.722 \pm 0.017$ | $87 \pm 58$ | $1005 \pm 573$ | $3296 \pm 1181$ | $4592 \pm 334^*$ |
| Scaffold Sample Size 15 - Diversity | $379 \pm 357$ $0.653 \pm 0.026$ | $257 \pm 227$ | $983 \pm 540$ | $1846 \pm 680$ | $3244 \pm 1133^*$ |
| Structure - Diversity | $66 \pm 18$ $0.769 \pm 0.029$ | $59 \pm 20$ | $1246 \pm 716$ | $2708 \pm 232$ | Failed |
| Structure Size 15 - Diversity | $987 \pm 211$ $0.704 \pm 0.030$ | $610 \pm 117$ | $736 \pm 166$ | $1122 \pm 154$ | $2189 \pm 181$ |
| Structure Sample - Diversity | $40 \pm 15$ $0.784 \pm 0.024$ | $34 \pm 11$ | $1119 \pm 1183$ | $3516 \pm 506$ | Failed |
| Structure Sample Size 15 - Diversity | $129 \pm 52$ $0.671 \pm 0.073$ | $117 \pm 50$ | $1208 \pm 660$ | $2799 \pm 476$ | $4037 \pm 0^{**}$ |

Table 9: MK2 (Argiriadi et al. (2010) case study hyperparameters grid search results for REINVENT (Olivecrona et al. (2017); Blaschke et al. (2020). All experiments were run in triplicate and the reported values are the mean and standard deviation. "Sample" denotes "sample" token sampling. All metrics are for the reward threshold >0.8. The IntDiv1 (Polykovskiy et al. (2020b) is annotated under Generative Yield. * and ** denote one and two replicates were unsuccessful, respectively.

| Experiment REINVENT MK2 | Generative Yield | Unique Scaffolds | Oracle Burden (1) | Oracle Burden (10) | Oracle Burden (100) |
|---|---|---|---|---|---|
| Baseline - Diversity | $2 \pm 0$ $0.424 \pm 0.031$ | $2 \pm 0$ | $1723 \pm 802$ | Failed | Failed |
| Scaffold - Diversity | $7 \pm 2$ $0.704 \pm 0.051$ | $7 \pm 2$ | $1272 \pm 884$ | $4948 \pm 0^{**}$ | Failed |
| Scaffold Size 15 - Diversity | $19 \pm 7$ $0.674 \pm 0.065$ | $18 \pm 7$ | $808 \pm 524$ | $3891 \pm 631$ | Failed |
| Scaffold Sample - Diversity | $6 \pm 2$ $0.677 \pm 0.075$ | $6 \pm 2$ | $1427 \pm 343$ | Failed | Failed |
| Scaffold Sample Size 15 - Diversity | $4 \pm 2$ $0.653 \pm 0.026$ | $3 \pm 1$ | $2600 \pm 1455$ | Failed | Failed |
| Structure - Diversity | $3 \pm 1$ $0.571 \pm 0.112$ | $3 \pm 1$ | $2571 \pm 1155$ | Failed | Failed |
| Structure Size 15 - Diversity | $179 \pm 241$ $0.670 \pm 0.020$ | $70 \pm 87$ | $1110 \pm 268$ | $1778 \pm 0^{**}$ | $3208 \pm 0^{**}$ |
| Structure Sample - Diversity | $1 \pm 0$ $0.192 \pm 0.271$ | $1 \pm 0$ | $1737 \pm 1595$ | Failed | Failed |
| Structure Sample Size 15 - Diversity | $8 \pm 5$ $0.357 \pm 0.255$ | $7 \pm 4$ | $1943 \pm 1153$ | $4851 \pm 0^{**}$ | Failed |

## D.3 EXAMPLES OF EXTRACTED SUBSTRUCTURES: *Structure* EXTRACTION WITH 'STRUCTURE MINIMUM SIZE' = 15

In this section, the top substructures at the end of the generative experiments (using Augmented Memory (Guo & Schwaller (2023)) are shown for all three drug discovery case studies (3 replicates). All experiments are for *Structure* extraction with 'Structure Minimum Size' = 15. The extracted substructures are commonly scaffolds with "branch points", i.e., a central scaffold with single carbon bond extensions outward, which heavily bias generation. We posit that this may be a reason why *Structure* extraction can be more performant than *Scaffold*, as observed in the hyperparameters grid search in the previous subsection.

**DRD2 – 'Structure' Extraction with 'Structure Minimum Size' = 15**

**Replicate 1**

**Replicate 2**

**Replicate 3**

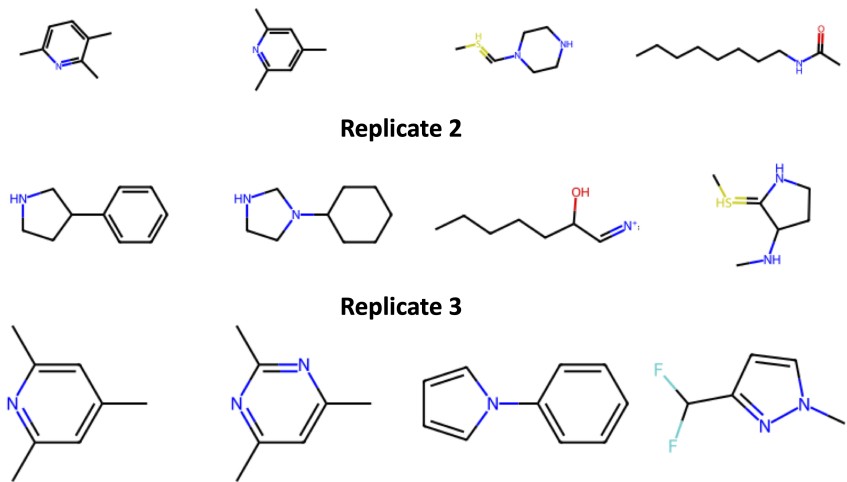

Figure D13: Augmented Memory (Guo & Schwaller (2023) DRD2 (Wang et al. (2018) substructures with *Structure* extraction and 'Structure Minimum Size' = 15 after 5,000 oracle calls.

**MK2 – 'Structure' Extraction with 'Structure Minimum Size' = 15**

**Replicate 1**

**Replicate 2**

**Replicate 3**

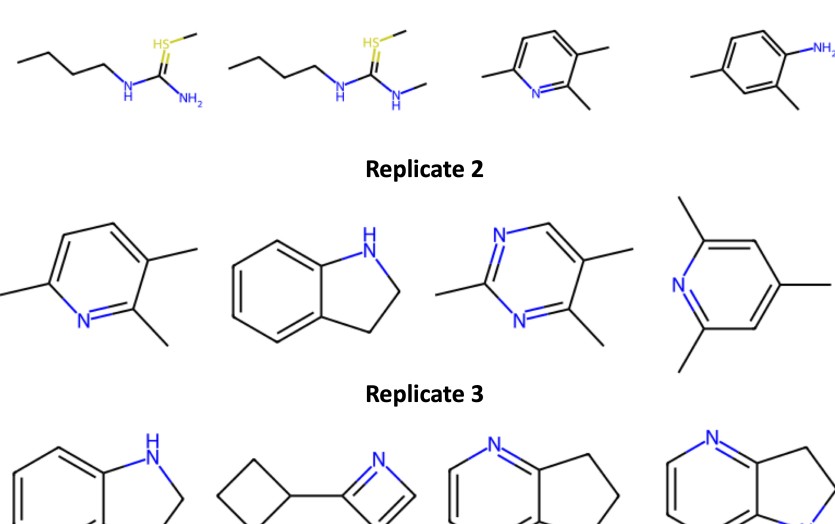

Figure D14: Augmented Memory (Guo & Schwaller (2023) MK2 (Argiriadi et al. (2010) substructures with *Structure* extraction and 'Structure Minimum Size' = 15 after 5,000 oracle calls.

Table 10: AChE (Kryger et al. (1999) case study hyperparameters grid search results for Augmented memory (Guo & Schwaller (2023). All experiments were run in triplicate and the reported values are the mean and standard deviation. "Sample" denotes "sample" token sampling. All metrics are for the reward threshold >0.8. The IntDiv1 (Polykovskiy et al. (2020b) is annotated under Generative Yield. * and ** denote one and two replicates were unsuccessful, respectively.

| Experiment Augmented Memory AChE | Generative Yield | Unique Scaffolds | Oracle Burden (1) | Oracle Burden (10) | Oracle Burden (100) |
|---|---|---|---|---|---|
| Baseline | $556 \pm 47$ | $544 \pm 50$ | $62 \pm 0$ | $380 \pm 0$ | $2021 \pm 89$ |
| - Diversity | $0.838 \pm 0.002$ | | | | |
| Scaffold | $1058 \pm 102$ | $1006 \pm 113$ | $62 \pm 0$ | $430 \pm 90$ | $1469 \pm 56$ |
| - Diversity | $0.823 \pm 0.005$ | | | | |
| Scaffold Size 15 | $2124 \pm 326$ | $1523 \pm 260$ | $63 \pm 0$ | $418 \pm 27$ | $884 \pm 162$ |
| - Diversity | $0.752 \pm 0.029$ | | | | |
| Scaffold Sample | $1187 \pm 48$ | $1075 \pm 39$ | $84 \pm 29$ | $409 \pm 77$ | $1519 \pm 141$ |
| - Diversity | $0.806 \pm 0.003$ | | | | |
| Scaffold Sample Size 15 | $1295 \pm 126$ | $1168 \pm 143$ | $188 \pm 103$ | $602 \pm 108$ | $1440 \pm 115$ |
| - Diversity | $0.750 \pm 0.021$ | | | | |
| Structure | $992 \pm 64$ | $946 \pm 52$ | $105 \pm 59$ | $558 \pm 94$ | $1635 \pm 81$ |
| - Diversity | $0.823 \pm 0.005$ | | | | |
| Structure Size 15 | $2059 \pm 327$ | $1552 \pm 344$ | $105 \pm 29$ | $462 \pm 25$ | $1110 \pm 265$ |
| - Diversity | $0.735 \pm 0.017$ | | | | |
| Structure Sample | $831 \pm 126$ | $790 \pm 130$ | $62 \pm 1$ | $357 \pm 29$ | $1617 \pm 220$ |
| - Diversity | $0.841 \pm 0.003$ | | | | |
| Structure Sample Size 15 | $1277 \pm 526$ | $1031 \pm 421$ | $127 \pm 52$ | $800 \pm 342$ | $1879 \pm 531$ |
| - Diversity | $0.657 \pm 0.070$ | | | | |

Table 11: AChE (Kryger et al. (1999) case study hyperparameters grid search results for REINVENT (Olivecrona et al. (2017); Blaschke et al. (2020). All experiments were run in triplicate and the reported values are the mean and standard deviation. "Sample" denotes "sample" token sampling. The IntDiv1 (Polykovskiy et al. (2020b) is annotated under Generative Yield. All metrics are for the reward threshold >0.8. * and ** denote one and two replicates were unsuccessful, respectively.

| Experiment REINVENT AChE | Generative Yield | Unique Scaffolds | Oracle Burden (1) | Oracle Burden (10) | Oracle Burden (100) |
|---|---|---|---|---|---|
| Baseline | $147 \pm 11$ | $146 \pm 11$ | $83 \pm 29$ | $481 \pm 108$ | $3931 \pm 286$ |
| - Diversity | $0.852 \pm 0.004$ | | | | |
| Scaffold | $245 \pm 50$ | $244 \pm 50$ | $63 \pm 0$ | $566 \pm 136$ | $3360 \pm 164$ |
| - Diversity | $0.844 \pm 0.003$ | | | | |
| Scaffold Size 15 | $310 \pm 207$ | $227 \pm 159$ | $84 \pm 29$ | $421 \pm 120$ | $3596 \pm 678$ |
| - Diversity | $0.744 \pm 0.038$ | | | | |
| Scaffold Sample | $257 \pm 77$ | $252 \pm 76$ | $63 \pm 0$ | $480 \pm 60$ | $2946 \pm 460$ |
| - Diversity | $0.847 \pm 0.004$ | | | | |
| Scaffold Sample Size 15 | $310 \pm 92$ | $271 \pm 70$ | $148 \pm 28$ | $673 \pm 107$ | $2881 \pm 475$ |
| - Diversity | $0.759 \pm 0.039$ | | | | |
| Structure | $356 \pm 22$ | $351 \pm 24$ | $63 \pm 0$ | $294 \pm 28$ | $2284 \pm 238$ |
| - Diversity | $0.841 \pm 0.002$ | | | | |
| Structure Size 15 | $323 \pm 58$ | $284 \pm 71$ | $62 \pm 0$ | $441 \pm 132$ | $3073 \pm 427$ |
| - Diversity | $0.795 \pm 0.009$ | | | | |
| Structure Sample | $213 \pm 26$ | $206 \pm 22$ | $84 \pm 30$ | $558 \pm 222$ | $3073 \pm 279$ |
| - Diversity | $0.844 \pm 0.005$ | | | | |
| Structure Sample Size 15 | $316 \pm 253$ | $190 \pm 146$ | $125 \pm 50$ | $561 \pm 140$ | $2683 \pm 320^*$ |
| - Diversity | $0.721 \pm 0.111$ | | | | |

## D.4 WALL TIMES

The wall times for all drug discovery case studies with every algorithm is presented in Table 12. The reported values are averaged over 3 replicates. In general, adding Beam Enumeration to the base Augmented Memory (Guo & Schwaller (2023) and REINVENT (Olivecrona et al. (2017); Blaschke et al. (2020) algorithms increased wall times but only slightly and it is negligible when considering expensive oracles. An interesting observation is that "sample" token sampling increases wall time variance. This is because less probable substructures lead to more "filter rounds', i.e., epochs where all the sampled molecules are discarded as none of them contain the Beam Enumeration extracted substructures. In addition, REINVENT generally has longer wall times even though the oracle budget is the same. The reason for this is because REINVENT optimizes the structure components of the MPO objective: QED (Bickerton et al. (2012) and MW constraint to a lesser extent. Consequently, REINVENT generates larger molecules, on average, which take longer to dock with Vina

**AChE – 'Structure' Extraction with 'Structure Minimum Size' = 15**

**Replicate 1**

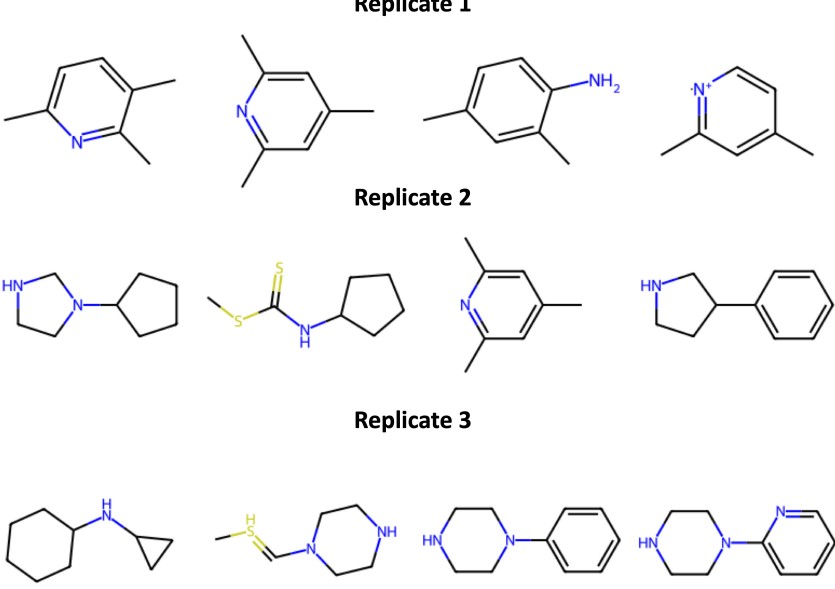

**Replicate 2**

**Replicate 3**

Figure D15: Augmented Memory (Guo & Schwaller (2023) AChE (Kryger et al. (1999) substructures with *Structure* extraction and 'Structure Minimum Size' = 15 after 5,000 oracle calls.

Table 12: Wall times for all drug discovery case studies hyperparameters grid search using Augmented Memory (Guo & Schwaller (2023) and REINVENT (Olivecrona et al. (2017); Blaschke et al. (2020). "Sample" denotes "sample" token sampling. All experiments were run in triplicate and the values are the mean and standard deviation.

| Target | Experiment | Augmented Memory Wall Time | REINVENT Wall Time |
|---|---|---|---|
| DRD2 | Baseline | 14h 0m ± 1h 26m | 16h 36m ± 0h 55m |
| | Scaffold | 12h 58m ± 1h 11m | 17h 9m ± 1h 28m |
| | Scaffold Size 15 | 12h 56m ± 0h 46m | 16h 51m ± 1h 58m |
| | Scaffold Sample | 12h 11m ± 0h 24m | 16h 32m ± 1h 3m |
| | Scaffold Sample Size 15 | 13h 32m ± 0h 50m | 16h 26m ± 2h 58m |
| | Structure | 14h 30m ± 0h 51m | 22h 5m ± 1h 52m |
| | Structure Size 15 | 14h 54m ± 2h 24m | 24h 33m ± 5h 8m |
| | Structure Sample | 13h 58m ± 0h 51m | 20h 5m ± 1h 42m |
| | Structure Sample Size 15 | 14h 52m ± 1h 32m | 19h 52m ± 3h 22m |
| MK2 | Baseline | 10h 46m ± 0h 3m | 15h 19m ± 0h 34m |
| | Scaffold | 11h 0m ± 0h 28m | 16h 21m ± 0h 53m |
| | Scaffold Size 15 | 11h 22m ± 2h 30m | 16h 38m ± 1h 33m |
| | Scaffold Sample | 12h 56m ± 0h 36m | 15h 49m ± 0h 36m |
| | Scaffold Sample Size 15 | 11h 52m ± 1h 5m | 16h 28m ± 0h 33m |
| | Structure | 12h 29m ± 0h 19m | 19h 40m ± 1h 55m |
| | Structure Size 15 | 11h 22m ± 1h 17m | 18h 39m ± 1h 33m |
| | Structure Sample | 12h 22m ± 0h 28m | 18h 12m ± 0h 57m |
| | Structure Sample Size 15 | 12h 37m ± 0h 47m | 16h 6m ± 1h 37m |
| AChE | Baseline | 10h 6m ± 0h 39m | 14h 12m ± 0h 59m |
| | Scaffold | 11h 46m ± 0h 51m | 15h 10m ± 1h 4m |
| | Scaffold Size 15 | 11h 10m ± 0h 44m | 15h 52m ± 1h 4m |
| | Scaffold Sample | 10h 55m ± 0h 44m | 15h 27m ± 0h 57m |
| | Scaffold Sample Size 15 | 10h 24m ± 0h 17m | 14h 53m ± 0h 53m |
| | Structure | 13h 0m ± 0h 47m | 19h 10m ± 0h 22m |
| | Structure Size 15 | 11h 26m ± 0h 51m | 18h 30m ± 0h 20m |
| | Structure Sample | 11h 23m ± 0h 22m | 15h 36m ± 0h 20m |
| | Structure Sample Size 15 | 17h 56m ± 4h 27m | 19h 16m ± 2h 43m |

(Trott & Olson (2010). This observation is in agreement with the original Augmented Memory work which compared to REINVENT.

# E  AUGMENTED MEMORY AND REINVENT MODEL HYPERPARAMETERS

Table 13: LSTM model hyperparameters for Augmented Memory (Guo & Schwaller (2023) and REINVENT (Olivecrona et al. (2017); Blaschke et al. (2020)

| Cell Type | LSTM |
|---|---|
| Number of Layers | 3 |
| Embedding Layer Size | 256 |
| Dropout | 0 |
| Training Batch Size | 128 |
| Sampling Batch Size | 64 |
| Learning Rate | 0.001 |

The same pre-trained prior on ChEMBL (Gaulton et al. (2012b) was used for Augmented Memory (Guo & Schwaller (2023) and REINVENT (Olivecrona et al. (2017); Blaschke et al. (2020). All shared hyperparameters (sampling batch size and learning rate) are the same. Default additional hyperparameters for Augmented Memory were used based on the original work (Guo & Schwaller (2023): two augmentation rounds and using Selective Memory Purge to prevent mode collapse. Experience replay (Lin (1992); Blaschke et al. (2020) was kept default in REINVENT (randomly sample 10 molecules out of 100 from the replay buffer at each epoch).

# F  BEAM ENUMERATION: BEYOND LANGUAGE-BASED FORMULATIONS

This section discusses potential extensions of Beam Enumeration to other classes of molecular generative models beyond language-based formulations.

## F.1  DIRECT APPLICATION

Beam Enumeration enumerates from local probability distributions and is thus well-suited for autoregressive models. Correspondingly, autoregressive molecular generative models are not limited to language-based formulations. As a concrete example, GraphINVENT (Mercado et al. (2021b;a) generates molecular graphs and has been coupled with RL (Atance et al. (2022). During the optimization process, instead of token probabilities being updated, *graph action probabilities* are tuned, which denote discrete actions to construct the molecular graph, such as adding a node or edge. Beam Enumeration can be directly applied by exhaustively enumerating the most probable set of graph actions and extracting substructures (conditional on the molecular graph being valid).

## F.2  APPLYING OBSERVED FINDINGS

Many generative models construct a substructure "vocabulary" such that the optimization process is defined by *how* to combine these building blocks (Jin et al. (2018); Guo et al. (2022). This is similar to token sampling, except the "tokens" are now "words" (groups of atoms). The empirical results from Beam Enumeration show that substructure biasing is beneficial, even early in a generative experiment when the model has not necessarily learned to generate favorable molecules. Given a promising seed molecule, existing models such as the JT-VAE (Jin et al. (2018) and the Data Efficient Graph Grammar (DEG) (Guo et al. (2022) models can be used to propose local neighbourhood expansions. Scoring these molecules enable the possibility of iteratively performing substructure (of increasing size) biasing, effectively leveraging both the advantages of the learned "vocabulary" of building blocks, and the iterative construction of increasingly favorable substructures. Results in this work suggest that this can be performant as enforcing larger substructure size *always* improved sample efficiency (up to a size of 15 in this work).

# G    BEAM ENUMERATION VS. BEAM SEARCH

Beam Enumeration is distinct from Beam Search (Graves (2012); Boulanger-Lewandowski et al. (2013). In this section, we contrast the behavior point-by-point.

**Objective:** Beam Search approximates finding the highest probability sequences by maintaining a "width" of the top k candidates at each step. The number of sequences decoded via Beam Search is exactly k. By contrast, Beam Enumeration exhaustively enumerates the top k tokens at each step to extract substructures, such that the total number of sub-sequences is $k^{Beam\ Steps}$.

**Search Space:** Beam Search keeps the top k candidates globally, pruning less probable sequences. Beam Enumeration keeps the top k *local* candidates per sub-sequence.

**Termination:** Beam Search terminates when a sequence ends or a maximum length is reached. Beam enumeration does a pre-defined number of steps, controlled by *Beam Steps*.

**Use of results:** Beam Search returns the highest probability candidates for direct use (for instance, as full decoded molecules in Moret et al. (2021). Beam Enumeration uses the enumerated sub-sequences to extract molecular substructures for self-conditioned generation.

In summary, Beam Enumeration performs local, exhaustive enumeration of sub-sequences to extract meaningful substructures, while Beam Search approximates finding a set of globally optimal (by probability) sequences. The exhaustive sub-sequence enumeration enables probabilistic extraction of knowledge from the model.

