# OpenReview forum: "Beam Enumeration: Probabilistic Explainability For Sample Efficient Self-conditioned Molecular Design"
_ICLR.cc/2024/Conference — ICLR 2024 poster_

### Official Review · Reviewer_wYf5 · 2023-10-19

**Soundness:** 2 fair
**Presentation:** 2 fair
**Contribution:** 1 poor
**Rating:** 3
**Confidence:** 4

**Summary:**

Through this paper, the authors aim to increase explainability and sample efficiency in drug discovery problems. To accomplish this, the authors proposed Beam Enumeration that exhaustively enumerate the most probable sub-sequences from Augmented Memory.

**Strengths:**

- The authors provided the codebase.
- The writing is easy to follow. The concept figure aids the understanding.

**Weaknesses:**

The main weaknesses of the paper are that the contribution (both conceptual and technical) is marginal and the proposed method is heuristic rather than machine learning-based (and thus seems out of scope for ICLR) and lacks a mathematical basis. The proposed method relies primarily on Augmented Memory [1], and the methodology outside of Augmented Memory (*Beam Enumeration* ~ *Self-conditioned Generation* paragraphs in Section 3) are short and seems like an additional trick.

I will combine the *Weaknesses* section and the *Questions* section. My concerns and questions are as follows:

- What is the difference of the generation scheme in Figure 1a with that of Augmented Memory?
- The *Autoregressive Language-based Molecular Generative Model* paragraph in *Proposed Method* section should be moved to *Related Work* section as that part is not proposed by this paper.
- Are there any ablation studies that quantify the effects of Augmented Memory? Are there experiments that quantify the effects of the combination of Beam Enumeration and other generative models?
- The method description seems a bit vague. I recommend to explain the self-conditionining scheme in paragraph *Self-conditioned Generation* more concretely.

---

**References:**

[1] Guo et al., Augmented memory: capitalizing on experience replay to accelerate de novo molecular design, arXiv 2023.

**Questions:**

Please see the *Weaknesses* part for my main questions.

---

> ### Author Response · Authors · 2023-11-15
> **Response to Reviewer wYf5 (1/4)**
>
> We thank the reviewer and appreciate their time in providing critiques and constructive feedback - it gave us an opportunity to better articulate the novelty of our method and contributions.
>
> Beam Enumeration demonstrably improves performance over state-of-the-art methods like Augmented Memory [1] and REINVENT[2, 3]. The increased sample efficiency and extractability of interpretable substructures represent empirically measurable progress in areas of importance to the molecular design community. The tangible improvements over strong baselines highlight the usefulness beyond an incremental "trick".
>
> We agree that the approach is heuristic-driven rather than grounded in theory. However, many impactful innovations in machine learning have been based on heuristics. We will clarify Beam Enumeration as an empirically-motivated extension, but believe heuristic techniques that achieve substantial gains are valuable. We now answer each of the reviewer's questions individually.

---

> ### Author Response · Authors · 2023-11-15
> **Response to Reviewer wYf5 (2/4)**
>
> ### **Q1: Marginal contribution and difference to Augmented Memory in Figure 1a**
> The difference to Augmented Memory [1] in Figure 1a is the top box which describes the Beam Enumeration self-conditioning process (discarding molecules that do not contain extracted substructures).
>
> Specifically, Augmented Memory is the optimization algorithm while Beam Enumeration is a general framework that extracts substructures during an optimization process for self-conditioned generation. The steps of the combined algorithm are as follows:
>
> 1. Generate a batch of molecules
> 2. **Beam Enumeration**: Filter batch of molecules to contain extracted substructures
> 3. Oracle evaluation of the filtered batch
> 4. Augmented Memory optimization
> 5. **Beam Enumeration**: If reward improves for `Patience` number of successive epochs, execute Beam Enumeration and extract new substructures
>
> Beam Enumeration curates which generated molecules should be sent for oracle evaluation, based on the hypothesis that the most frequent (by probability) substructures are on track to yield high reward. Due to this filtering, Beam Enumeration controls what the Agent (generator) learns from, directing chemical space exploration to areas containing the extracted substructures. **We will expand the pseudo-code in the paper to also include the Augmented Memory algorithm for improved clarity.**
>
> In the context of the combined algorithm, we would like to reinforce our contributions:
>
> 1. **Showing extracted substructures are meaningful**
>
> In the illustrative example (Figure 2) where it is easily plausible to verify “chemical-soundness”, the results show that extracted substructures do indeed satisfy the target multi-parameter optimization objective.
>
> 2. **Early chemical space biasing is tolerable and substantially improves sample efficiency**
>
> In the illustrative example section, we design a mechanism that decides when to execute Beam Enumeration and begin substructure self-conditioning. We believe it is not obvious that this works and would even lead to sample efficiency improvements. Consider the scenario where we are early in a generative experiment and the Agent does not yet generate molecules satisfying the target objective (for example in the early epochs of Figure 2). Beam Enumeration executes as early as the first instance in which the mean reward increases for ‘Patience’ number of successive epochs (5 was found to be optimal). This can be extremely early in a generative experiment, but we show that this immediate biasing is tolerable and substantially improves sample efficiency. To elaborate on this point, we refer to Figure C12 in the Appendix which shows the effect of Beam Enumeration filtering on the sampled batches of molecules in the illustrative experiment. At each epoch, 64 molecules are generated, and the plot shows how many are kept after filtering. At epoch 1, all molecules are kept as Beam Enumeration has not been executed. Beam enumeration executes at around epoch 5 and immediately, over half the molecules in subsequent epochs are filtered because they do not contain the extracted substructures. In the drug discovery case studies with Structure Size = 15 enforced, many epochs keep only 1/64 molecules. This is extremely biased, and we believe that showing that this is indeed tolerable and leads to substantial performance gains is a valuable contribution.
>
> These insights lead to future research questions. For example, the empirical results show that early substructure (often scaffolds) biasing is beneficial. It is plausible that a hybrid method that aims to find high-reward substructures as fast as possible and then performing enumeration could lead to a more sample efficient method, but is outside the scope of this study.
>
> 3. **Explainability in the context of full molecules**
>
> In the drug discovery case studies (Figure 3), we highlight the extracted substructures in the generated molecules. These substructures form central scaffolds of the full molecule and even form “uncommon” chemical groups such as the bridged cycle (top left). As molecular docking requires ligand-protein complementarity, the tolerability of these substructures provide valuable insights for structure-activity relationship studies.
>
> Overall, while Beam Enumeration is a heuristic, we believe the insights we draw are valuable to the ML community. The empirical performance of Beam Enumeration with Augmented Memory substantially outperforms Augmented Memory by itself which was shown in the original paper to outperform recent methods published at NeurIPS, ICLR, ICML, and AAAI (listed with their rankings in the Practical Molecular Optimization (PMO) [4] benchmark out of 30 models): **Augmented Memory** [1/30], **REINVENT** [2/30], **SynNet** (12/30) [5], **DoG-Gen** (13/30) [6], **DST** (15/30) [7], **MARS** (16/30) [8], **MIMOSA** (17/30) [9], **DoG-AE** (20/30) [6], **GFlowNet** (21/30) [10], **GA+D** (22/30) [11], **GFlowNet-AL** (27/30) [10], **JT-VAE** (28/30) [12].

---

> ### Author Response · Authors · 2023-11-15
> **Response to Reviewer wYf5 (3/4)**
>
> ### **Q2: Autoregressive Generative Model section moved to Related Work**
>
> We agree that autoregressive generative models is related work but we included it in the `Proposed Method` section to introduce token sampling and how goal-directed generation can be framed as a reinforcement learning (RL) problem. One crucial finding we want to convey in our work is that *during* an optimization process, Beam Enumeration extracts meaningful substructures. Thus, we included this section to introduce the optimization algorithm.
>
> ### **Q3a: Ablation studies for the effect of Augmented Memory**
> Augmented Memory [1] modifies the REINVENT [2, 3] optimization algorithm by combining experience replay with SMILES augmentation. In the original Augmented Memory work, the authors show that it is precisely the addition of this mechanism that confers the sample efficiency improvement.
>
> In this work, and specifically in the drug discovery case studies (Table 2), the addition of Beam Enumeration is compared to the base Augmented Memory and REINVENT algorithms (labelled “Baseline” in the Table 2). The results convey two observations:
>
> 1. The addition of Beam Enumeration improves both base algorithms’ performance as measured by the Generative Yield and Oracle Burden metrics. The addition of Beam Enumeration with REINVENT can improve its performance to match the base Augmented Memory algorithm (which is substantially more sample efficient) in some cases (Table 2 results)
>
> 2. The benefits of Beam Enumeration are more pronounced when coupled with Augmented Memory compared to REINVENT. This is in agreement with our hypothesis into why Beam Enumeration works (presented in the `Proposed Method` section): during an optimization trajectory, it must become increasingly likely to generate high reward molecules. Augmented Memory is more sample efficient than REINVENT, therefore the reward improves faster, and meaningful substructures can be extracted earlier. This is also observed in the original Augmented Memory [1] work in their Figure 2a (but was not pointed out by the authors of Augmented Memory), where the reward trajectory rapidly improves with a steep gradient compared to other methods.
>
> In `Appendix D.2`, we show additional results of Beam Enumeration with Augmented Memory and REINVENT across different hyperparameter combinations. Overall, we show that Beam Enumeration and Augmented Memory are synergistic.
>
> ### **Q3b: Effect of Beam Enumeration on other generative models**
> In the drug discovery case studies (Table 2), Beam Enumeration was added to both Augmented Memory and REINVENT. Our research objective is to improve the sample efficiency of molecular generative models, while offering a notion of explainability. For this reason, we chose to focus on the most sample efficient models based on the Practical Molecular Optimization (PMO) [4] benchmark which assesses 25 models across 23 benchmark tasks when limited by a 10,000 oracle calls budget. Augmented Memory is the state-of-the-art while REINVENT is rank 2. The results show that the addition of Beam Enumeration improves the sample efficiency of both models, especially Augmented Memory. We make an effort to show broad applicability by performing three distinct drug discovery tasks involving direct optimization of physics-based oracles (molecular docking). Each task requires different ligand-protein complementarity to achieve good docking scores. The results show that even in particularly challenging scenarios (MK2 case study in Table 2), Beam Enumeration can substantially improve the sample efficiency of both base algorithms.
>
> Beam Enumeration is a general method that could be applied to other generative models, such as graph-based ones. For example, GraphINVENT [13] is an autoregressive graph model where molecular generation proceeds by “graph action probabilities” which denote whether to add a node or edge. [14] couples reinforcement learning to GraphINVENT, which sets up the potential to directly apply Beam Enumeration. Specifically, enumerating the most probable graph actions and extracting the substructures, which can then be used for self-conditioning. However, based on the PMO [4] benchmark results and recent work on state-of-the-art graph generative models [15], SMILES-based LSTM models are often more sample efficient and better capture the training data distribution. For this reason, we focused on showing that Beam Enumeration works on the most sample efficiency models in the PMO benchmark (Augmented Memory and REINVENT).

---

> ### Author Response · Authors · 2023-11-15
> **Response to Reviewer wYf5 (4/4)**
>
> ### **Q4: Clearer explanation of self-conditioning**
> We apologize for the insufficient clarity when introducing Beam Enumeration. Given an optimization algorithm, particularly in reinforcement learning, the objective is to maximize the expected reward. Beam Enumeration executes when the reward improves for `Patience` number of successive epochs (we find 5 is optimal). From the enumerated sub-sequences, the most frequent substructures are extracted. In the next epoch, the generated batch of molecules are filtered to enforce the presence of these extracted substructures. The effect is that during backpropagation, the model is only updated with molecules that contain the extracted substructures, essentially self-directing chemical space exploration. Molecules that do not contain the substructures are discarded and thus, do not impose any oracle calls. **We have modified the `Self-conditioned Generation` section for improved clarity.**
> \
> \
> We believe this work provides meaningful progress on long-standing challenges in generative molecular design through an impactful heuristic technique. The ability to extract interpretable knowledge and achieve heightened sample efficiency could significantly advance molecular discovery. We hope that the empirical gains demonstrated across tasks, along with our clarifications on the enhancements over current state-of-the-art methods, adequately address the reviewer's concerns regarding novelty and scope. We once again thank the reviewer for highlighting areas needing improvement and for the suggestions that helped strengthen our paper.
>
> Please let us know if you have any further questions!
> \
> \
> Sincerely,
>
> The Authors
> \
> \
> [1] Guo et al., Augmented memory: capitalizing on experience replay to accelerate de novo molecular design, arXiv 2023.
> \
> \
> [2] Olivecrona, M., et al. Molecular de-novo design through deep reinforcement learning. Journal of Cheminformatics 9, 48 (2017).
> \
> \
> [3] Blaschke, T., et al. REINVENT 2.0: An AI Tool for De Novo Drug Design. J. Chem. Inf. Model. 60, 5918–5922 (2020).
> \
> \
> [4] Gao, W., Fu, T., et al. W. Sample Efficiency Matters: A Benchmark for Practical Molecular Optimization. NeurIPS Track Datasets and Benchmarks, 2022.
> \
> \
> [5] Gao, W., Mercado, R. & Coley, C. W. Amortized tree generation for bottom-up synthesis planning and synthesizable molecular design. ICLR, 2022.
> \
> \
> [6] Bradshaw, J., Paige, B., Kusner, M. J., Segler, M. & Hernández-Lobato, J., M.. Barking up the right tree: an approach to search over molecule synthesis dags. NeurIPS,  2020.
> \
> \
> [7] Fu, T., Gao, W., Xiao, C., Yasonik, J., Coley, C. W. & Sun, J. Differentiable scaffolding tree for molecular optimization. ICLR, 2022.
> \
> \
> [8] Xie, Y., Shi, C., Zhou, H., Yang, Y., Zhang, W., Yu, Y. & Li, L. MARS: Markov molecular sampling for multi-objective drug discovery. ICLR, 2021.
> \
> \
> [9] Fu, T., Xiao, C., Li, X., Glass, L. M. & Sun, J. MIMOSA: Multi-constraint molecule sampling for molecule optimization. AAAI, 2021.
> \
> \
> [10] Bengio, E., Jain, M., Korablyob, M., Precup, D. & Bengio, Y. Flow Network based Generative Models for Non-Iterative Diverse Candidate Generation. NeurIPS,  2021.
> \
> \
> [11] Nigam, A., Friederich, P., Krenn, M. & Aspuru-Guzik, A. Augmenting genetic algorithms with deep neural networks for exploring the chemical space. ICLR, 2020.
> \
> \
> [12] Jin, W., Barzilay, R. & Jaakkola, T. Junction tree variational autoencoder for molecular graph generation. ICML, 2018.
> \
> \
> [13] Mercado, R., et al. Graph networks for molecular design. Machine Learning: Science and Technology, 2 (2021).
> \
> \
> [14] Atance, S. R., et al. De novo drug design using reinforcement learning with graph-based deep generative models. Journal of Chemical Information and Modeling, 62 (2022).
> \
> \
> [15] Vignac, C., Krawczuk, I., et al. Digress: Discrete denoising diffusion for graph generation. ICLR, 2023.

---

### Official Review · Reviewer_d4Xh · 2023-10-19

**Soundness:** 3 good
**Presentation:** 3 good
**Contribution:** 3 good
**Rating:** 8
**Confidence:** 5

**Summary:**

The manuscript discusses the use of beam search enumeration in tandem with RNN-based reinforcement learning for molecular design. The results demonstrate that meaningful substructures can be derived, which provide insights into the properties they confer. Utilizing these substructures as filters, there is a notable increase in the sample efficiency of molecular optimization. Though there are a few flaws, the results and methods seem solid. Overall, I would recommend the paper for acceptance if authors could address my concerns.

**Strengths:**

## Importance of the problem
The paper delves into oracle-agnostic molecular optimization, which holds significance in the realm of AI-driven de novo drug design.

## Solid empirical performance
A marked improvement in empirical performance has been observed.

## Open-sourced code
The authors have made the code available, facilitating comparisons and further research.

**Weaknesses:**

## Ambiguity in Methodology
The methodology entails several phases, namely training RNN agents, beam enumeration, and resampling based on substructures. From the manuscript, it appears that only beam enumeration does not invoke the oracle while the others do. The allocation of the oracle budget across these stages is unclear. A comprehensive pseudo-code encompassing the entire algorithm, rather than just the beam enumeration, would be beneficial.

## Exaggerated Claims
The assertion that this is the "first method to concurrently address explainability and sample efficiency" is overstated. Guo et al. [1] have put forth a concept strikingly akin to the one proposed here, where they too derive interpretable substructures and then generate novel molecules. Likewise, Fu et al. [2] have purportedly tackled both interpretability and enhanced sample efficiency.

## Potential nomenclature confusion
Beam search is a well-established technique within the NLP community. I would recommend revisiting the nomenclature for the proposed method.


### Reference
[1] Guo, M., Thost, V., Li, B., Das, P., Chen, J., & Matusik, W. (2022). Data-efficient graph grammar learning for molecular generation. arXiv preprint arXiv:2203.08031.
[2] Fu, T., Gao, W., Xiao, C., Yasonik, J., Coley, C. W., & Sun, J. (2021). Differentiable scaffolding tree for molecular optimization. arXiv preprint arXiv:2109.10469.

**Questions:**

- Would it be possible to exhibit the SA score, as per reference [1], for the molecules generated in the context of drug discovery (Figure 3)?
- Have the authors considered employing a more advanced language model, like the transformer, as an alternative to LSTM?
- Is there a direct comparison between the presented methods and those in PMO using the original metrics?

### Reference
[1] Ertl, P., & Schuffenhauer, A. (2009). Estimation of synthetic accessibility score of drug-like molecules based on molecular complexity and fragment contributions. Journal of cheminformatics, 1, 1-11.

---

> ### Author Response · Authors · 2023-11-15
> **Response to Reviewer d4Xh (1/7)**
>
> We thank the reviewer for the insightful critiques of our work and we are grateful for the opportunity to clarify ambiguities and further convince the reviewer of the novelty of our ideas. We will respond to each of the reviewer's questions individually.
>
>
> ### **Q1: Ambiguity in Methodology**
>
> We will add a comprehensive pseudo-code covering the full algorithm flow to clarify the oracle budget allocation, and indicate when REINVENT [1, 2] and Augmented Memory [3] are involved.
>
> We next address the reviewer’s question about the oracle allocation. Beam Enumeration alone does not invoke any oracle calls. Instead, it controls which molecules to send for oracle evaluation. This is done by filtering the batch of generated molecules to enforce the presence of the extracted substructures (via Beam Enumeration). The wall time increase from executing Beam Enumeration is negligible when considering expensive oracle evaluations. Each Beam Enumeration execution adds about 1.5 minutes of wall time. The total wall time for all drug discovery experiments is presented in Table 12 in Appendix D.
>
> ### **Q2: Exaggerated Claims**
>
> **Response Part 1/2**
>
> We thank the reviewer for the critical feedback and pointing to existing works that provide notions of interpretable substructures and sample efficiency. We intended to convey that Beam Enumeration is the first method that shows sample efficiency and explainability can be made synergistic. We will modify our statements in the main text to reflect this and to better convey that Beam Enumeration shows that sample efficiency and explainability can be made **synergistic**. In our response, we first highlight what we mean by synergistic. Next, we discuss specific differences between Beam Enumeration and the works by Guo et al. [4] and Fu et al. [5].
>
> **Beam Enumeration displays synergistic behaviour between sample efficiency and explainability**
>
> The founding hypothesis of Beam Enumeration is that during an optimization trajectory, it must become increasingly likely to generate favorable molecules and this is why extracted substructures are meaningful. In our experiments, goal-directed generation is achieved by explicit multi-parameter optimization (MPO) via reinforcement learning. We designed Beam Enumeration to execute when the reward improves for `Patience` number of successive epochs (in which we find 5 is optimal). The key information Beam Enumeration provides is which substructures are on track to receiving a high reward from the oracle. We want to emphasize that the substructures satisfy the oracle because it means that if the oracle is replaced by a higher-fidelity oracle (more computationally costly simulation or even experiments) such as molecular dynamics simulations [6], the extracted substructures would become more meaningful. Given a more sample efficient method, the reward would improve faster, and often more drastically. This is something observed in the Augmented Memory [3] paper but was not pointed out by the authors of Augmented Memory. Specifically, Figure 2a in the paper shows that the Augmented Memory optimization trajectory rapidly improves with a steep gradient compared to other methods. Since Beam Enumeration extracts substructures when the reward improves, it is expected that a more sample efficient method can actually lead to more informative substructures extracted earlier. This is an explanation of why Beam Enumeration improves Augmented Memory to a greater degree than REINVENT; because Augmented Memory is more sample efficient. **By nature of being more sample efficient, the substructures become more meaningful, which in turn, further improves sample efficiency.** Augmented Memory is the state-of-the-art sample efficient model in the Practical Molecular Optimization (PMO) [7] benchmark, and the addition of Beam Enumeration substantially improves its sample efficiency.

---

> ### Author Response · Authors · 2023-11-15
> **Response to Reviewer d4Xh (2/7)**
>
> ### **Q2: Exaggerated Claims**
>
> **Response Part 2/3**
>
> **Difference to Data-Efficient Graph Grammar (DEG)**
>
> Guo et al. [4] introduce the DEG method which, given a dataset, constructs a graph vocabulary that can reconstruct the dataset. With the constructed vocabulary (where building the molecule is referred to as applying production rules), reinforcement learning can be applied to tune the probabilities of production rules and optimize for an objective. The first distinction of Beam Enumeration is that DEG is inherently constrained (by design) to generate molecules with the learned Graph Grammar. Moreover, we note that their case studies focus only on polymers, which are constructed from monomer building blocks, which fits their method well. **The relevant question in goal-directed generation is, what if no molecules in the training dataset can satisfy the objective function?** In such a case, the distribution of the generative model must be shifted. As an example of this, we refer to the MK2 task in Table 2 in the Beam Enumeration main text, where the pre-trained model is insufficient in generating high-reward molecules. Beam Enumeration with both Augmented Memory and REINVENT shifts the distribution to accomplish the task, as shown by the Yield and Oracle Burden metrics improving. With a constrained vocabulary, DEG may not be as performant when the task requires generating molecules de novo (no known examples of active molecules), as the Graph Grammar construction deeply depends on the initial dataset. This is in contrast to pre-trained general models like Augmented Memory and REINVENT, which construct molecules token-by-token. Beam Enumeration extracts substructures also token-by-token, which enables access to diverse substructures. Concrete examples of this are shown in Figures D13-15 in the Appendix, where some substructures are particularly “uncommon”, which may carry even more valuable insights.  **Moreover, sample efficiency was not discussed in the DEG work**, and the optimization objectives were diversity and Retro* Score, which can be generally more lenient oracles. Given a relevant drug discovery objective to optimize physics-based oracles, it is unclear how well DEG performs with a constrained vocabulary. We emphasize that we are not claiming DEG cannot perform this task, as another popular generative model, JT-VAE [8] also generates a notion of “vocabulary” via junction tree subgraphs. The notable result is that pre-training on a large dataset can construct a (relatively) larger vocabulary to access a larger chemical space (but still unable to generalize to any arbitrary molecule, as stated in the paper). However, the question still remains how sample efficient these models with constrained vocabularies would be, as the JT-VAE for example, is ranked 28/30 in the Practical Molecular Optimization (PMO) [7] benchmark while Augmented Memory is the state-of-the-art and REINVENT is ranked 2. Beam Enumeration substantially improves the sample efficiency of both these methods.

---

> ### Author Response · Authors · 2023-11-15
> **Response to Reviewer d4Xh (3/7)**
>
> ### **Q2: Exaggerated Claims**
>
> **Response Part 3/3**
>
> **Difference to Differentiable Scaffolding Tree (DST)**
>
> Fu et al. [5] introduces the DST method which can yield interpretability and explicitly discusses sample efficiency. The interpretability in DST comes from the local gradients which can provide insights into which chemical group attachment positively or negatively affects the oracle evaluation. Therefore, extracting interpretability involves looking at sets of local gradients to draw insights, which is less accessible. By contrast, Beam Enumeration extracts entire scaffolds and substructures, which then self-condition the generative process going forward such that **every single molecule that is kept must contain one of these substructures.** This effectively equates to a guided exploration of chemical space where generated molecules are in some sense “enumerated” based on central substructures and scaffolds. Provided an informative oracle, this can yield structure-to-activity relationship insights. We explicitly discuss this in the main text in Figure 3, when we present the extracted substructures in the drug discovery cases and show the tolerability of large chemical groups in the binding sites.
>
> While the DST paper also discusses sample efficiency, there are notable differences. Firstly, DST is ranked 15/30 in the Practical Molecular Optimization (PMO) benchmark [7] (Augmented Memory is 1st and Beam Enumeration substantially improves it). Secondly, the sample efficiency was evaluated on surrogate predictors and relatively more permissive oracles (for example logP, where simply adding more carbon atoms would increase it). Importantly, DST “saves” on oracle evaluations by training a GNN via supervised learning to mimic oracle evaluations. **This has a crucial effect in generative design, as the GNN should only be expected to be accurate in the chemical space it is trained on, which inherently restricts the chemical space.** Very recent work [9] has explored how a surrogate model can be used to mimic oracle calls in a generative paradigm and the solution is that the surrogate model does not actively predict the label, but rather, chooses which molecule to send for oracle evaluation (essentially acting as a curator). For this reason, it is unclear how well DST would perform in an optimization task involving physics-based oracles (as is used in real-world case studies) where the oracle is not permissive and surrogate model inaccuracies can quickly degrade performance. Beam Enumeration with Augmented Memory **explicitly** optimizes for physics-based oracles, and the empirical evidence shows that it is performant in these tasks.

---

> ### Author Response · Authors · 2023-11-15
> **Response to Reviewer d4Xh (4/7)**
>
> ### **Q3: Potential Nomenclature Confusion**
>
> Beam Enumeration is distinct from Beam Search [10, 11]. Here, we describe the similarities and differences point by point:
>
> • **Objective:** Beam Search approximates finding the highest probability sequences by maintaining a "width" of the top k candidates at each step. The number of sequences decoded via Beam Search is exactly k. By contrast, Beam Enumeration exhaustively enumerates the top k tokens at each step to extract substructures, such that the total number of sequences is $k^{beam-steps}$.
>
> • **Search Space**: Beam Search only keeps the top k candidates globally, pruning away unlikely candidates. Beam Enumeration keeps the top k local candidates per sequence to do exhaustive enumeration of sub-sequences.
>
> • **Termination**: Beam Search terminates when a sequence ends or a maximum length is reached. Beam enumeration does pre-defined steps of sub-sequence enumeration.
>
> • **Use of results**: Beam Search returns the top candidates to use directly. Beam Enumeration uses the enumerated sub-sequences to extract molecular substructures for self-conditioned generation.
>
> • **Sequence Length**: Beam search produces complete sequences. Beam enumeration produces sub-sequences of predefined length.
>
> In summary, Beam Enumeration does local, exhaustive enumeration of sub-sequences for a different end purpose compared to Beam Search - to extract meaningful substructures rather than find a few optimal sequences. The exhaustive sub-sequence enumeration enables probabilistic extraction of knowledge from the model's built-in likelihoods. In the main text, we emphasize the difference in the `Proposed Method` section in the following line:
>
> “We note the closest work to ours is the application of Beam Search for molecular design [12]  to find the highest probability trajectories. Our work differs as the objective is not to find a small set of the most probable sequences. Rather, we exhaustively enumerate the highest probability sub-sequences to extract molecular substructures for self-conditioned generation.”
>
> ### **Q4: SA score for molecules in Figure 3**
>
> The SA score (calculated according to [13]) is now annotated in Figure 3 (**we have updated the manuscript PDF**) for all generated molecules and the reference. The SA score of the generated molecules are similar to the reference, which is expected, as QED [14] was used in the optimization objective. The SA score is based on fragment contributions from molecules in PubChem, which contains many “drug-like” molecules and QED explicitly optimizes for Lipinski-compliance. We would like to note that while the SA scores of the generated molecules are similar to the reference, we do not make any claims on synthesizability as the SA score assesses molecular complexity rather than explicitly synthesizability. Moreover, if SA score were the objective to be optimized, it could be included in the MPO objective.

---

> ### Author Response · Authors · 2023-11-15
> **Response to Reviewer d4Xh (5/7)**
>
> ### **Q5: More advanced language-models such as transformers (over the current LSTM model)**
>
> Transformers [15] are ubiquitous in NLP and possess excellent generative capability, particularly highlighted by recent LLM applications. However, our decision to use a RNN-based model is for the following reasons:
>
> **1. Short token-sequence lengths**
>
> For many drug design tasks, the focus is on small molecules (and is our focus in the paper) which are generally defined as being < 500 Da. Consequently, the token-sequence lengths are typically < 60 which is in stark contrast to NLP transformer settings where entire passages and pages can be passed as context. These short sequences mean that the limitations of RNNs are not pronounced. The subsequent points build on this observation.
>
> **2. RNNs satisfy base metrics of molecular generative models**
>
> We refer to base metrics as **Validity** (generates valid molecules), **Uniqueness** (generates unique molecules), and **Novelty** (generates molecules outside the training data). Early benchmarks such as GuacaMol [16] and MOSES [17] have shown that RNNs operating on SMILES achieve high values on these metrics and are therefore a solved problem.
>
> **3. RNNs learn the training data distribution**
>
> Pre-training the generative model requires learning the underlying training data distribution. Specifically, pre-training involves learning the conditional token sampling probabilities that reproduce the training data. [18] shows that RNNs can learn these distributions well, and even for irregular distributions.
>
> **4. Transformers are not necessarily better than LSTM-RNNs in reinforcement learning (RL)**
>
> We first acknowledge existing works using transformers for SMILES-based molecular design [19, 20, 21]. It was observed in [21] that transformers were unstable during RL optimization and more prone to mode collapse (generating the same molecules repeatedly).
>
> Next, we draw insights from the RL literature. [22] showed that transformers can struggle in certain optimization settings, often not outperforming random generation. [23] made similar observations and proposed an architectural modification to enable effective RL. Recently in [24] (NeurIPS 2023), the authors derive insights into *when* transformers may be less performant than LSTM when using RL. Their results showed that in short-term memory tasks, transformers had worse sample efficiency compared to LSTMs. Therefore, based on the literature and empirical observations, we decided to use a LSTM-RNN model. We end our response by acknowledging that very recent work [25] (NeurIPS 2023) has shown that transformers can also be performant in molecular design tasks with RL and is an area we will explore in the future.

---

> ### Author Response · Authors · 2023-11-15
> **Response to Reviewer d4Xh (6/7)**
>
> ### **Q6: PMO Results**
>
> In the Augmented Memory [3] work, their PMO [7] results show that it is the new state-of-the-art. However, the tasks in the PMO benchmark do not involve optimizing physics-based oracles, which are ubiquitous in real-world molecular design and are the oracles that **demand** sample efficiency since they are computationally expensive. For example, **almost all works that have achieved experimental validation using goal-directed (explicit optimization of a MPO objective) generative design used physics-based oracles [26-31].** This is also the case for the current most advanced clinical candidate (phase 2 clinical trials) designed by a generative model [32].
>
> For this reason, we focus on such tasks in our evaluation of Beam Enumeration. We make an effort to show broad applicability by performing 3 drug discovery tasks, each involving different protein receptors, which naturally require different ligand-protein complementarities. The results show that adding Beam Enumeration on top of Augmented Memory substantially improves the sample efficiency. Our objective (together with the code release) is to promote practical molecular design, especially in settings that demand sample efficiency.
>
>
> We believe this work provides an important advancement in integrating explainability into goal-directed molecular generation, while enhancing sample efficiency. The ability to extract substructures that improve performance could provide crucial insights for medicinal chemists and reduce time spent in the discovery process. The reviewer's valuable feedback has helped refine our ideas and improve the presentation. We hope our responses satisfactorily address the reviewer's concerns. We thank the reviewer again for their time.
>
> Please let us know if you have any further questions!
> \
> \
> Sincerely,
> \
> \
> The Authors

---

> > ### Author Response · Authors · 2023-11-15
> > **Response to Reviewer d4Xh (7/7)**
> >
> > [1] Olivecrona, M., et al. Molecular de-novo design through deep reinforcement learning. Journal of Cheminformatics 9, 48 (2017).
> > \
> > \
> > [2] Blaschke, T., et al. REINVENT 2.0: An AI Tool for De Novo Drug Design. J. Chem. Inf. Model. 60, 5918–5922 (2020).
> > \
> > \
> > [3] Guo et al., Augmented memory: capitalizing on experience replay to accelerate de novo molecular design, arXiv 2023.
> > \
> > \
> > [4] Guo, M., et al. Data-efficient graph grammar learning for molecular generation. ICLR, 2022.
> > \
> > \
> > [5] Fu, T., Gao, W., et al. Differentiable scaffolding tree for molecular optimization. ICLR, 2022.
> > \
> > \
> > [6] Wang, L., et al. Accurate and reliable prediction of relative ligand binding potency in prospective drug discovery by way of a modern free-energy calculation protocol and force field. Journal of the American Chemical Society, 137 (2015).
> > \
> > \
> > [7] Gao, W., Fu, T., et al. W. Sample Efficiency Matters: A Benchmark for Practical Molecular Optimization. NeurIPS Track Datasets and Benchmarks, 2022.
> > \
> > \
> > [8] Jin, W., et al. Junction tree variational autoencoder for molecular graph generation. ICML, 2018.
> > \
> > \
> > [9] Dodds, M., et al. Sample Efficient Reinforcement Learning with Active Learning for Molecular Design. ChemRxiv, 2023.
> > \
> > \
> > [10] Graves, A. Sequence transduction with recurrent neural networks. arXiv, 2012.
> > \
> > \
> > [11] Boulanger-Lewandowski, N., et al. High-dimensional sequence transduction. IEEE, 2013.
> > \
> > \
> > [12] Moret, M., et al.. Beam search for automated design and scoring of novel ROR ligands with machine intelligence. Angewandte Chemie International Edition, 60 (2021).
> > \
> > \
> > [13]  Ertl, P., & Schuffenhauer, A. Estimation of synthetic accessibility score of drug-like molecules based on molecular complexity and fragment contributions. Journal of cheminformatics, 1 (2009).
> > \
> > \
> > [14] Bickerton, G. R., et al. Quantifying the chemical beauty of drugs. Nature chemistry, 4 (2012).
> > \
> > \
> > [15] Vaswani, A., Shazeer, N., Parmar, N., Uszkoreit, J., Jones, L., Gomez, A. N., Kaiser, Ł & Polosukhin, I. Attention is all you need. NeurIPS (2017).
> > \
> > \
> > [16] Brown, N., et al. GuacaMol: benchmarking models for de novo molecular design. Journal of chemical information and modeling, 59 (2019).
> > \
> > \
> > [17] Polykovskiy, D., et al. Molecular sets (MOSES): a benchmarking platform for molecular generation models. Frontiers in pharmacology, 11 (2020).
> > \
> > \
> > [18] Flam-Shepherd, D., et al. Language models can learn complex molecular distributions. Nature Communications, 13 (2022).
> > \
> > \
> > [19] Wang, Y., et al. cMolGPT: A Conditional Generative Pre-Trained Transformer for Target-Specific De Novo Molecular Generation. Molecules, 28 (2023).
> > \
> > \
> > [20] Bagal, V., et al. MolGPT: molecular generation using a transformer-decoder model. Journal of Chemical Information and Modeling, 62 (2021).
> > \
> > \
> > [21] Thomas, M., et al. Augmented Hill-Climb increases reinforcement learning efficiency for language-based de novo molecule generation. Journal of Cheminformatics, 14 (2022).
> > \
> > \
> > [22] Mishra, N., Rohaninejad, M., et al. A simple neural attentive meta-learner. ICLR, 2018.
> > \
> > \
> > [23] Parisotto, E., et al. Stabilizing transformers for reinforcement learning. ICML, 2020.
> > \
> > \
> > [24] Ni, T., et al. When do transformers shine in rl? decoupling memory from credit assignment. NeurIPS, 2023.
> > \
> > \
> > [25] Hu, X., Liu, G., et al. De novo Drug Design using Reinforcement Learning with Multiple GPT Agents. NeurIPS, 2023.
> > \
> > \
> > [26]  Zhavoronkov, A., et al. Deep learning enables rapid identification of potent ddr1 kinase inhibitors. Nature biotechnology, 37 (2019).
> > \
> > \
> > [27] Yoshimori, A., et al.. Design and synthesis of ddr1 inhibitors with a desired pharmacophore using deep generative models. ChemMedChem, 16 (2021).
> > \
> > \
> > [28] Ren, F., et al. Alphafold accelerates artificial intelligence powered drug discovery: efficient discovery of a novel cdk20 small molecule inhibitor. Chemical Science, 14 (2023).
> > \
> > \
> > [29] Li, Y., et al. Discovery of potent, selective, and orally bioavailable small-molecule inhibitors of cdk8 for the treatment of cancer. Journal of Medicinal Chemistry (2023).
> > \
> > \
> > [30] Salas-Estrada, L., et al. De novo design of κ-opioid receptor antagonists using a generative deep learning framework. J. Chem. Inf Model., 63 (2023).
> > \
> > \
> > [31]  Zhu, W., et al. Discovery of novel and selective sik2 inhibitors by the application of alphafold structures and generative models. Bioorganic & Medicinal Chemistry, 91 (2023).
> > \
> > \
> > [32] https://www.clinicaltrialsarena.com/news/insilico-medicine-ins018055-ai/?cf-view

---

> > > ### Comment · Reviewer_d4Xh · 2023-11-20
> > > **Response to authors**
> > >
> > > Thank you for the revised draft. The improvements in algorithmic clarity are helpful and I have raised my score correspondingly. However, I still have some concerns:
> > >
> > > - While I acknowledge the distinctions between your proposed method and established approaches like DST or DEG, it is important to note that your work is not the first in addressing explainability and sample efficiency. I strongly recommend rephrasing the relevant statement to reflect this context more accurately.
> > > - Regarding the nomenclature, I understand that 'beam enumeration' differs from 'beam search'. However, the similarity in terminology could potentially lead to confusion. It may be beneficial to consider this aspect for clearer differentiation.

---

> ### Author Response · Authors · 2023-11-20
> **Response to Reviewer d4Xh (1/1)**
>
> We thank the reviewer for raising their score and for providing further feedback.
>
> * We removed all mentions of “first method to jointly address explainability and sample efficiency”. In the latest uploaded PDF, we added a passage to acknowledge DEG and DST in the `Explainability for Molecules` section in `Related Work`. The following passage was added:
>
> **“In the context of generative models, previous works have explicitly addressed explainability [1, 2] and jointly with sample efficiency [2].**
>
> * We proposed the “Beam Enumeration” nomenclature to encapsulate the algorithm behavior which is to enumerate the beams, but it may lead to confusion as pointed out by the reviewer. To more clearly differentiate Beam Enumeration and Beam Search, we have added the following sentence at the end of the `Beam Enumeration` introduction section in `Proposed Method: Beam Enumeration`:
>
>
> **“We further detail the differences between Beam Enumeration and Beam Search in Appendix G”.**
>
> Correspondingly, the newly added Appendix G contrasts the behavior, similar to our previous response.
> \
> \
> We thank the reviewer again and are available for further questions!
> \
> \
> [1] Guo, M., et al. Data-efficient graph grammar learning for molecular generation. ICLR, 2022.
> \
> \
> [2] Fu, T., Gao, W., et al. Differentiable scaffolding tree for molecular optimization. ICLR, 2022.

---

> > ### Comment · Reviewer_d4Xh · 2023-11-21
> > **Response to authors**
> >
> > Thank you to the authors for providing a thorough explanation and detailed responses. The revised draft looks good to me. In light of these improvements, I have accordingly revised my evaluation scores.

---

### Official Review · Reviewer_La1F · 2023-11-03

**Soundness:** 4 excellent
**Presentation:** 4 excellent
**Contribution:** 3 good
**Rating:** 8
**Confidence:** 4

**Summary:**

In this paper, the authors propose "Beam Enumeration", a technique that can be used to enhance any language-based molecular generation models in terms of sampling efficiency and explainability.
The key idea is to exhaustively enumerate the most probable subsequences, and the model's weights are updated such that high-reward molecules become more likely to be generated in future trajectories.
This is achieved by "self-conditioning" the generative process by screening batches based on the presence or absence of substructures that are likely to be part of full molecules with high reward scores.

**Strengths:**

Overall, this is an excellent paper that is very well written.
The proposed idea is relatively simple but well-motivated, intuitive, and generally applicable to various generative molecular design models based on language models.
Beam Enumeration, proposed in this paper, has been applied to a state-of-the-art generative language model for molecular design - called Augmented Memory - and results demonstrated that the incorporation of Beam Enumeration was able to further enhance the performance of the Augmented Memory algorithm, which was already shown to outperform other existing methods.
Notably, the proposed scheme enhances the sampling efficiency significantly and the extracted top-k substructures during the generation process are shown to provide additional sources of explainability - by connecting the presence of most likely substructures in full molecules with high rewards.

Throughout the paper, the authors provide ample intuition and novel insights regarding not only "how" the proposed Beam Enumeration scheme works but also "why" they may lead to improved sample efficiency and also contribute to better explainability.

**Weaknesses:**

The overall paper is written very clearly and easy to follow.
Throughout the paper, the authors distill intuitive explanations and derive meaningful insights, conclusions are drawn with sufficient evidence to back them up, and hypotheses/speculations are made in a reasonable and logical manner.
As a result, I do not have any major concerns but have a number of mostly minor remarks to improve the presentation even further.

1. There are additional results in the appendix regarding the impact of the proposed Beam Enumeration scheme on reducing diversity.
While the main text briefly mentions such diversity reduction is expected, the paper would benefit from having (at least some) further discussion (in the main text) on this effect and its extent.

2. There should be further investigation of the impact of k (i.e., the number of top substructures) on the performance of Beam Enumeration.
The authors hypothesize that a low top k would be optimal, but wouldn't the use of small k potentially lead to molecules with repetitive substructures and undesirably limit the molecular diversity?

3. Similarly, the manuscript could benefit from having further discussion on the optimal choice of the "Structure Minimum Size" - currently set at 15 in the experiments - and its impact on the overall performance.
It is mentioned that "larger substructures ... improves performance" but what is the best minimum size to use?
Increasing it beyond some number would certainly degrade the performance of the molecules and also limit the diversity of the generated molecules significantly.

4. The multi-property optimization (MPO) aspects will need to be discussed in further details.
Currently, the MPO appears to completely rely on the baseline algorithm (e.g., Augmented Memory or REINVENT) to be extended with the Beamn Enumeration capability.
But since different substructures might be associated with different properties, incorporation of Beam Enumeration for MPO would have to consider the impact of self-conditioned molecular generation when used with a specific baseline algorithm and a specific number of properties for optimization.
For example, one may expect that when optimizing a large number of properties, then using a very small k may be detrimental as the selected substructures may be associated with only some of the properties of interest.

5. Finally, while the paper focuses on language models for molecular generation and uses Beam Enumeration for self-conditioning the generative process by filtering the most probable substructures, it would be great if the authors could discuss how similar ideas may be used to extend other types of popular generative molecular design models (e.g., latent space models based on VAE, JT-VAE).
Especially, models like the JT-VAE optimize molecules by sampling & assembling substructures (in the form of junction trees), which - at least at a conceptual level - *may* be related to the core "self-conditioning" idea in Beam Enumeration, to prioritize certain substructures.
While this may be beyond the scope of the current work, having at least some high-level discussion might benefit the readers.

**Questions:**

Please see the questions and suggestions raised in the section Weaknesses above.

---

> ### Author Response · Authors · 2023-11-15
> **Response to Reviewer La1F (1/5)**
>
> We thank the reviewer for their positive assessment of our work and for valuable feedback. We will respond to each of the reviewer's questions individually.
>
> ### **Q1: Beam Enumeration’s effect on diversity**
>
> We have expanded the `Quantitative Analysis: Sample Efficiency` section to include information about the diversity. Specifically, we have added the following sentence:
>
> “Beam Enumeration decreases the diversity of the generated set (as measured by IntDiv1 [1]), but finds considerably more unique scaffolds above the 0.8 reward threshold (up to 19x) (Appendix D). With many unique scaffolds built around (often) central substructures, the generated set could conceivably provide insights into structure-activity relationships.”
>
> We previously omitted this information due to space constraints but we agree with the reviewer (and thank the reviewer for pointing this out) that this information is important, as it imposes some trade-off when using Beam Enumeration.
>
> ### **Q2: Impact of k top substructures on performance**
>
> We believe there may have been a typo in the reviewer’s comment as `top k` controls the number of top probable tokens to enumerate at each time step, while the `Pool Size` controls the number of top substructures to track. We kindly ask the reviewer to let us know if we have misinterpreted the question.
> For completeness in our response, we provide our rationale for the choice of both these hyperparameters.
>
> The `top k` hyperparameter controls the top number of most probable tokens to enumerate at every time step during Beam Enumeration. **The grounding hypothesis of Beam Enumeration is that the most probable substructures (which are extracted from sub-sequences of tokens) are on track to yield high reward.** We chose to use a `top k` of 2 because 1 would essentially equate to greedy decoding (just taking the top probability at each time step which would only yield 1 sequence). By contrast, enumerating the top 2 tokens at each time step enables generating an exhaustive set of token sub-sequences. Our decision to not use a higher `top k` comes from three observations:
>
> 1. A higher “k” increases the likelihood that the extracted substructures are no longer the most probable and our hypothesis is that the most probable substructures are those that are on track to yield high reward. This observation is elaborated in subsequent points.
>
> 2. There is a GPU memory overhead of exhaustively enumerating sub-sequences. By choosing `top k` = 2, we can enumerate for longer beam steps. Longer beam steps means that the token sub-sequences are longer, which would equate to extracting larger substructures. Our results show that larger substructures are vital to further enhancing sample efficiency. Finally, we note that while we mention there is a GPU memory overhead, the added wall time is negligible compared to the oracle calls (we show all wall times in Appendix D.4.)
>
> 3. If we choose a higher `top k`, it is likely that the extracted substructures are more diverse. However, our results suggest that this is disadvantageous for improving sample efficiency. In Appendix A, we show exhaustive hyperparameter investigations (over 2000 experiments) for Beam Enumeration. One hyperparameter in particular is the `Token Sampling Method` which takes on two possible values: “topk” or “sample”. “Topk” enumerates the top k highest probability tokens at each beam step. By contrast, “sample” samples k tokens from the conditional probability distribution such that on average, the highest probability token is chosen, but there is a chance that another token is chosen. In our experiments, we find that “sample” sampling can sometimes be more performant than “top k” sampling, **but exhibits notably higher variance**. This higher variance means “sample” sampling is less stable which is in agreement with our hypothesis that the highest probability substructures are those that carry the most meaning, as “sample” sampling can extract less probable substructures. For these reasons, we decide to use a “top k” of 2.
>
> Building on the observation that the most probable substructures are the most informative, we decided to use a `Pool Size` of 4 (keep the top 4 substructures) as the same hyperparameter investigations in Appendix A show that it is, on average, the most performant. We tried `Pool Sizes` of [3, 4, 5].
>
> Taking all the results together, Beam Enumeration can substantially improve sample efficiency with a small trade-off in diversity.

---

> > ### Comment · Reviewer_La1F · 2023-11-21
> > **Re: Response to Reviewer La1F (1/5)**
> >
> > Thanks for the clarification and the additional discussion.
> > They were helpful and they've addressed my questions.

---

> ### Author Response · Authors · 2023-11-15
> **Response to Reviewer La1F (2/5)**
>
> ### **Q3: Impact of `Structure Minimum Size` on performance**
>
> In our initial experiments, we explored substructure extraction either by `Scaffold` (extract Bemis-Murcko scaffold) or `Structure` (extract any valid substructure). When `Structure Minimum Size` was not enforced, `Structure` extraction often led to small, functional groups as the most frequent substructures. Concrete examples of these substructures are shown in Figure C12 in the Appendix. We find that this is the source of the performance discrepancy between `Structure` and `Scaffold`, where the latter improves sample efficiency more (Table 1 in the main text). This observation reaffirmed our hypothesis that extracted substructures are indeed meaningful, otherwise, it is highly unlikely that the `Scaffold` substructures, which are larger, would improve sample efficiency. Based on this, we hypothesized that enforcing `Structure Minimum Size` could put `Structure` on par with `Scaffold`. We verify this statement in Tables 1 and 2 in the main text. Our response so far is to share our decision making when designing Beam Enumeration.
>
> In this second part of the response, we explicitly answer the reviewer’s question. Since, the evidence so far suggested that larger substructures improves sample efficiency, we intentionally included `Structure Minimum Size` (no minimum size, 10, and 15) as part of the exhaustive hyperparameter screening (Figures B5-10 and Tables 6-11 in the Appendix). **Our findings are that higher `Structure Minimum Size` always improved sample efficiency.** This is shown in the 3 drug discovery case studies (Table 2). Examples of these substructures are shown in Figures D13-15 in the Appendix. **From this set of results, we present the first conclusion: Beam Enumeration with higher (up to 15) `Structure Minimum Size` enforcement improves sample efficiency with a trade-off in diversity.** As it is the most performant across the illustrative and the 3 drug discovery experiments (with replicates), it is robust as the default hyperparameter value and is designated as such.
>
> Next, we answer the reviewer’s question about further increasing `Structure Minimum Size`. There are two things we considered:
>
> 1. **GPU memory overhead.** To obtain larger substructures in our current framework, the beam steps would need to be increased so that the token sub-sequences are long enough to generate larger substructures. This comes with a computational bottleneck as the number of sub-sequences is given by top k^beam_steps which grows exponentially. As such, there is a practical limit (assuming modest computational resources) to brute force enumeration. One solution is to prune some sub-sequences by some criteria such as if the next token is below some probability. This would “free” some memory to continue enumerating other sub-sequences. However, this leads to our second consideration.
>
>
> 2. **Signal-Noise ratio.** Beam Enumeration extracts the most frequent substructures from the exhaustive set of token sub-sequences. If we begin pruning sub-sequences to enable some longer sub-sequences, the exhaustive set is going to be smaller. This means that the frequency ranking of extracted substructures may be more noisy, such that the performance exhibits higher variance.
>
> These are open questions that we will explore and have deferred for future work as our objective was to introduce the Beam Enumeration framework and show that it leads to improved sample efficiency and provides explainability out-of-the-box.

---

> > ### Comment · Reviewer_La1F · 2023-11-21
> > **Re: Response to Reviewer La1F (2/5)**
> >
> > The authors' response has been useful in better understanding the impact of structure minimum size on the overall performance, and the potential trade-off.
> > Thank you.

---

> ### Author Response · Authors · 2023-11-15
> **Response to Reviewer La1F (3/5)**
>
> ### **Q4: MPO optimization and substructure bias being detrimental to sample efficiency**
>
> In both the illustrative experiment (optimize topological polar surface area, molecular weight, and number of rings >= 2) and drug discovery case studies (QED score, molecular weight, docking), the objective functions are MPO objectives. The results in the paper therefore show that Beam Enumeration with both Augmented Memory [2] and REINVENT [3, 4] can handle MPO settings. However, we supplement our response with other challenging scenarios that are often encountered in real-world drug discovery efforts and how the Beam Enumeration bias does not have to be detrimental to sample efficiency.
>
> **1. Simultaneous optimization of many properties**
>
> The reviewer brings up a good point about substructures potentially being associated with different properties. In the experiments in the paper, all individual properties to be optimized are combined into a single scalar value during reinforcement learning optimization and this can handle the MPO nature of the problem. However, it is possible to put more weight on certain properties (for example, docking may be more important than molecular weight). By doing so, the substructures would naturally be more biased towards the docking optimization since the weight updates of the Agent depend on this oracle feedback (in which docking now has more onus). This is something that has been done in REINVENT [4] and has shown success.
>
>
> **2. Orthogonal MPO objectives**
>
> We further discuss another scenario where properties to optimize may be orthogonal (they impose a trade-off). Optimizing property 1 may impose a trade-off on property 2 (pareto-front problem). This can be encountered in real-world drug discovery problems such as designing for affinity towards 1 protein while discouraging affinity for another protein. A solution here is to apply curriculum learning [5]. One could decompose the MPO objective into sequential objectives, for instance, to first optimize for affinity to protein 1 and then optimizing to discourage affinity for protein 2 (negative design). Beam Enumeration would then extract substructures in line with each individual objective, removing the problem of “conflicting signals”. However, the substructure themselves would always need to be interpreted relative to each other, when extracting insights.
>
> Overall, there are scenarios where the MPO nature of the problem may impose difficulties in optimization (not just for Beam Enumeration, but generally). However, there are procedures and established techniques that can enable Beam Enumeration to be applied without conflicting substructures.

---

> ### Author Response · Authors · 2023-11-15
> **Response to Reviewer La1F (4/5)**
>
> ### **Q5: Applying Beam Enumeration beyond language-based formulations**
>
> In the first part of our response, we highlight some notable differences in the specific formulations of VAE [6] and JT-VAE [7] models that may prevent **direct out-of-the-box** application of Beam Enumeration. In the second part of the response, we discuss how **concepts and findings** from Beam Enumeration can be used to formulate a combined algorithm with JT-VAE.
>
> VAEs and the JT-VAE learn a latent space encoding of the training data and decode to generate new molecules. One difference is that to achieve goal-directed generation (explicitly satisfy a target objective), an optimization algorithm needs to be applied. In the JT-VAE paper, bayesian optimization is applied to optimize for logP. While showing the capability for optimization, logP is generally not representative of real-world drug design tasks since simply adding carbon atoms would increase the logP. Based on the results, it is unclear how well the optimization process is for multi-objective optimization with physics-based descriptors. Beam Enumeration is built on the observation that it must become increasingly likely to generate favourable molecules if the optimization process is successful. This leads to the second difference, particularly for the JT-VAE. During training, the **JT-VAE learns substructures as a function of the training data** and is not guaranteed that a junction tree exists for an arbitrary molecule (as stated in the paper). This inherently constrains the generalization of the model, especially in the often case where desirable molecules are out-of-distribution to the training data. By contrast, **Beam Enumeration learns substructures as a function of the oracle**, which means that as long as the oracle is optimized over time, extracted substructures are meaningful for that oracle (which should be designed to be informative and correlated with the desired objective). Out-of-distribution generation is handled innately since molecules are generated token-by-token, which allows very different molecules to be generated to the training data. A concrete example of this is in the MK2 case study in Table 2. Very few molecules are generated in the base algorithms which indicate the pre-trained model is not particularly well-suited to satisfy the MK2 docking objective. **The addition of Beam Enumeration to Augmented Memory shifts the distribution and finds substructures that satisfy this oracle.**
>
> We next discuss how **concepts and findings** from Beam Enumeration can be applied to JT-VAE. JT-VAE shows that junction trees can be used to construct local neighbourhoods. Given a favourable seed molecule (a molecule that satisfies the target objective reasonably well), using JT-VAE to propose a local neighbourhood expansion and then scoring these molecules would set-up the possibility to iteratively perform substructure biasing. Specifically, taking the set of all molecules in the local expansion, the most frequent substructures could be used for the next epoch of generation. This could potentially build up larger substructures (sub-graphs) over time.
>
> We end our response by providing an example where Beam Enumeration could be applied out-of-the-box to graph generative models. GraphINVENT [8] is an autoregressive graph model where molecular generation proceeds by “graph action probabilities” which denote whether to add a node or edge. [9] couples reinforcement learning to GraphINVENT, which sets up the potential to directly apply Beam Enumeration by enumerating the most probable graph actions and extracting the substructures. However, based on the PMO [10] benchmark results and recent work on state-of-the-art graph generative models [10, 11], SMILES-based LSTM models are often more sample efficient and better capture the training data distribution. For this reason, we focus on Augmented Memory and REINVENT which are the most sample efficient models in the PMO benchmark.
>
> **We have added a section (Appendix F) discussing the application of Beam Enumeration beyond language-based models.**

---

> > ### Author Response · Authors · 2023-11-15
> > **Response to Reviewer La1F (5/5)**
> >
> > We believe this work represents an important advancement in leveraging the implicit knowledge learned by generative models to achieve enhanced molecular design. The ability to extract meaningful substructures provides actionable insights and the significant gains in sample efficiency could reduce compute resources and time required to discover promising candidate molecules. We hope our responses satisfactorily address the reviewer’s concerns and thank the reviewer again for their time and consideration of our work.
> >
> > Please let us know if you have any further questions!
> > \
> > \
> > Sincerely,
> > \
> > \
> > The Authors
> > \
> > \
> > [1] Polykovskiy, D., et al. Molecular sets (MOSES): a benchmarking platform for molecular generation models. Frontiers in pharmacology, 11 (2020).
> > \
> > \
> > [2] Guo et al., Augmented memory: capitalizing on experience replay to accelerate de novo molecular design, arXiv 2023.
> > \
> > \
> > [3] Olivecrona, M., et al. Molecular de-novo design through deep reinforcement learning. Journal of Cheminformatics 9, 48 (2017).
> > \
> > \
> > [4] Blaschke, T., et al. REINVENT 2.0: An AI Tool for De Novo Drug Design. J. Chem. Inf. Model. 60, 5918–5922 (2020).
> > \
> > \
> > [5] Guo, J., Fialková, V., et al. Improving de novo molecular design with curriculum learning. Nature Machine Intelligence, 4 (2022).
> > \
> > \
> > [6] Gómez-Bombarelli, R., et al. Automatic chemical design using a data-driven continuous representation of molecules. ACS central science, 4 (2018).
> > \
> > \
> > [7] Jin, W., et al. Junction tree variational autoencoder for molecular graph generation. ICML, 2018.
> > \
> > \
> > [8] Mercado, R., et al. Graph networks for molecular design. Machine Learning: Science and Technology, 2 (2021).
> > \
> > \
> > [9] Atance, S. R., et al. De novo drug design using reinforcement learning with graph-based deep generative models. Journal of Chemical Information and Modeling, 62 (2022).
> > \
> > \
> > [10] Gao, W., Fu, T., et al. W. Sample Efficiency Matters: A Benchmark for Practical Molecular Optimization. NeurIPS Track Datasets and Benchmarks, 2022.
> > \
> > \
> > [11] Vignac, C., Krawczuk, I., et al. Digress: Discrete denoising diffusion for graph generation. ICLR, 2023.

---

> > ### Comment · Reviewer_La1F · 2023-11-21
> > **Re: Response to Reviewer La1F (4/5)**
> >
> > Thank you for the added discussion.
> > This is very helpful, and I believe the added section in Appendix F would be useful for potential readers.

---

> ### Comment · Reviewer_La1F · 2023-11-21
> **Re: Response to Reviewer La1F (3/5)**
>
> Thank you for the clarification and I mostly agree with your points.
> Having said that, as Beam Enumeration makes use of (and depends on) high probable tokens/substructures, when different substructures are associated with different properties/objectives, the approach may face unique challenges that do not necessarily rise in other MPO problems that do not utilize such "Beam Enumeration" approach for molecule generation.
> I believe this is an issue that needs further investigation.

---

### Official Review · Reviewer_3svX · 2023-11-16

**Soundness:** 4 excellent
**Presentation:** 3 good
**Contribution:** 4 excellent
**Rating:** 8
**Confidence:** 4

**Summary:**

Generative molecular design has transitioned from theoretical validation to practical utility, evidenced by a recent surge in papers featuring experimental confirmation. In this evolving landscape, addressing challenges related to explainability and sample efficiency becomes crucial for optimizing high-fidelity oracles efficiently and providing valuable insights to domain experts. This paper introduces Beam Enumeration as a solution, aiming to exhaustively enumerate the most probable sub-sequences from language-based molecular generative models. The method demonstrates the extraction of meaningful molecular substructures. When integrated with reinforcement learning, these extracted substructures contribute to enhanced explainability and improved sample efficiency through self-conditioned generation. Notably, Beam Enumeration is adaptable to any language-based molecular generative model and significantly enhances the performance of the Augmented Memory algorithm, a recent state-of-the-art achievement in sample efficiency on the Practical Molecular Optimization benchmark. The joint focus on explainability and sample efficiency makes Beam Enumeration a pioneering method in the field of molecular design.

**Strengths:**

- The paper is well-written, well-organized, and easy to follow.
- The idea is quite intuitive and straightforward. It is based on the state-of-the-art language model--augmented memory.
- The paper also emphasizes sample efficiency, which is an essential problem in practical molecular design. It does not only require the optimization ability but also the oracle efficiency. The oracle can be computationally expensive and the bottleneck of the modern molecular design.
- The whole setup is realistic. Novel Oracle efficiency-based metric is designed.
- Code is easy to read and is public, guaranteeing the reproducibility.
- The experimental results are thorough and solid. The proposed beam enumeration significantly outperforms REINVENT (the strongest baseline in the existing benchmark). It also exhibits explainability during the generation.

**Weaknesses:**

- It would be great if authors could incorporate more baseline methods.
- Some minor issue, e.g., repetitive statements, like
"To address this, Gao et al. (Gao et al. (2022) proposed the Practical Molecular Optimization (PMO) (Gao et al. (2022) benchmark." Too many "Gao et al".
- Do you use the same setup with PMO? If not, it seems unfair to compare with PMO's best baseline REINVENT.

**Questions:**

Is it possible to incorporate MD-based simulation as oracle? How long does it take for each oracle call? Can you provide more details?

---

> ### Author Response · Authors · 2023-11-17
> **Response to Reviewer 3svX (1/2)**
>
> We thank the reviewer for their feedback and we will address each question individually.
>
> ### **Q1: More baseline methods**
>
> In the Augmented Memory [1] work, their PMO [2] results show that it is the new state-of-the-art in sample efficiency (REINVENT [3, 4] being ranked 2nd). For this reason, we focused on applying Beam Enumeration to Augmented Memory and REINVENT to demonstrate that their sample efficiency can be further improved. Our goal in the present work is to push the limits of sample efficiency and demonstrate synergy with explainability by proposing the general Beam Enumeration framework.
>
> However, Beam Enumeration is not limited to Augmented Memory and REINVENT. We have added a section (Appendix F) which discusses how Beam Enumeration could be extended to graph-based autoregressive models and other sub-graph methods. In the future, we will explore extensions of Beam Enumeration to other classes of generative models, and other language-based architectures.
>
> ### **Q2: Repetitive statements**
>
> We have removed the redundant “Gao et al.”.
>
>
> ### **Q3: Same set-up as PMO**
>
> In the Augmented Memory [1] work, the authors benchmarked their model on the PMO [2] benchmark, following the same set-up (10,000 oracle calls and across 23 benchmarking tasks). In the Beam Enumeration manuscript, we do not benchmark performance on the PMO benchmark. Our reasoning is because none of the tasks in the PMO benchmark involve optimizing physics-based oracles (such as molecular docking). These are the oracles that **demand** sample efficiency since they are computationally expensive. Importantly, **almost all works that have reported experimental validation** of a generated molecule used physics-based oracles [5-10]. This is also the case for the current most advanced clinical candidate (phase 2 clinical trials) designed by a generative model [11]. For this reason, we focused on optimization tasks involving physics-based oracles. We make an effort to show broad applicability by performing 3 drug discovery tasks, each involving different protein receptors. The comparisons between Augmented Memory [1], REINVENT [3, 4], and the addition of Beam Enumeration to both, are fair, as all experiments enforced an oracle budget of 5,000 calls.
>
> ### **Q4: MD-based simulation as oracle**
>
> MD-based oracles are much more computationally expensive compared to molecular docking. For example, free energy perturbation [12] which is much more accurate than docking in predicting binding affinity, can take 6 GPU hours per molecule. A generated batch of molecules could be parallelized across multiple GPUs. **However, we want to be prudent in making definitive claims without empirical evidence and thus, our answer is that Beam Enumeration with Augmented Memory [1] is not sample efficient enough.**
>
> This is also one of the motivations of introducing the Oracle Burden metric. With these higher-fidelity oracles, we may not need to generate a large number of ideas to triage, as is the standard practice currently; optimizing for molecular docking and then selecting a subset for MD simulations. With higher-fidelity oracles, generating even 10-100 favourable molecules may be sufficient. Beam Enumeration is a step towards achieving this aspiration.
>
> We believe Beam Enumeration is a valuable general framework to extract explainability and improve sample efficiency in current state-of-the-art (based on sample efficiency) generative models. We want to thank the reviewer again for their time and feedback.
> \
> \
> Please let us know if you have any further questions!
> \
> \
> Sincerely,
> \
> \
> The Authors

---

> ### Author Response · Authors · 2023-11-17
> **Response to Reviewer 3svX (2/2)**
>
> [1] Guo et al., Augmented memory: capitalizing on experience replay to accelerate de novo molecular design, arXiv 2023.
> \
> \
> [2] Gao, W., Fu, T., et al. W. Sample Efficiency Matters: A Benchmark for Practical Molecular Optimization. NeurIPS Track Datasets and Benchmarks, 2022.
> \
> \
> [3] Olivecrona, M., et al. Molecular de-novo design through deep reinforcement learning. Journal of Cheminformatics 9, 48 (2017).
> \
> \
> [4] Blaschke, T., et al. REINVENT 2.0: An AI Tool for De Novo Drug Design. J. Chem. Inf. Model. 60, 5918–5922 (2020).
> \
> \
> [5]  Zhavoronkov, A., et al. Deep learning enables rapid identification of potent ddr1 kinase inhibitors. Nature biotechnology, 37 (2019).
> \
> \
> [6] Yoshimori, A., et al.. Design and synthesis of ddr1 inhibitors with a desired pharmacophore using deep generative models. ChemMedChem, 16 (2021).
> \
> \
> [7] Ren, F., et al. Alphafold accelerates artificial intelligence powered drug discovery: efficient discovery of a novel cdk20 small molecule inhibitor. Chemical Science, 14 (2023).
> \
> \
> [8] Li, Y., et al. Discovery of potent, selective, and orally bioavailable small-molecule inhibitors of cdk8 for the treatment of cancer. Journal of Medicinal Chemistry (2023).
> \
> \
> [9] Salas-Estrada, L., et al. De novo design of κ-opioid receptor antagonists using a generative deep learning framework. J. Chem. Inf Model., 63 (2023).
> \
> \
> [10]  Zhu, W., et al. Discovery of novel and selective sik2 inhibitors by the application of alphafold structures and generative models. Bioorganic & Medicinal Chemistry, 91 (2023).
> \
> \
> [11] https://www.clinicaltrialsarena.com/news/insilico-medicine-ins018055-ai/?cf-view
> \
> \
> [12] Wang, L., et al. Accurate and reliable prediction of relative ligand binding potency in prospective drug discovery by way of a modern free-energy calculation protocol and force field. Journal of the American Chemical Society, 137 (2015).

---

### Author Response · Authors · 2023-11-17
**Global Response (1/1)**

We want to thank all the reviewers for their time and effort in providing valuable feedback and constructive criticism on our work. We hope our individual responses to each reviewer adequately addresses their questions. Please let us know if there are further questions!

In this global response, we want to highlight all the changes to the updated manuscript after incorporating feedback from all reviewers:

1. **Manuscript Typos**

Missing asterisks added in Table 2 for **REINVENT DRD2 Oracle Burden (100) Beam Structure 15** and **Augmented Memory MK2 Oracle Burden (100) Beam Scaffold 15**. The conclusions drawn are not affected.

2. **Pseudo-code for Beam Enumeration Integrated with Augmented Memory**

An additional pseudo-code block has been added to Appendix B.5.2 showing *where* Beam Enumeration is executed in the context of Augmented Memory, as suggested by *Reviewer d4Xh*.

3. **Synthetic Accessibility (SA) Score Annotated in Figure 3**

The SA score is now annotated for all generated molecules and the reference ligand in Figure 3, as suggested by *Reviewer d4Xh*.

4. **Beam Enumeration beyond Language-based Models**

An extra Appendix section (F) has been added to discuss how Beam Enumeration could be extended to generative models beyond language-based formulations, as suggested by by *Reviewers La1F and d4Xh*.

5. **Improved clarity in the “Self-conditioned Generation” Section**

We have added more details about how Beam Enumeration filters the generated molecules for improved clarity, as suggested by *Reviewer wYf5*.

6. **Beam Enumeration Contribution Rephrasing**

We have rephrased Beam Enumeration's contribution from the first method to jointly address sample efficiency and explainability to showing how sample efficiency and explainability can be made synergistic, as suggested by *Reviewer d4Xh*.

7. **Discussion about Diversity in the Main Text**

We now more explicitly discuss diversity in section `Quantitative Analysis: Sample Efficiency` in the main text, as suggested by *Reviewer La1F*. Previously, additional results and details around diversity were only in the Appendix.
\
\
We thank all reviewers again for their feedback and we are available for further questions!
\
\
Sincerely,
\
\
The Authors

---

### Meta-Review · Program_Chairs · 2024-01-15

**Metareview:**

The paper proposes a method for generative molecular design that enumerates the most probable sub-sequences from language-based molecular generative models. It obtains state of the art results in established benchmarks while also offering explainability in the generation process. The reviewers found the contribution of the paper important and with good empirical performance. They raised concerns regarding the paper being incremental over “Augmented memory” of Guo et al., as well as lacking a theoretical basis. The positives outweigh the negatives and the paper is accepted for publication.

**Justification For Why Not Higher Score:**

The paper very heavily builds upon recent related work.

**Justification For Why Not Lower Score:**

The paper shows state of the art results over baselines.

---

### Decision · Program_Chairs · 2024-01-16

Accept (poster)